# Information dynamics and the emergence of high-order individuality in ecosystems
Hardik Rajpal [1,2,3] ✉, Clem von Stengel[4,5], Pedro A. M. Mediano [1,6,7], Fernando E. Rosas [1,8,9,10], Eduardo Viegas [1,2,11], Pablo A. Marquet [12,13,14,15] & Henrik J. Jensen [1,2,11]

At what level does natural selection occur? When considering the reproductive dynamics of interacting and mutating agents, it has long been debated whether selection is better understood by focusing on the individual or if hierarchical selection emerges as a consequence of joint adaptation. Despite longstanding efforts in theoretical ecology, there is still no consensus on this fundamental issue, most likely due to the difficulty in obtaining adequate data spanning a sufficient number of generations and the lack of adequate tools to quantify the effect of hierarchical selection. Here, we capitalise on recent advances in information-theoretic data analysis to advance this state of affairs by investigating the emergence of high-order structures- such as groups of species- in the collective dynamics of the *Tangled Nature* model of evolutionary ecology. Our results show that evolutionary dynamics can lead to clusters of species that act as a self-perpetuating group that exhibits greater information-theoretic agency than a single species for a broad range of stable mutation rates. However, this higher-order organization breaks down for mutation rates close to the error threshold, where increased information processing is observed at the level of a single species. For mutation rates higher than the error threshold, no stable population of species are observed in time, and all individuality is lost in the ecosystem. Overall, our findings provide quantitative evidence supporting the emergence of higher-order structures in evolutionary ecology from relatively simple processes of adaptation and reproduction.

The dynamics of life around us are characterised by a plethora of complex, interdependent relationships. These relations span across different scales, from cells to organisms, and even the biotic and abiotic environment in which they exist. Adaptation through selection is widely agreed to be the motor behind both macro-evolution, as documented in the fossil record[1], and micro-evolution, as observed in microbial experiments[2]. However, the collective and mutually interdependent nature of evolutionary dynamics raises a critical question: What is the level at which selection *effectively* operates on a set of entangled co-adapting entities?

Since Darwin, the standard view is to assume individual organisms as the drivers of evolutionary change. An extreme version of this view is to regard individual genes as the selective unit, still acting through the organism to express changes in the behaviour[3,4] (for an insightful discussion, see the book by Jablonka and Lamb[5]). Alternative perspectives, where selection also acts at the level of higher-order entities (such as groups of species, ecosystems, or even the whole biosphere) in a hierarchical fashion, have been the subject of long debates[1,5–7]. Crucially, there is a possibility that non-reproducing higher-order systems may be subjected to a different kind of selection. For instance, theories of persistence selection argue that evolution selects for the persistence of interaction structures and processes that are implemented by various entities: genes, cells, individual organisms or species[8–11]. This view amounts to a drastic

[1]Centre for Complexity Science, Imperial College London, London, UK. [2]Department of Mathematics, Imperial College London, London, UK. [3]The Alan Turing Institute, London, UK. [4]Alignment of Complex Systems, Centre for Theoretical Study, Charles University, Prague, Czechia. [5]Department of Ecology, Faculty of Science, Charles University, Prague, Czech Republic. [6]Department of Computing, Imperial College London, London, UK. [7]Division of Psychology and Language Sciences, University College London, London, UK. [8]Department of Informatics, University of Sussex, Brighton, UK. [9]Department of Brain Sciences, Imperial College London, London, UK. [10]Centre for Eudaimonia and human flourishing, University of Oxford, Oxford, UK. [11]Department of Computer Science, Tokyo Institute of Technology, Yokohama, Japan. [12]Facultad de Ciencias Biológicas, Pontificia Universidad Católica de Chile, Santiago, Chile. [13]The Santa Fe Institute, Santa Fe, NM, USA. [14]Centro de Modelamiento Matemático (CMM), Universidad de Chile, International Research Laboratory, Santiago, Chile. [15]Instituto de Sistemas Complejos de Valparaíso (ISCV), Valparaíso, Chile. ✉e-mail: h.rajpal15@imperial.ac.uk

shift in the way we conceive selection and evolution more broadly. Such views raise important questions regarding how group-level dynamics may make the fate of individuals depend not only on their genetic information, but also on how this is integrated into a larger system of interacting entities. Put simply, these views suggest to shift the focus from the singer (i.e. organism, species, etc.) to the song[9] (i.e. relationships among the organisms or species).

A way to advance these questions is to avoid reducing them to a dichotomous choice between selection at the basic level of reproduction versus at some higher collective level, and instead consider that different types of selection pressure may be working in tandem at different levels of ecosystemic organisation. However, in order to pursue such a view, it is crucial to have quantitative tools that are capable of identifying and differentiating degrees of (possibly time- and scale-dependent) cooperative selective structure, which could be used to investigate these ideas on empirical or simulated data. Although there have been efforts to quantify multi-level selection using statistical analysis[12–14], they suffer from limitations of misspecification when it comes to identifying the mechanisms of such a selection[15], as multiple mechanisms can result in the observed statistical measures. On the other hand, models that focus on mechanisms can simulate evolutionary systems that exhibit emergent higher-order selection[16,17]. However, metrics that track the strength of this selection in the models often rely on the knowledge of the underlying interaction structure or assume temporal stationarity. These conditions are usually not met in laboratory or field experiments. Therefore, there is a need for model-agnostic tools that can quantify the strength of higher-order interactions from population-level datasets.

In this paper, we address these questions by using information-theoretic tools to investigate the emergence of higher-order interactions in a model of co-evolving species. Information-theoretic measures such as entropy, mutual information and synergy have been useful in quantifying diversity[18], interdependence[19] and complementarity[20] in systems with many interacting parts, respectively. Moreover, recent theoretical advancements have introduced promising methods based on information storage and predictive information to quantify individuality[21]. The primary thesis behind the information-theoretic approach to individuality is that if a group of individuals can cohesively persist in time (i.e. their future population can be predicted using their combined past), then they can better adapt and survive. The enhanced persistence among groups of individuals can arise due to interdependencies that stabilize the population of the group through positive and negative feedback, being in line with the persistence theories of evolution[22,23] that propose the selection of lineages[24] and interaction structures[10]. Despite its attractiveness, it is currently unknown if information individuality can spontaneously emerge in a co-evolutionary setting. Krakauer et al.[21] highlight the need for such a study in their discussions on the implications of the information theory of individuality.

Building on these information-theoretic foundations, here we investigate the factors that support higher-order individuality in ecosystems by analysing co-evolving species based on the well-studied Tangled Nature (TaNa) model[25]. The TaNa model has been used to study multiple aspects of co-evolutionary dynamics, including the observed species abundance curves[26], the entropy of species distributions[27], hierarchical organization of ecosystems[28], and the statistics of mass extinctions[29]. The TaNa model establishes the dynamics of the population of multiple species co-evolving over time. In the model, the fitness of each species depends on the population of species with which it interacts, as well as the total population of the ecosystem. Based on this fitness, species produce offspring that can undergo mutation, thus introducing new species into the ecosystem. Here, each species is represented by a pangenome. Thus, a mutation in an offspring of the species introduces a member of a completely different species than its parent. As the system evolves over many generations, it gets into punctuated stable states or quasi Evolutionary Stable Strategies (q-ESS). These metastable states become longer in duration[30] and support more mutualistic interactions[28] as systems evolve to more resilient configurations. Further details of the model are presented in Section 'Methods'. Here, we use the co-evolutionary dynamics of the TaNa model to investigate whether there are conditions under which the evolution selects for the interactions at the level of groups of species instead of a single species. We explore this question from various complementary angles, including analyses of information-theoretic 'individuality'[21], integrated information[31,32], and other measures of information dynamics[33,34]. We explain the applicability of the information theoretic measures, using a schematic diagram in a simple case of two co-evolving species (see Fig. 1).

The analysis of the model highlights the crucial role of mutation rates in enabling adaptability and higher-order organization among the co-evolution of species. We find that groups of species can persist cohesively to maximize their joint information individuality for a broad range of mutation rates. As mutation rates increase beyond this range, this cohesion is lost, and maximum information individuality is observed at the scale of a single species near the Eigen error threshold[35]. For mutation rates higher than the error threshold, no temporally stable or persistent population of species are observed as mutations get decoupled from fitness and survival. The role of mutation rates in affecting adaptability and survival under changing environmental conditions has been a topic of research interest[36,37]. Some researchers have argued that environmental conditions, such as nutrition availability[38] and temperature[39], can alter mutation rates. Others have suggested that the effect of such factors is limited in the case of sexual reproduction[40]. Our work provides new evidence on the role of moderate mutation rates in facilitating higher-order individuality. Crucially, this higher-order phenomenon is observed in the evolutionary dynamics arising from a simple underlying mechanism of reproduction and mutation, which does not include explicit group-level interactions[41] or interaction delays[42]. Furthermore, the analysis does not assume a preferred level of selection and allows quantification of the degree of individuality at all levels. Thus, these results identify the spontaneous emergence of groups of cooperating species in a model of co-evolving species.

## Results

The results presented in this section are obtained from 10,000 simulations (of $10^5$ generations each) each for different values of the mutation rates and calculating ensemble averages over the results (see the 'Methods').

We begin by visualising the dynamics of the model in terms of existing species at each generation (see Fig. 2). It can be observed that for low mutation rates, the system quickly gets into a metastable regime where only a subset of species exists. These metastable states are disrupted by reorganization regions where existing species die out and others come into being. The reorganization is often initiated by the appearance of a disruptive mutant, which negatively impacts the population of existing species. The population of species undergoes mutations in response to the mutant, enabling exploration the space of all feasible species to identify another subset of species, with balanced interactions, to stabilize their population (see Fig. 6 in ref. 28). Finally, at the end of this exploration, a new metastable state is established with a distinct subset of species. As mutation rates increase, the duration of the reorganization region increases. Eventually, beyond $p_{mut} > 0.05$, no metastable states are seen to be established.

### Error threshold and population diversity

As a first step in our analyses, we investigated how the total population and diversity of species are affected by the mutation rate of the evolving agents. The diversity is calculated using the exponential of the Shannon index of the population distribution. This measure is also known as Hill's Diversity[43]. Special attention is paid to the dynamics observed in the vicinity of the 'error threshold,' which is the limit on the mutation rate for species beyond which the existing population of species provides no information about their future[44,45].

Our simulations show that the total population progressively decreases with mutation rate, with a sudden drop after mutation rate 0.04 (see Fig. 3). In contrast, the diversity of species increases with the mutation rate and peaks at 0.04. This helps us identify the error threshold associated with the

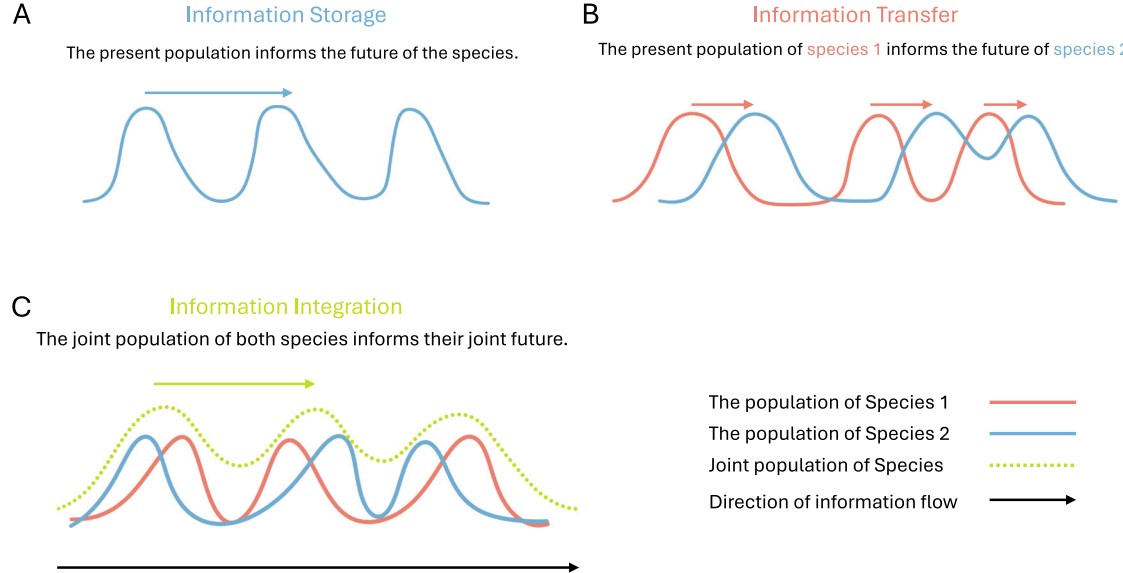

**Fig. 1 | The figure provides a schematic representation of the information-theoretic measures of storage, transfer and integration to populations of two co-evolving species. A** shows a periodically oscillating population of species. Here, the past population can accurately predict the future population of the species, implying that the information about the future of the population has been stored in the past population. This information storage is measured using the mutual information between the past and the future population of the species. **B** illustrates the measure of information transfer. Here, species 2 asymmetrically responds to the peak of species 1 by growing its population. This enhanced predictability of the future population of one species by the knowledge of the past population of the other is measured using Transfer Entropy. **C** shows a joint coherence between two aperiodically varying populations of species. Here, the joint population behaves more predictably than the population of any species alone. This global coordination is measured using an improved measure of integrated information. The measure quantifies the excess mutual information between the past and the future of the joint population as compared to the sum of the information storage of each species.

model as close to 0.05, beyond which no stable population of species are observed.

By tracking the proportion of time spent by the system in the metastable states and the reorganization regime (see Fig. 3), we identify a region close to the threshold where the system exhibits a non-trivial behaviour. For mutation rates between 0.04 and 0.05, it is seen that the number of generations spent by the system in reorganization between the metastable states increases. We refer to this range of mutation rates as the transition region. At a mutation rate of 0.042, the system spends an equal amount of time searching for stable configurations as staying in them, thereby enhancing the variability at the level of individual species.

## Information individuality

After identifying the error threshold, we investigated the potential presence of higher-order individuality by estimating the organismal 'individuality scores' for groups of different numbers of species via the framework introduced by Krakauer et al.[21]. Briefly, this approach proposes an 'individuality score' that represents the degree to which a group behaves as a single entity in the sense that it is maximally self-predictive (i.e. the group's future population is maximally predicted from knowledge of the group's past population). According to this framework, if a group of species achieves a higher individuality score than a single species, then the group is able to leverage its interactions to adapt to the changing environment. This enables the group to persist for longer as an evolutionary unit than a single species alone.

Individuality scores were calculated over 10,000 different combinations for each group size (i.e. the number of species considered in a group, which we refer to as *scale*)—singlets (scale = 1), dyads (scale = 2), triplets (scale = 3), and so on. These scores were then normalised based on the size of the group. Results reveal that the mutation rates play an important role in modulating higher-order individuality in coevolution. In Fig. 4, we interpret the results relatively across the scales at any given mutation rate. For instance, if increased normalised individuality scores are observed at higher scales than on a scale of 1, we identify higher-order organizations to be present in the system. Increased individuality at higher scales suggests groups of species interacting together in a manner that enhances the group's persistence. At low mutation rates, organisation into cliques of higher scales becomes apparent, and higher individuality scores are observed for scales between 3 and 9 (compared to scale 1). Though this enhanced individuality at higher scales flattens out for mutation rates 0.03 and 0.04, the higher-order organisation still persists (i.e. average individuality scores for higher scales are greater than the individuality score for scale 1). For mutation rates in the transition range, the higher-order organisation is lost, and the single species level becomes the most optimal self-predicting scale.

Formal definitions of the individuality score presented here can be found in the Methods section 'Information individuality'. These results are also replicated on other proposed measures of individuality that take into account the effect of the environment on the group of species, which confirms the presence of higher-order organisation (see Supplementary Information B), irrespective of the measure used.

The normalised individuality scores presented here can be interpreted as information carrying capacity[46], predictive information[47] or information storage[48]. All of these definitions refer to the amount of information in the past of the system that can be used to predict its future. In summary, the individuality scores quantify the average level of persistence exhibited by groups of species for every scale. We visualize the individuality scores at scale 6, a scale that exhibits higher organismal individuality as compared to scale 1 for mutation rates below the transition region. (see Fig. 5).

It can be seen that the individuality score of species at scale 6 peaks for low mutation rates and then decreases monotonically through the transition region before going to zero near the error threshold. However, at the scale of a single species, peak individuality is observed in the transition region. A clear crossover can then be observed in the transition region, where higher-order organisation loses individuality while the singular species gains organismal individuality.

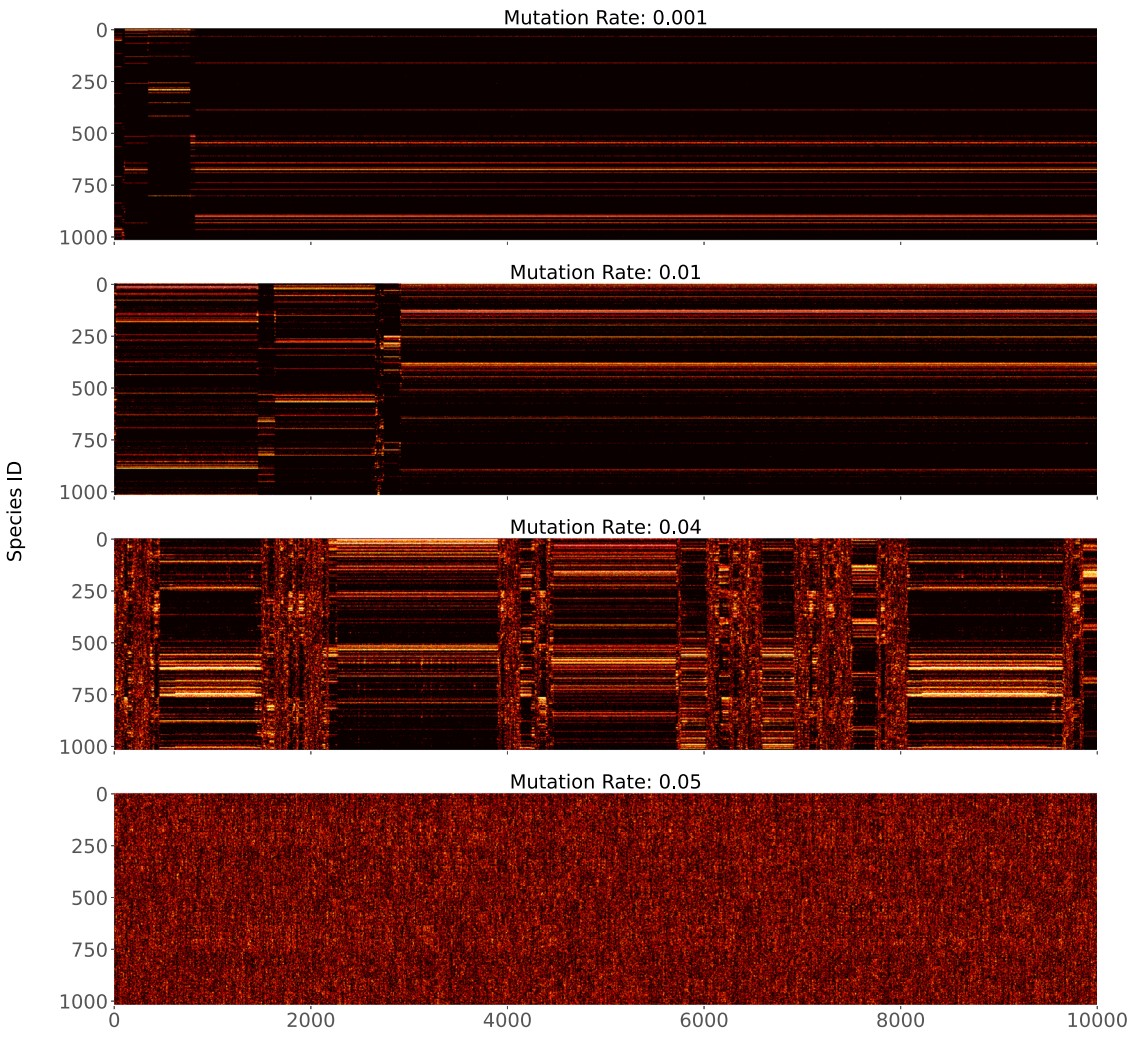

**Fig. 2 | The figure shows TaNa evolution recorded for 10,000 generations for four different mutation rates.** The bright dots indicate the existing species out of the $2^{10}$ available species, identified by a unique Species ID. It can be seen that metastable states are present for $p_{mut} < 0.05$. These states are separated by a transition regime, which becomes larger with increasing mutation rates. The simulations are generated using standard model parameters, $L = 10$, $\Theta = 0.25$, $p_{kill} = 0.2$, $k = 33$ and $\mu = 1/143$.

In the Supplementary Information E, we show that the model selects for a very specific interaction structure at scale 6 to support high individuality. The structure includes a high degree of asymmetry in the interaction amongst the species with a salient predator-prey interaction. This asymmetry decays during the transition region, where the higher-order organization is lost. To directly establish the role of the interaction structure, we simulated a neutral version of the Tangled Nature model. Here, we allowed all species to interact with each other with the same weight. We do not see any higher-order interaction emerge beyond scale 2. These results highlight the role of an asymmetric interaction structure, with positive and negative feedback, in the emergence of higher-order organization among co-evolving species.

In the following section, we explore species-environment information transfer and integration to explore dynamics in the error-transition region further.

**Interaction between species and their environment**

As a last step of our analysis, we characterise the species-environment interactions using measures of information transfer and integration. Information transfer quantifies the excess predictability provided by the population of a species or a group of species about the environment and vice versa. Additionally, Information integration quantifies the joint predictability of the future population of the species-environment system (see Fig. 1 for a schematic representation). Here, the *environment* is defined using the informational boundary of the individual considered. For instance, if a subset of $K$ species is regarded as an individual, then this boundary implies the rest of the species as its biotic environment (see Methods section 'Information individuality' for more details). As discussed before, we compare these interdependencies at the level of single species (i.e. scale 1) against scale 6, which our previous analysis highlighted as the scale that showed higher individuality than scale 1 for most mutation rates below the error transition region. We estimate these measures for various mutation rates in order to identify how a species (or a group of species) interact with the environment near the error threshold.

For the purposes of this analysis, we use the population of a single species (for scale 1), a vector of the population of species (for scale 6) or the total population of all remaining species (for environment) as random variables to estimate the information measures discussed below. The population of the environment is calculated as the difference between the total population of the ecosystem and the total population of the species (or the group of species). Further details about these estimations are provided in the Methods section 'Information-theoretic measures'.

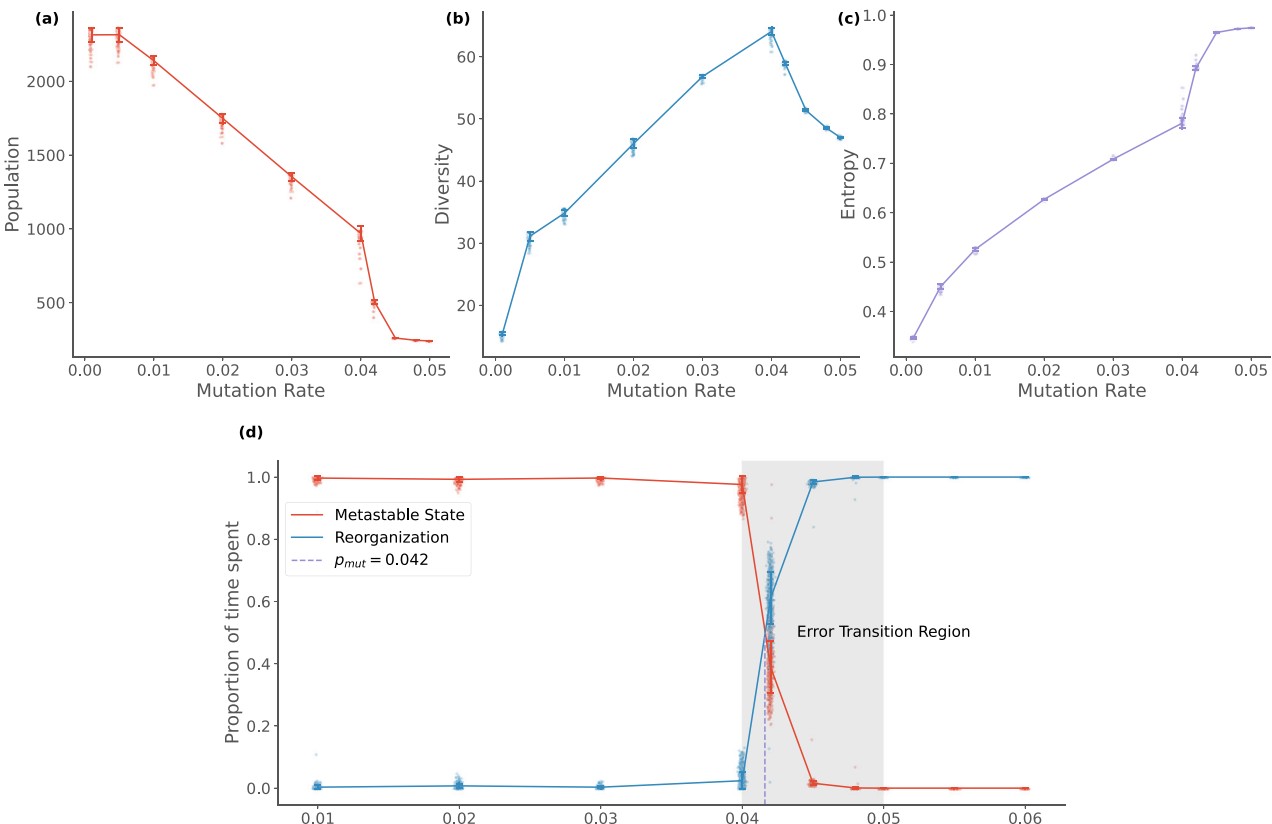

**Fig. 3 | Effect of mutation rate on the population dynamics. a** As mutation rate increases the overall population decreases monotonically, while **b** the diversity of species increases until it reaches a peak value for a mutation rate of 0.04. **c** shows the Shannon entropy of the population size distribution across the existing species. It can be seen that the entropy gradually increases until the mutation rate of 0.04. Beyond which, a sudden jump to a uniformly distributed population distribution is observed (Entropy ≈ 1). For mutation rates higher than 0.04, the time spent by the system in the reorganization region increases (**d**). This area is highlighted as the *Error Transition* region in the lower panel. For a mutation rate of 0.042, the system spends almost the same number of generations in reorganization between the metastable states as it does in them. The lines show the ensemble average values of the measures calculated from the simulations of the Tangled Nature Model, using the standard parameters described in the methods. The standard deviation of measures from the mean is shown using the error bands around the lines.

First, we focus on the information transfer measured by Transfer Entropy (TE)[49] between species and environment. This directed measure estimates the information that a *source* variable provides about the future state of a *target* variable, over and above the information contained in the present state of the target itself. In short, TE quantifies the statistical influence that species and environment have upon each other's future populations.

The information transfer to and from the environment also varies differently at the two levels of organization (see Fig. 6). For scale 1, the information flow is predominantly from the environment to the single species, but decreases with increasing mutation rate until the transition region. However, in the transition region, information flow peaks in both directions, but at either end of the region.

The trends of information flow look different for the higher-order organisation of species (Fig. 6). For scale 6, a significant amount of information flow exists in both directions, though the environment-to-species information flow is larger until the transition region. In this case, the information flow peaks in both directions and stays almost equal throughout the transition region. Thus, a near-symmetric information flow is established during the transition region.

The transfer entropy from the environment to the group of species, in the case of the Tangled Nature model, is mathematically equivalent to another kind of individuality measure, the *Environment determined individuality*[21]. This measure quantifies how much of the persistence of the group is predicted by the environment beyond what the group can predict. In essence, the interactions with the environment guide the group's persistence. However, this effect is much weaker compared to organismal

individuality scores. When we compare this quantity across the two scales, we can see that in contrast to organismal individuality, scale 6 still possesses some environment-determined individuality in the error-transition region. On the other hand, the single species exclusively peaks in both individuality scores in the transition region.

Finally, we look at the integrated information measured using $\Phi$[R34,50], which quantifies the level of integration among species and the environment at the single species level (scale 1). This measure would be zero if the species and the environment do not interact, and the future of the species (or environment) depends only upon its current population. The measure is positive if both species and the environment provide some information about their joint future, which cannot be obtained independently from either one (see Methods Information transfer and integration for details).

Here, we also recover a peak of integrated information during the transition region (see Fig. 7) for scale 1. These results confirm that at the level of single species, a peak in different modes of information processing: information storage (organismal individuality), transfer (Environmental determined individuality) and integration is observed in the error-transition region. This finding is highly suggestive of a critical phase transition[33,51].

Overall, this section presents two major findings: First, the species-environment interactions at the level of a single species differ from the higher levels of organisation. This difference is particularly significant near the error threshold, where the higher-order groups of species lose organismal individuality while still sharing some influence from the environment. Meanwhile, at the single-species level, both organismal and environment-determined individuality peak during the error-transition

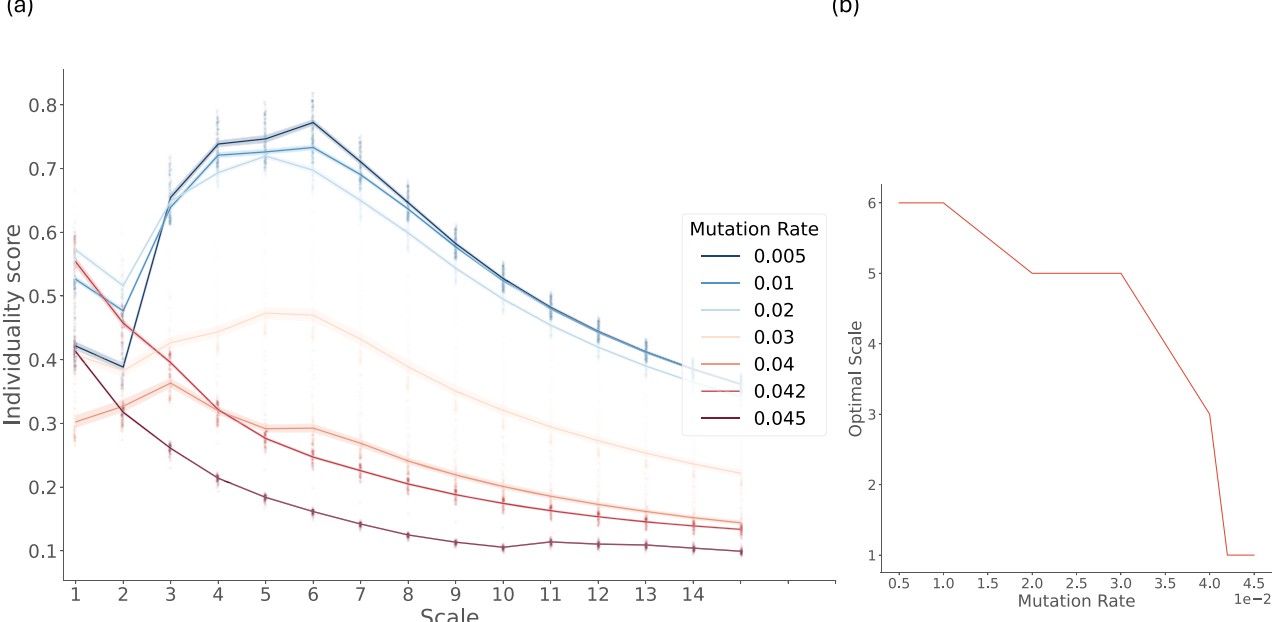

**Fig. 4 | Average Individuality scores observed at different scales and mutation rates. a** Increased information individuality is observed for higher scales (between 3 and 9) and low mutation rates ($p_{mut} \in [0.005, 0.04)$). Conversely, in the transition region (mutation rates of 0.042 and 0.045), single species (scale = 1) have the highest individuality scores. The lines show the ensemble average values of the measures calculated from 10,000 simulations of the Tangled Nature Model, using the standard parameters described in the methods. The standard deviation of measures from the mean is shown using the error bands around the lines. **b** We highlight the scales with optimal information individuality for each mutation rate. Higher order organization at scales 5-6 is observed until $p_{mut} = 0.03$. For $p_{mut} = 0.04$, we do observe enhanced individuality at scale 3; however, beyond this rate, the optimal scale is observed at the scale of a single species.

**Fig. 5 | Average Individuality scores for the two scales of organization—Single species (scale 1) and Higher Order (scale 6)—varying with mutation rates.** Individuality at scale 6 is higher than scale 1 for low values of mutation rates less than 0.04, i.e. before the error transition region. Individuality at scale 1 peaks in the error transition region before decreasing to zero beyond the error transition. The lines show the ensemble average values of the measures calculated from the simulations of the Tangled Nature Model, using the standard parameters described in the methods. The standard deviation of measures from the mean is shown using the error bands around the lines.

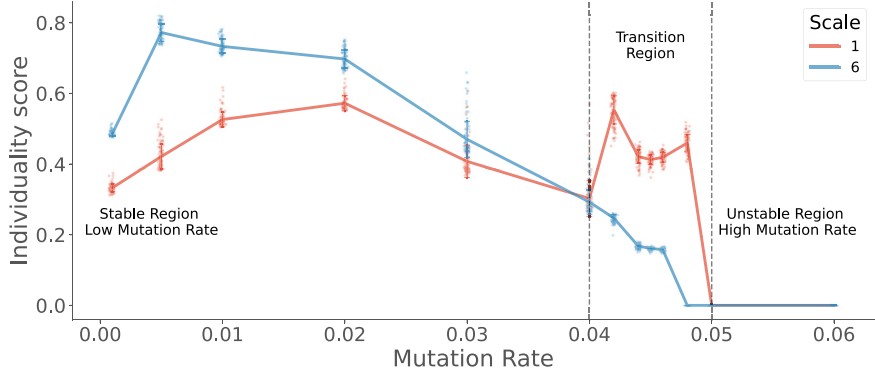

region. This leads to the second finding of this analysis. Enhanced information processing among the parts of the system during the phase transitions is in line with the literature on criticality in complex systems[32,33,51–53]. We find that information storage, transfer and integration peaks at the single species level during the error transition region. These peaks occur near the mutation rate $\approx 0.042$ when the system spends an equal amount of time in and between q-ESS. This metastable switching between the two regimes increases the entropy of the temporal behaviour at the group level (as evidenced by the reduced duration of metastable states). However, information individuality (predictability) is seen to be increased at the level of individual species for the mutation rates close to the error-transition. This exhibits a trade-off between the higher-order organization (at low mutation rates) and independent single-species level predictability (at high mutation rates, see Fig. 5). The stability afforded by lower mutation rates affords the existence of stable group-level interactions that co-evolve and persist in time. As mutation rates increase, this group-level coordination deteriorates and enhanced predictability is observed at the level of a single species

during the shorter metastable states observed in the transition region (see Fig. 2).

## Discussion

This paper investigated hierarchical selection in the Tangled Nature (TaNa) model of ecological evolution[25]. In contrast to prior work, our work leverages recent advances in information theory[21] to provide analyses that are quantitative and mathematically rigorous, which allows us to objectively estimate the degree of high-order individuality in this self-organising evolutionary system. Crucially, these tools provided evidence of how relatively simple processes of adaptation and selection pressure can result in the emergence of groups of cooperating species that act as effective units within the evolutionary process. Specifically, our results identified signatures of higher-order organization in the simulated ecosystems, with groups of 3–9 species acting as individual evolutionary units. We argue that the emergence of this higher-order organization implies the presence of persistence selection at the higher level. Interestingly, the dominance of multi-species

**Fig. 6 | Information flow between species and environment as measured using Transfer Entropy for various mutation rates.** The top panel shows the variation at the scale of a single species (scale 1), and the bottom panel shows the higher-order grouping (scale 6). Generally, more information flows from the environment to the species. Except for scale 6 in the transition region, where an almost symmetric information flow is observed in both directions. A peak in the transfer entropy is seen at both scales during the transition region. The lines show the ensemble average values of the measures calculated from 10,000 simulations of the Tangled Nature Model, using the standard parameters described in the methods. The standard deviation of measures from the mean is shown using the error bands around the lines.

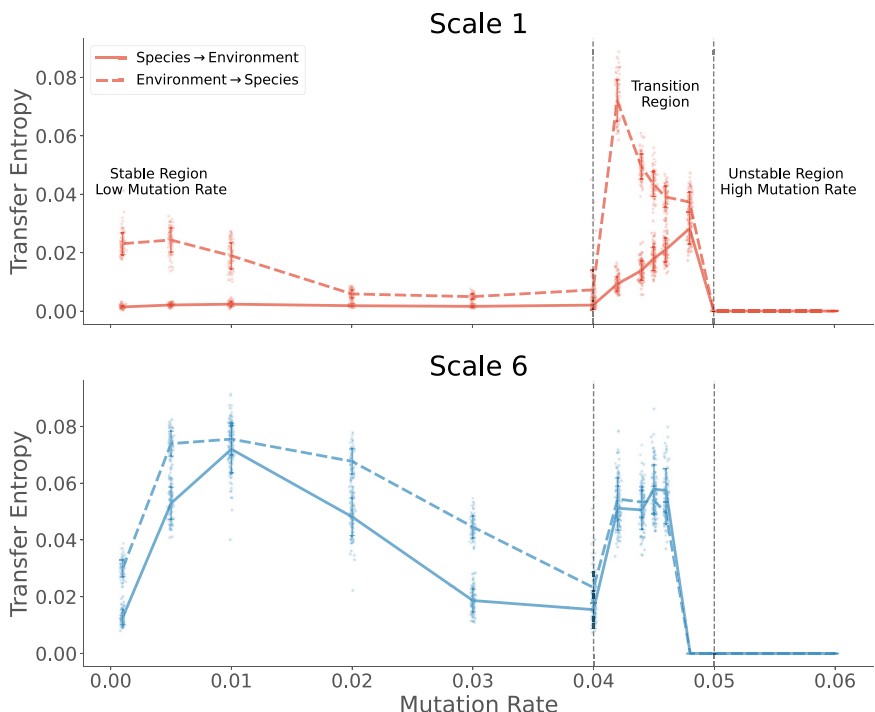

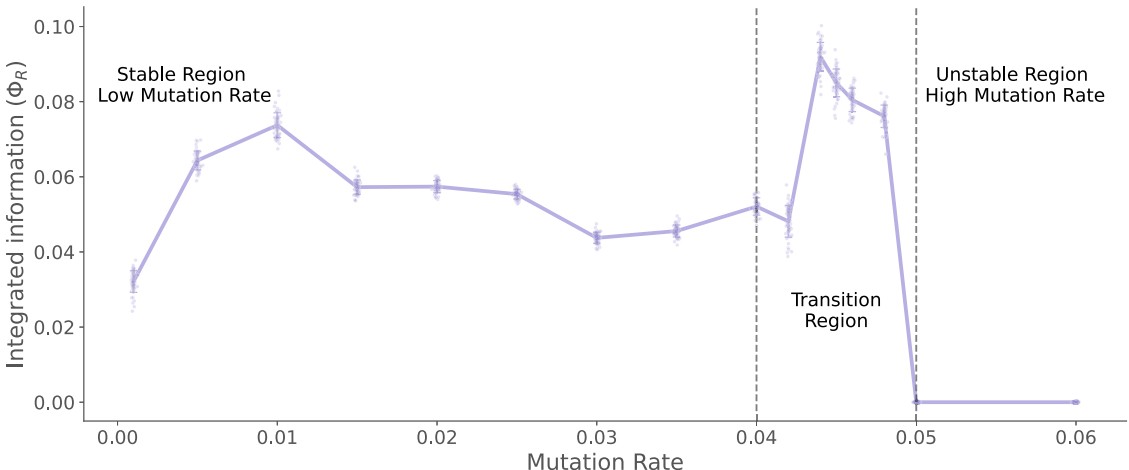

**Fig. 7 | Average Integrated Information between the species and the environment at the single species level.** The improved measure $\Phi_R$ is positive across the different values of mutation rates. This implies the species and the environment jointly co-evolve. This integration peaks in the transition region close to the error threshold. The lines show the ensemble average values of the measures calculated from 10,000 simulations of the Tangled Nature Model, using the standard parameters described in the methods. The standard deviation of measures from the mean is shown using the error bands around the lines.

evolutionary units breaks down when mutation rates are close to the error threshold, suggesting evolutionarily relevant interactions between hierarchical selection and mutation rates.

The peak in each of the information-theoretic measures for groups of 3–9 species at intermediate mutation rates is highly suggestive of hierarchically organised units of selection. In different biological contexts, this could be interpreted as the selection of ecological communities[54], holobionts[55], or viral quasispecies[44]. These interpretations would need to be tested in more focused experimental/computational studies. Given the design of the model, these results further highlight the action of selection on types of interaction beyond the individual species that are involved in a given interaction[10], since the subset of species changes in every metastable state while the interactions within the existing subset of species become more mutualistic overtime[28,56]. The results in this paper suggest that it is highly

plausible that emergent units of selection such as these could arise, and the information-theoretic measures provide a quantitative toolkit for demonstrating this. That said, more work is needed to translate these methods to real-world datasets.

Clarifying how the proposed measures of hierarchical selection are affected by mutation rates is useful to deepen our understanding of the mechanisms driving ecosystems from an ecological point of view. Increasing mutation rates are generally related, among other things, to harsh environmental conditions or adversity[57]. Mutations during times of adversity are a strategy to adapt and survive in a changing environment. Therefore, we can understand the mutation rates in the TaNa model as a proxy to varying the conditions of the abiotic environment. However, other parameters of the model also relate more directly to environmental conditions, such as $p_{kill}$, which we have not analyzed in the current study. Further work is needed to

directly study the impact of adverse environmental conditions on the co-evolution of species. Error transition—that marks the limit of mutation rate for the existence of stable populations of species—shows some interesting information processing properties. Firstly, as discussed above, the emergent higher-order organisation breaks down in this region. Simultaneously, individuality scores, as well as other information processing measures, peak for individual species. This suggests a trade-off between collective persistence through interactions (at the macro level) for increased individual persistence (at the micro level) for continued survival.

Overall, this work is a step towards enabling quantitative investigations about hierarchical selection and establishing a formal method of analysis. Although the methods discussed here focus on computational simulations, the increasing availability of ecological data and advanced Bayesian methods for inferring probability distributions[58] will make it possible to apply the proposed framework to real-world datasets. However, proper normalization techniques will need to be developed, similar to recently proposed methods[59], to enable the comparison of individuality scores across datasets. It is worth emphasising that any ecosystem has a variety of different species interacting with each other and the environment in unique ways. While the present study focused on average species-environment properties, future investigations could consider more dedicated species-level analyses. Furthermore, the framework can be deployed to other models of co-evolution to study the individuality across different timescales (from microscopic to ecosystem) of evolution and the relationship between the different scales. Finally, another interesting extension of this work could be to apply similar methods to applications of the TaNa model on social scenarios to investigate if high-order phenomena also take a central role within the dynamics of cultural[60], organisational[61], and opinion[62] changes.

## Methods

Here, we discuss the details of the model and the information-theoretic measures used for the simulation and subsequent analysis of the model. We provide a brief overview of the Tangled Nature model[25], followed by the information individuality framework[21] and other information dynamics measures presented above.

### The model

In the Tangled Nature model, species—represented by a binary vector—form the dynamical units of evolution. The binary vector is an abstract representation of the pangenome of the species. The number of agents with the same pangenome represents the population of the species. The model does not assign distinct genomes to each member; therefore, species (identified by distinct binary vectors) act as the primary unit of interaction in the model. Agents from these individual species are subject to three stochastic processes: asexual replication, mutations, and annihilation. No structured groups or hierarchies are defined a priori. The evolutionary dynamics occur in a space of pangenomes where random interactions connect different species. The probability that an agent reproduces is determined by a sum over influences from other co-existing species the agent interacts with. This simple mathematical model captures a broad range of evolutionary phenomena. Starting with a few existing species, the model dynamics evolves to a quasi-stable configuration (also known as *quasi evolutionary stable strategies* or q-ESS) where only a select group of species exist. These subsets of species with stable populations are disrupted by mutations in the system that drive them to extinction, and a new group of species emerges as a result of the reorganization. Over time, the system evolves to more and more stable configurations, where the durations of the metastable states increase[30] and the interactions between the existing subset of species become more mutualistic over time[28,56], thus avoiding these extinction events and developing resilience.

Although the model has evolved into different variations over the years, for this study, we consider the model as defined in the original paper[30]. Each species is determined using a unique binary pangenome vector of length $L$, comprising an ecosystem of a total of $M = 2^L$ possible species. Interactions

between species are encoded in an interaction matrix $J_{M \times M}$. All entries in the interaction matrix are sampled at random from a uniform distribution, $J_{i,j} \sim \mathcal{U}(-1, 1)$, thus allowing potentially symbiotic, competitive or predator-prey relationships between any pair of species. However, of all possible interactions, each interaction is permitted with the coupling probability $\Theta$. The coupling probability controls the overall connectivity of the interaction matrix. The population of a given species at a given time is represented as $n_i(t)$, and the total population of the ecosystem is represented as $N(t)$.

Starting from a random initial condition, where only a subset of species exist in the system, each timestep starts with an annihilation step in which a member of a species, selected uniformly at random, is killed with a probability $p_{kill}$. This is followed by asexual reproduction, where a member of a species, selected uniformly at random, creates an offspring with probability $p_{off}$. Each element of the pangenome of the offspring undergoes a mutation with a probability $p_{mut}$. These mutations introduce new species in the ecosystem, which then interact with the existing species. Since the population of species is updated using a random selection of an agent at each timestep, a sufficient number of updates are needed to ensure each agent is updated at least once. Therefore, the state of the system is recorded after each generation, i.e. $N(t)/p_{kill}$ timesteps, which is the average number of timesteps required to kill all currently existing species. The reproduction probability $p_{off}$ depends upon the fitness of the species at the given timestep. The fitness function, which is a weighted sum of interactions with all other species, is defined as

$$\mathcal{H}(n_i, t) = \underbrace{\frac{k}{N(t)} \sum_{j=1}^{M} J_{i,j} n_j(t)}_{\text{Inter-species interaction}} - \underbrace{\mu N(t)}_{\text{Resource Constraints}}, \quad (1)$$

where the parameter $\mu$ relates to the inverse of the carrying capacity of the ecosystem. It characterizes the impact of resource constraints driven by increasing population $N(t)$, which negatively contributes to fitness. Meanwhile, $k$ is a scaling parameter for the strength of the interactions. The interaction strength is calculated using the sum of the influences from neighbouring species $J_{i,j}$, weighted by their corresponding populations $n_j$. Thus, the fitness of a given species depends not only on how it interacts with other neighbouring species but on the population of the ecosystem as well. The fitness function is related to the reproduction probability $p_{off}$ for a given species $i$ at timestep $t$ as

$$p_{off}(n_i, t) = \frac{1}{1 + \exp^{-\mathcal{H}(n_i, t)}}. \quad (2)$$

Note that the probability of reproduction is non-linearly related to the fitness function. Although the probability of reproduction is higher for species with positive fitness, some non-zero probability of reproduction exists for negative fitness values, which enables non-performing species to reproduce and mutate towards fitter species. We visualize the fitness ($\mathcal{H}$) and reproduction probability ($p_{off}$) values in an example of interacting species (see Fig. 8)

For the purposes of our study, the fixed parameters used for the model are $L = 10$, $\Theta = 0.25$, $p_{kill} = 0.2$, $k = 33$ and $\mu = 1/143$. These parameters are chosen based on the standard parameter ranges used in previous studies[30] that have recreated intermittent co-evolutionary dynamics observed in fossil records[63]. We study the changes observed in the dynamics of the model when a key parameter $p_{mut}$ is varied. This parameter represents the selection pressure introduced by the ecosystem. Since changing temperature and weather conditions lead to more mutations[38,39], $p_{mut}$ can be considered a proxy for controlling abiotic environmental selection pressures[57]. This parameter has a significant impact on the dynamics of the system: visually, it can be observed (see Fig. 2) that the dynamics are more selective and stable with fewer transitions at very low mutation rates ($p_{mut} = 0.001$). In an intermediate range ($p_{mut} = 0.01$), more species are observed during the

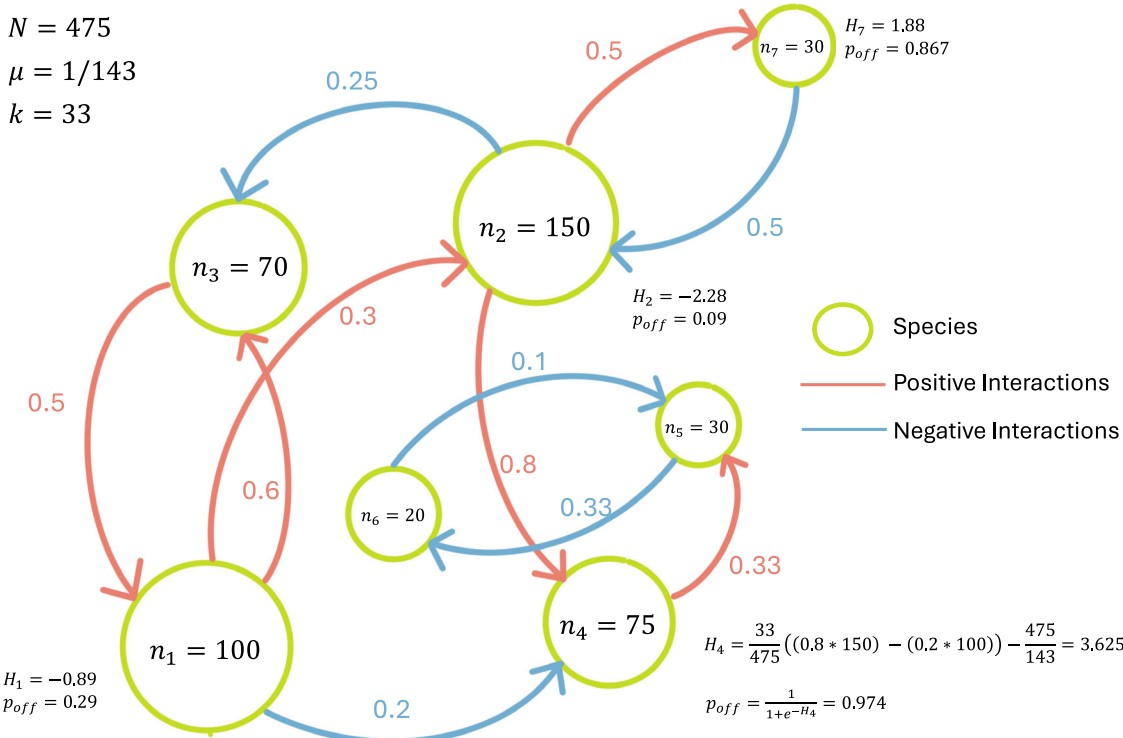

**Fig. 8 | An example network of interacting species in the Tangled Nature model.** The size of the species nodes (green) in the network represents the population of the species ($n_i$). The species influence each other using positive (red) and negative (blue) links. The strength of the influence is denoted next to the edges. We use the total population of the example ecosystem ($N = 475$) and the standard parameters ($\mu = 1/143$ and $k = 33$) to calculate fitness and reproduction probability for a few species. The fitness calculation is explicitly shown for Species 4, which gets positive and negative influences from species 2 and 1, respectively.

metastable states, along with more transitions. As the mutation rate increases in the transition region, the transition time between two metastable states increases throughout this range. Finally, for very high mutation rates ($p_{mut} \geq 0.05$), new species emerge, old species die every generation, and the metastable states are non-existent (i.e. no stable population of species emerge). Thus, the system exhibits a phase transition where the system moves from order to disorder between the range of $p_{mut} \in (0.4, 0.5)$ (see ref. 63 for a detailed discussion).

Computationally efficient Rust code was used to simulate the Tangled Nature Model. The code is available with documentation on GitHub.

### Information-theoretic measures

In this paper, we use tools from information theory to estimate the various measures presented in the Results. Primarily, we use multivariate mutual information (MI) to quantify interdependencies between time series of species populations obtained from the simulations. Typically, the numerical estimation of MI requires stationary probability distributions of the random variables. However, the Tangled Nature model exhibits non-stationary evolution, i.e. the distribution of population across species changes with generational time. Therefore, we use ensembles of simulations to estimate mutual information at each time point. The population distributions are estimated across ensembles at each generational timestep, providing statistically stable empirical distributions for the estimation of MI. This method is commonly used in neuroimaging studies where brain activity changes in time in response to a stimulus[64]. Details of this ensemble method of estimating mutual information are provided in Supplementary Information C. Below, we briefly discuss the measure of information individuality, as well as other measures used to quantify species-environment interactions.

We use the Gaussian estimators from the Java Information Dynamics Toolkit[65]. A Python implementation of these estimators on a subset of simulations generated using the Tangled Nature Model is available on GitHub.

**Information individuality**. In recent work, Krakauer and others[21] put forward an information-theoretic solution to identifying the boundary between an individual and their environment. This definition is based on principles of optimal self-prediction- i.e. if a subsystem can predict its future better than any of its parts, and any addition to the subsystem hinders its predictability, then that subsystem is deemed an information individual. This optimal self-predictability enables biological entities to control and navigate their environment[66], implying that selection for persistence[10,11] could be a putative explanation for their emergence. Once such a boundary is identified, the rest of the parts can be considered as the complement or the environment of the individual. This partition between the individual and the environment is an informational boundary[21]. Therefore, it should not be conflated with the natural environment, which includes abiotic factors not modelled in this study.

For the analyses above, let $\mathbf{S}(t)$ be a joint vector representing the population of a subset of $K$ species, $\mathbf{S}(t) = (n_1(t), n_2(t), \ldots, n_K(t))$, at a given time $t$. Where $n_i(t)$ is the population of the species $i$. If $N(t)$ is the total population of the ecosystem, the population of the corresponding environment $E(t)$ can then be written as

$$E(t) = N(t) - \sum_{i=1}^{K} n_i(t). \tag{3}$$

Then, based on the properties laid out in the original paper[21], Krakauer et al. propose three different individuality measures as follows:

$$\text{Organismal Individuality } A^* = I(\mathbf{S}(t); \mathbf{S}(t+1)),$$
$$\text{Colonial Individuality } A = I(\mathbf{S}(t); \mathbf{S}(t+1)|E(t)),$$
$$\text{Environmental determined}$$
$$\text{Individuality } nC = I(E(t); \mathbf{S}(t+1)|\mathbf{S}(t)).$$

The operator $I(X; Y)$ refers to the mutual information between variables $X$ and $Y$. The organismal individuality $A^*$ measures the total information shared between the current population vector $\mathbf{S}(t)$ to the future $\mathbf{S}(t + 1)$ of the subset of species. Essentially, it tracks how the joint probability distribution of the population of a subset of species evolves over time. The colonial individuality $A$ focuses on the extra information that is shared between the present and the future of the species beyond what can be gained from the environment $E(t)$. Finally, $nC$ measures the information shared between the future population of the subset of species and the present population of the environment beyond what is already known from the present population of the species. Here, we focus on the organismal individuality, which is closely linked to persistence selection[10,11]. This individuality measure includes both the collective predictive information of the group of species, as well as the redundancy they share with the environment (see ref. 21 for more details). Such information is useful for the species to maintain cohesion and respond to environmental fluctuations.

For our analyses, we first generate an ensemble of 10,000 simulations of the Tangled Nature model with different initial conditions for each mutation rate. We then sample a maximum of $10^5$ different subsets of $K$ species from a rank-ordered list of all possible subsets of species. The subsets are ranked in the order of population of species they contain, starting with the subset of the most populous $K$ species. The rank ordering of subsets of species ensures sampled subsets with similar population distributions and interaction structures across time, improving comparison across different subset sizes. Finally, we measure the average organismal individuality of each subset by estimating the mutual information between the current and future populations of the species (see Supplementary Information C). This process is repeated for different $K$ values ranging from $K = 1–15$. To account for the bias introduced by increasing dimensions as we calculate multivariate mutual information, we normalize using the group size (see Supplementary Information D). Thus, the normalized individuality score shown in Fig. 4 can be written as

$$\text{Individuality score} = \frac{I(\mathbf{S}(t); \mathbf{S}(t + 1))}{K}. \qquad (4)$$

**Information transfer and integration**. Finally, we briefly describe the measures shown in Section 'Interaction between species and their environment' to quantify species-environment interaction. We keep the same notation as above, using $\mathbf{S}(t)$ to denote the vector representing the population of $K$ species at time $t$, and $E(t)$ to denote the state of their environment at time $t$.

Transfer Entropy (TE) is a conditional mutual information (CMI)-based measure of Granger causality. TE quantifies information transfer from a source variable to the target as CMI between the past of the source and the future of the target conditioned on the past of the target. For instance, TE from a subset $\mathbf{S}$ of $K$ species to their environment $E$ can be written under the Markov condition as,

$$\text{TE}\,(\mathbf{S} \rightarrow E) = I(\mathbf{S}(t); E(t + 1)|E(t)) \qquad (5)$$

Measures of integrated information (generally denoted by $\Phi$) were first introduced by Tononi et al.[67] to measure integration among different regions in the brain. Since then, multiple related measures have been proposed, with some adapted to more practical scenarios[31,68] and applied to quantify interactions across a broad range of complex systems[32]. In essence, these measures quantify the extent to which the interactions between parts of a system drive the joint temporal evolution of the system as a whole; a system has high integrated information if its dynamics strongly depend on the interactions between its parts.

Here, we estimate two measures of integrated information (whole-minus-sum integrated information, $\Phi^{\text{WMS}[31]}$, and its revised version, $\Phi^{\text{R}[69]}$) between a single species and its environment jointly evolving over time. Denoting the population of species $i$ and its environment $E$ by the joint random variable $\mathbf{X} = (n_i, E)$, $\Phi^{\text{WMS}}$ is given by

$$\Phi^{\text{WMS}} = I(\mathbf{X}(t); \mathbf{X}(t + 1)) - \sum_{i=1}^{2} I(X_i(t); X_i(t + 1)), \qquad (6)$$

where $X_i$ denotes the $i^{\text{th}}$ element of $\mathbf{X}$.

Despite its intuitive formulation, $\Phi^{\text{WMS}}$ has one crucial disadvantage: it can become negative in systems where the parts are highly correlated[69,70]. To address this problem, Mediano et al. proposed a revised measure of integrated information, $\Phi^{\text{R}}$, based on the mathematical framework of integrated information decomposition ($\Phi$ID)[69]. This revised measure simply adds a new term to $\Phi^{\text{WMS}}$, correcting for the correlation, or redundancy[71], between the parts of the system:

$$\Phi^{\text{R}} = \Phi^{\text{WMS}} + \min_{i,j} I(X_i(t); X_j(t + 1)). \qquad (7)$$

In the main text, we report results using $\Phi^{\text{R}}$, due to its better interpretability. For completeness, we provide a comparison between the two measures of integrated information in Supplementary Information Fig. F.1.

## Reporting summary

Further information on research design is available in the Nature Portfolio Reporting Summary linked to this article.

## Data availability

The simulations analyzed in this study were generated using the Tangled Nature Model. A subset of simulated data is made available on GitHub. Data used to generate the plots of the manuscript are provided in Supplementary Data.

## Code availability

A computationally efficient RUST code is made available publicly on GitHub. A Python implementation of the information-theoretic analysis using JIDT, along with the associated code and a subset of simulated data, is also made available on GitHub.

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

## Acknowledgements

H.R. and H.J.J. are supported by the Statistical Physics of Cognition project funded by the EPSRC (Grant No. EP/W024020/1). H.R. is also supported by the Ecosystem Leadership Award under the EPSRC Grant EP/X03870X/1; the Economic and Social Research Council under Grant ES/T005319/2; & The Alan Turing Institute. C.S. was supported by PRIMUS grant 22/HUM/020 from Charles University. F.R. was supported by the Fellowship Programme of the Institute of Cultural and Creative Industries of the University of Kent and the DIEP visitors programme at the University of Amsterdam. P.A.M. acknowledges support from grants Fondecyt 1200925, Proyecto Exploración 13220168 and through Centro de Modelamiento Matemático (CMM), Grant FB210005, BASAL funds for Centers of Excellence from ANID-Chile. HJJ also thanks EPSRC for supporting this work as part of the Quantifying Agency in time evolving complex systems project (Grant no. EP/W007142/1).

## Author contributions

All authors contributed to the conceptualization of the research programme, along with the review and editing of the manuscript. H.R. conducted the investigation, formal analysis, code development, data visualisation and prepared the first draft of the manuscript. C.S. developed the Rust code used for simulations in the project. P.A.M.M., F.R. and H.J.J. contributed to the methodological development for the study. P.M. and E.V. contributed to the theoretical formalisation of the methodology. H.J.J. and F.R. supervised the study.

## Competing interests

The authors declare no competing interests
