## [Transparent Peer Review file · Communications Biology]

Information dynamics and the emergence of high-order individuality in ecosystems

Corresponding Author: Dr Hardik Rajpal

Version 0:

Reviewer comments:

Reviewer #1

(Remarks to the Author)

Review of 'Information dynamics and the emergence of high-order individuality in ecosystems'

Summary:

This is an interesting theoretical modelling paper. The paper uses a computational re-implementation of a model of interacting genomes called the 'tangled nature model' (Christensen et al, 2002). The authors explain in the methods that the tangled nature model includes a fitness function in which the chance of reproduction by genomes is affected by the interactions between the genomes that coexist at a given time. The 'genomes' are strings of zeros or ones of a set length. The ecological interactions are randomly allocated to distinct genomes and can be positive or negative. The study then calculates from the simulations under this 'tangled nature' model a range of measures from information theory that aim to quantify 'interdependencies between time series of species populations obtained from the simulations'.

Throughout the manuscript the term species appears to refer to a distinct type of genome, therefore I have suggested that the use of species be replaced by another term to avoid confusion with the standard biological species definition and to add clarity with regard to level of selection debates.

One of the key informatic measures used by the study is an implementation of a measure proposed as an 'information-theoretic solution to identifying the boundary between an individual and its environment' by Krakauer et al 2020. These measures could do with more explanation within the current manuscript e.g. definition of the notation used below equation 3.

The paper also considers a measure of transfer entropy, which would also benefit from further plain language explanation.

The authors state 'In essence, these 249 measures quantify the extent to which the interactions between parts of a system drive the joint temporal evolution of the system as a whole— a system has high integrated information if its dynamics strongly depend on the interactions between its parts.'

I found it hard to get an overview of what the study is measuring, but I think it may be something like as follows. The study models the reproduction and mutation of model genomes (which are strings of zeros and ones). The genomes have randomly set, pairwise interactions which can be positive or negative if they exist. There is an overall population size and some resource limitation term to the fitness function for each genome type. One can then consider overall population fitness based on the numbers of positive versus negative (or non-existent) interactions, or the fitness of subsets of the overall population of genomes, over time. I think the 'environment' may designate the rest of the total population other than a considered subset, though this was not clear to me (e.g. as this is not what the use of the term environment would have initially led me to expect: I would have expected environment to refer to physical environment). I think the transfer entropy measure may measure something like the extent to which current population sizes (which result from the effect of the fitness function on genome reproduction), for a subset of genome types, predict future population sizes for the rest of the population other than the subset (with this remainder constituting the "environment").

The results state 'Special attention is paid to the dynamics observed in the vicinity of the 'error threshold,' which is the $2/16$ limit on the mutation rate for species beyond which any biological information needed for continual survival is destroyed

in 66 subsequent generation^{31, 32.}

So, I interpret that the study is looking at the information on future population sizes inherent in current population sizes (resulting from genome fitnesses based on their interactions) when the genome mutation rate varies (possibly rates?). And, especially near the point where the mutation rate gets so high that, I guess, future genomes get decoupled from genome fitness (e.g. that of the parental genomes), as there is so much mutation that genomes are effectively randomly generated at each time step.

My biggest question about the results and conclusions is to what extent the variation in information measures used is a necessary consequence of the persistence of groups of interacting 'species' (which I believe is a previously known result as it has citations 28 and 30 in the text in the first section of results). What alternatives are mathematically possible? If you have fixed versus variable total population size, is it possible to have groups of genomes which are set to have interactions and persist for some period of time, that do not behave e.g. as 'individuals' using the information measures shown? At the moment, I don't clearly see a distinction made between null and alternative hypotheses for the information measures applied, and I think this should be added, particularly as I am wondering whether information individuals are the null (or perhaps only possible?) hypothesis given the known properties of the tangled nature model.

Because the paper builds on previous theoretical work, I found it quite hard to tell in the abstract and results summary which specific parts were new to the present study. This should be more clearly stated. I believe, from reading the paper, that the novel results are those that relate to the effects of mutation rate specifically and the application of information measures to the studied ecological model. However, mutation rates don't appear to be clearly highlighted in the abstract, where mutation rates are not explicitly mentioned. I suggest more clearly demarking in the abstract and conclusion the novel results.

The authors state in the main text 'We study the 210 changes observed in the dynamics of the model when a key parameter p_{mut} is varied. This parameter represents the selection 211 pressure introduced by the ecosystem. Since changing environmental conditions lead to more mutations, p_{mut} can be considered 212 as a proxy for controlling environmental selection pressures^{44.}

p_{mut} is the genome mutation rate. This could perhaps do with more introduction and discussion e.g. with regard to the general biological plausibility of variation in mutation rate under changes in environmental conditions. E.g. It would be helpful to briefly review the literature on when mutation rate may vary, or conversely, be invariant, over specific timescales, in different parts of real genomes and biological groups. For example, below is a link to an article with similar suggestions in humans which was presented as controversial:

<https://www.eurekalert.org/news-releases/941828>

The recent paper below also presents related suggestions as counter to standard views:

[https://physoc.onlinelibrary.wiley.com/doi/full/10.1113/JP284411?](https://physoc.onlinelibrary.wiley.com/doi/full/10.1113/JP284411?casa_token=WT7LBnHneL8AAAAA%3Alby7XNJsBHp3cyiAetUExVajlgA3_HHlf6AXfzKEoChMmUybGF1BOQ-h6MFb1yYnqlUliqSqoUJHA)

[casa_token=WT7LBnHneL8AAAAA%3Alby7XNJsBHp3cyiAetUExVajlgA3_HHlf6AXfzKEoChMmUybGF1BOQ-h6MFb1yYnqlUliqSqoUJHA](https://physoc.onlinelibrary.wiley.com/doi/full/10.1113/JP284411?casa_token=WT7LBnHneL8AAAAA%3Alby7XNJsBHp3cyiAetUExVajlgA3_HHlf6AXfzKEoChMmUybGF1BOQ-h6MFb1yYnqlUliqSqoUJHA)

I wonder how easy it is to distinguish between variation in mutation rate and variation in survival/fixation rate in the real world for mutations with fitness effects.

Suggestions on presentation:

This is a paper which has been submitted to a general biology journal but is likely to be quite difficult to understand for many biologists, as we should not assume the general biological reader will have any familiarity with information theory etc. I therefore suggest a number of improvements to the accessibility of the paper. I have suggested a number of specific points below where more explanation of the concepts used should be given in the abstract and introduction. I think all the technical terms in the paper from "information theory" down should be briefly explained at, or close to, first use. I also believe that terms like 'species', 'environment' and 'mass extinction' are used in ways that differ from standard biological uses of these words, and I suggest changing the terminology to make this less confusing where possible (see details above and below), and clarifying with an explicit definition anywhere else. I also suggest working to make the key mathematical modelling concepts easier to follow. I would suggest listing the notation after each equation, as at the moment one has to pick back through the preceding text to work out what the variables are. I also suggest some presentation methods could be used such as coloured maths, in which conceptual parts of the key equations are coloured with a key to plain English explanations of what they mean. I would also suggest adding simple worked examples to the methods. For example, some toy example genomes could be fed through the equations and the output shown e.g. an example of the calculation of fitness with equation 1 showing explicit examples of the types of inputs and variable values used. This (worked examples) would also be extremely helpful for the information measures. I also suggest that plots of the methods would be helpful e.g. it is stated 'Note that the probability of reproduction is non-linearly related to the fitness function.' At equation 2. This would be easier to follow if method plots were provided showing examples of inputs and outputs to equation 2. The figure captions would also all benefit from more information, including stating the meaning of all variables and replacing initialisms with full text.

Specific comments:

Abstract 'transition region close to the error-threshold' Please clarify at this point the usage of transition e.g. state from what to what. Indeed this whole sentence could do with much more explanation. There is a good, brief explanation of the error threshold in the results, but this should be explained earlier.

'Our results show that evolutionary dynamics can lead to clusters of species that act as a self-perpetuating group, that acquire information-theoretic agency. However, this higher order organization breaks down during the transition region close to the error-threshold, where single species gain higher information-processing abilities.'

Please can you clarify whether the first result mentioned above is novel to this paper i.e. not in earlier 'tangled nature' analysis. I.e. it would be helpful here to more clearly demark all of the results which are novel to this study specifically.

'Overall, our findings provide quantitative evidence supporting the relevance of high-order structures in evolutionary ecology, which can emerge even from relatively simple processes of adaptation and selection.'

Relevance to what?

Line 1: grammar: dynamics are

<https://english.stackexchange.com/questions/488664/the-dynamics-is-or-the-dynamics-are>

See also line 193 etc.

Species produce offspring or something rather than 'species reproduces offsprings'.

There are quite a few apparent plural/singular mismatches throughout, perhaps Grammarly might pick these up if you run the text through.

Line 4: Please complete this sentence so it is comprehensible without looking at the references e.g. such as in experimental evolution.

Second paragraph. I would suggest that the standard current emphasis is still on the individual as the fundamental level of selection. E.g. Even under a 'selfish gene' view, genes still have to act, to some extent cooperatively, within their vehicle, the individual (e.g. as discussed in Dawkin's 'the selfish gene').

Line 11. Indeed, though these debates don't just cover above-individual levels. I would suggest debates extend across all possible levels of selection including genes, cells etc. within individuals. E.g. See work of Okasha (already cited) on evolution of cancer etc. So, for accuracy in representing the literature I suggest slightly modifying this sentence to note this.

Line 12. Please briefly introduce/further explain persistence theory.

Line 15. Emphasis from singer to song. It is very surprising to me that you cite the species as the singer here when the standard view would be that selection acts on the individual and the species is a higher order group. What is the justification for this?

Line 26. Citation or justification needed.

Line 27. Do you mean combining or using, if combining, what with what?

Line 33. The 'tangled nature' model is stated to be well-studied. I had, personally, not heard of this so I looked at the cited paper. This has 9 citations so is not likely to be generally known (I suggest removing 'well-known' from the paper). Please therefore introduce what this model is in the introduction and explain why it is a suitable study model for the research question.

The first paragraph of the methods, at least (and possibly the 2nd and 3rd), appears to me better placed in the introduction as it is an introduction of previous work rather than a description of methods new to the current paper. Putting this sort of explanation in the introduction is likely to make the paper more understandable if read from start to finish. Given the comms bio format, with methods at the end it might also be valuable to place a clearer explanation of the key information measures in the introduction e.g. in a box.

Please explain in the introduction what you mean by selection effectively acting. Why effectively not just acting?

Line 30 'The basic hypothesis behind these approaches is that if a group of 30 individuals can enhance the prediction of their joint future, they can better adapt and thus survive.' Is this your hypothesis or that of Krakauer et al. 2020, cited in the previous sentence? I read that paper some time ago and what I got from it was that it was about how information theory can be used to identify the level, or levels, of individuality. It would be helpful if you could be more explicit about how far that paper went towards this hypothesis e.g. by quoting what you think is the most relevant text.

Line 35. Every endeavor should be made to make this paper accessible to a biological reader. Therefore, I suggest here briefly introducing in plain language what all the terms/measures mentioned are under an assumption that many potential readers will not be familiar with information theory, integrated information, information dynamics, the measure of individuality mentioned, error in the informatic sense etc.

Line 37/ 'Overall, our results provide quantitative evidence 37 suggesting that groups of species exhibit higher-order individuality for biologically plausible mutation rates.'

Assuming the tangled nature model (which hasn't at this point been described) is a theoretical model, this should be rephrased to make clear that what is provided is a theoretical analysis (i.e. modelling on a model) in support of the conclusions, not evidence in the sense of experimental or observational biological data. Theory is fine, but it's not very clear to me reading this whether data has featured or not in the study, so I suggest clarifying.

Line 38 plural individuals

Line 37-43. There is a results summary here which looks like it would be better as a conclusion. Here, it's not very easy to follow. Especially, if you want to leave a results summary at the beginning, I suggest first expanding the introduction so the reader has more preparatory information at this point.

The captions for the figures particularly, 4 onwards are too brief. Imagine a general biological reader reading in order of abstract then figures before the main text. I think the captions are incomprehensible if used to initially survey the paper and, ideally, they should be understandable without referring to the text. E.g. Fig 4. 'Mean AIS for the two scales of organization – Individual (Scale 1) and Higher Order (Scale 6) – varying with mutation rates.' Please briefly explain what all these things are without abbreviations and set them in some context. Please look at all the figure captions in this regard. I suggest starting with a title in plain English that gives an overview of the point of the figure, then explanations of all the technical terms and information on where the data come from for each figure e.g. including info such as "theoretical model of x based on y with parameters z showing a" etc.

Typo in caption of Fig. 6 species.

Line 188 Please explain what a binary genome is.

Line 189 Please give more detail on the level/unit of individuality here. Species are described as the units but replicating genomes are also mentioned: are the species not higher-level groups of genomes i.e. individuals? It seems to me that the genome i.e. individual is the unit. If not, why not?

'No groups or hierarchies 190 are defined a priori.' Then how are genomes matched to species?

I am wondering here if the terms species is being used for what is more like a biological individual or perhaps a clone: a copy of a genome. Species may perhaps be used more in the chemical sense of an identical molecule than in the standard biological sense.

'Each species is defined using a unique binary genome of length L, comprising an ecosystem of a total of $M = 2^L$ possible species.' This suggests that each 'species' used in the current manuscript is one of the combinatorially possible, distinct genomes, where a genome is a string of zeros and/or ones of a set length L.

If this is the case, I suggest changing the term species to individual, or some other more suitable term, or simply using something like "distinct genome type" (without adding some extra name) throughout the manuscript. In any case, please clarify the definition of each name used and explain how this relates to standard definitions e.g. of biological species, individual etc., particularly considering asexual/sexual reproduction (as asexual reproduction is modelled in the paper). First, this is currently confusing since a biological species, in standard use, is a group of sexually reproductively connected individuals, which are not all expected to have identical genomes. Second, the paper relates to the unit of selection and if the 'species' in the paper are potentially most like biological clones i.e. copies of the same individual, everything stated as at the level of species in the paper would, I would have thought, be potentially better biologically interpreted as at the level of the individual.

'populations of existent species' This is confusing because it would be more common to have biological populations within species. If you are referring to groups of species please use a more appropriate terms e.g. clade if they are monophyletic groups of species, or a more general term such as set or group, in which case please explain how membership is defined.

Line 194. Singular strategy?

Line 196 what does q-ESS mean? I would suggest avoiding abbreviations as much as possible, throughout, as the paper is likely to be very hard for an average biological reader to understand even without them.

'Starting from a random set of populations of existent species, each timestep starts with an annihilation step where a member of a species, selected uniformly at random, is killed with a probability pkill. This is followed by asexual reproduction, where a member of a species, selected uniformly at random, creates an offspring with probability poff. Each element of the genome of the offspring undergoes a mutation with a probability pmut. These mutations introduce new species in the ecosystem which then compete with the existing species. The state of the system is recorded after each generation, $N(t)/pkill$ timesteps, which is the average number of timesteps required to kill all currently existing species. The reproduction probability poff depends upon the fitness of the species at the given timestep.'

'a member of a species, selected uniformly at random, is killed with a probability pkill' suggests that survival is random. While 'The reproduction probability poff depends upon the fitness of the species at the given timestep'. If I follow this, it suggests a model of evolution in which reproduction is subject to a fitness function but survival is not. This strikes me as biologically surprising. One could perhaps collapse survival and reproduction into reproduction probability, but in that case why have pkill – it seems to me this may introduce a genetic drift component, which is the population is small may have a large effect. What is all of this supposed to be modelling, how and why?

Line 203 'However, of all possible interactions, the 203 number of permitted interactions is controlled by coupling probability Θ .' Please explain this e.g. how are permitted/disallowed interactions set.

Line 225. Please explain what mutual information is in this context.
Please also explain stationary and non-stationary evolution in your context.

Lines 231-235 'In recent work, Krakauer and others¹⁸ put forward an information-theoretic solution to identifying the boundary between an individual and its environment. This definition is based in principles of optimal self-prediction — i.e. if a subsystem can predict its future better than any of its parts, and any addition to the subsystem hinders its predictability, then that subsystem is deemed an information individual. This optimal self-predictability enables biological entities to control and navigate their environment⁴⁸, implying that selection for persistence^{10, 11} could be a putative explanation for their emergence.'

I think this would also be better placed in the introduction to increase the understandability of the text and as it seems more appropriately placed there. It would be helpful to expand also on the explanation of predicting the future and the justification for the definition of corresponding individuality e.g. is this theoretical or actual, where is this information potentially or actually held and what does this mean if only theoretical. For example, most individual people might think of themselves as paradigmatically an "individual" but much physiological, nervous and cognitive function is apparently subconscious and/or distributed. Therefore, it doesn't seem to me that the human brain holds an integrated model of itself or the wider body anywhere, nor is it able to predict its future states in a full sense. Would this mean that under the definition above, people are not individuals because they cannot predict the future states of their brain, body or their sub-components? Can you give a/some examples of how this prediction you hypothesise might actually occur in biological systems. A justification of these underpinnings in the introduction, if possible, might considerably strengthen the paper as well as making it more accessible.

'Carrier capacity' Do you mean "carrying capacity"? Please also give a bit more detail on this variable μ to make clear what the last term in equation 1 is like. E.g. If this was carrying capacity (i.e. maximum population size in standard use) and carrying capacity was higher this would then be multiplied with population size and subtracted to make fitness lower, which is counterintuitive as, I would have thought, overall population fitness should be higher if resource constraints are reduced i.e. carrying capacity is higher. So, more detail would be helpful here to help the reader figure out how the model does work.

'Thus, the fitness of a given species depends not only on how it interacts with other neighbouring species but with the rest of the environment as well.'

Please explain thus i.e. what specifically does this follow from?

Please consider/explain the use of the term 'environment' for equation 3. This looks like it is a population size. How does this have anything to do with "environment" in the usual biological sense?

Please define the notation for the individuality measures at their use immediately below equation 3.

Line 47 distributions plural

Line 67 expected why? Because it is your hypothesis or because previous work has shown this result?

Line 70 'quasi stable q-ESS and the transition regime' need to be explained.

Figure 3. Please change the colour scheme so the line colours transition in a perceptible spectrum through the key values.

There are a few grammar errors in the first paragraph of the results.

Line 160 in the discussion. This seems speculative. Can you investigate this more directly, or if not rephrase as a hypothesis for future work.

I suggest removing the word 'important' from the conclusion (and elsewhere if used): this should be for the reader to decide, in context, from a clear statement of the aspects of novelty of the analysis and results.

Reviewer #2

(Remarks to the Author)

Note: I am reviewing from the perspective of general mathematical modeling.

Summary:

The authors use an existing explicit mechanistic model that simulates evolutionary systems which the authors believe has emergent "higher-order selection". Their central claim is that they can retrieve the amount of selection induced by the parameters chosen at various units of organization using information theoretic tools. While it is not clear to me if solving the debate the authors are aiming at will be revolutionary for the field in some way, the topic and paper are definitely of at least theoretical interest, and the paper is interesting to read.

Overall:

Generally, the paper is good, but the narrative around the paper is repeatedly unclear due to inconsistent or poorly explained language. Given the interdisciplinary nature of the paper, the definitions should be tight and the language clear and consistent without hoping the audience will be up-to-date on all of the papers cited. Almost all of my critiques are about the use of language, rather than any clearly biologically or mathematically flawed arguments. With that having been said, I cannot comment on the appropriateness and validity of any statistical analyses until Appendix C is clear and explicit, as all analyses in the paper hinge on it (see lines 226 - 227). Due to its influential nature, I am unsure why it is left as an appendix. As to whether I am convinced, I am not, but I think that may be due near exclusively to the miscommunications I've noted below.

Comments:

1. I always find the linguistic usage of processes, like evolution or selection, and actions to be controversial with quite a few of my biologically minded colleagues. I don't mind, e.g., the first sentence of the abstract, but I think some might be annoyed at the idea that selective pressure "acts". Maybe "At what level (or perhaps scale?) does selective pressure apply?" The rest of the abstract agrees with my understanding of my colleagues' usage, but some of the text wobbles back over. (E.g., "selective unit" is fine, but I don't think they'd be pleased by saying individuals or genes are the "drivers of evolutionary change".)
2. Lines 24 - 25 can be misread as indicating the authors are using a model that explicitly simulates higher-order selection, rather than that these properties are emergent.
3. Line 38: it isn't quite clear from this sentence what it means for "single species [to] act as information individual". More generally, I find this paragraph a bit hard to follow. I'm not sure if the same concepts are being referred to or if they are new each time because the language is not quite consistent. E.g., is "this higher-order phenomenon" "this cooperation", the "information individual", the "higher-order individuality", or the "error-transition threshold"? And in line 40, does "or the model" mean the authors conflate the analysis with the model, or are they not sure which does not assume a preferred level of selection, or do both not assume a preferred level of selection?
4. There are occasional distracting grammar configurations and mistakes, e.g., line 50: "Based on this fitness, species reproduces offsprings..." A general readthrough by the authors would probably pick up most of these.
5. Is it intended that line 56's "generation 1" links to Figure 1?
6. It would be useful to note whether there is some relationship to a temporal scale compared to the generational scale, as well as whether there is time resolution of the dynamics within each generation (by which I mean, are the populations dynamically adjusting through time or is the next steady-state calculated and the system assumed to reach it immediately). This is in the methods, but is not quite as explicit there as I would expect. It's easy to miss that the 10^5 generations are not 10^5 timesteps, for instance. It's also not immediately clear that "generations" is the right timescale, since it invokes the size of the population, which is affected by the mutation rate (Figure 2).
7. Do the dynamics described in lines 56 - 61 constitute a phase transition?
8. The legend in Figure 2c could be improved, as there are 3 vertical lines that could be interpreted at a glance as $p_{mut} = 0.042$.
9. While visually the transition regions are obvious, how do the authors define it precisely? How long/stable do the states need to be to be q-ESS?
10. It isn't obvious to me why the error threshold as defined should be interpreted as if-and-only-if the results shown in Figure 2. How do I know, for instance, that this isn't an artefact of the resolution or timescale the data is displayed at? The error threshold implies something akin to white noise, but it isn't clear to me if that is the sort of noise I am seeing at mutation rate 0.05, or if there might be some memory in the form of coloured noise. Could the authors explain?
11. Maybe I have misread, but it appears that the authors refer to both $0.04 \leq p_{mut} \leq 0.05$ and the actual dynamics of the populations both as the transition regime, see caption of Figure 1 versus lines 72 -- 73 or Figure 4.
12. Figure 3: I think the main message is the mutation rates and scales, rather than the individuality scores. The colour scale doesn't help with this (e.g., two greens are visually adjacent but are 0.001 and 0.03). Maybe plot the scale at which the maximum is achieved against the mutation rate and move this to a supplement? The accompanying text is hard to follow as a result.
13. When and how is scale 6 identified as the optimal scale of organismal individuality?
14. Is normalisation based on the size of the group mathematically obvious? See 36 below for more queries about this at the end of the paper.
15. Are the heights of the individual curves in Figure 3 of any interest?

16. Is the valley in the transition region of Figure 4, scale 1 interesting?

17. I'm not sure that "environment" or "ecosystem" are clearly defined in the text in reference to the model. I get the sense that the environment of a focal species is everything not of that species in this paper, but some ecologists exclude everything biotic from the environment. While I don't go quite that far, I do find the description uncomfortable here. Is ecosystem then the environment along with the focal species? This is defined in the methods, but without reference here.

18. Line 114 "the information that a source variable about the future state of a target variable"? Information provided by? And presumably this means that previously the authors meant information at time $t-1$ predicting time $t+1$, whereas here they are interested in time t predicting time $t+1$? Additionally, which timescale is this t acting at?

19. When the authors discuss information transfer, I find myself wondering what is "information" here. The language used suggests that it is something exchanged between species and environment, but the description suggests instead that it would be something an observer would make use of. Returning to earlier parts of the paper didn't make the subject clearer. This is critical for getting the right message across to the audience.

20. Lines 124 - 125: does equivalent mean "mathematically equivalent", or "essentially the same as"? If the former, why bother with two names for the same thing for an audience who is likely to be unfamiliar with the subject?

21. Lines 124 - 129 provokes many questions. How much environment versus group? If the environment is predicting things about the group, should the group subsume more of the environment, since these are mathematically, if not user, determined rather than biologically determined? Is scale 6 the correct scale to evaluate this at? How do the scales change with the environmental information provided? Does it mean anything that "higher-order organisation still possesses environment determined individuality in the error-transition region", since the authors have effectively argued that higher-order organisation is non-existent? Is there a necessity to control for noise here?

22. Figure 5: is the last panel necessary? I thought the text said that the curves in the last panel are equivalent to two of the curves above.

23. Line 130 appears to lack all useful definitions. Lines 130 - 134 strike me as unfinished and unexplained for the reader.

24. The authors may want to reconsider the placement of the figures, as Figure 6 is mentioned on page 5 but is placed at the bottom of page 7.

25. The last paragraph before Section 3 is not clear to me and again provokes many questions. Where does, for instance, resource sharing occur in the model? What does robustness mean in this context? Are there mutations that happen at a level other than that of individual species? What do the authors mean by criticality? What is processing information in this context? What do adaptability and dynamic range mean in this context? What does "in order to maintain persistence" mean? Is the system actively resisting in some way the transition regime?

26. As written, this paper seems to really struggle with having the methods section last. Some reorganization of material might help here, but it might be satisfactory for the authors to better direct readers to the methods at opportune moments. As it is, the methods is referenced once, as if the reader should read the entire methods at that point. If that's the case, why put them last? Possibly similarly for the supplement, as I didn't see it mentioned in the text.

27. It might be beneficial for the authors to discuss, if not test the robustness of, other parameters. Is the transition governed solely by mutation rate, or is it modulated by the number of species or any of the other standard parameters? (And why are these parameters standard? Are they selected for biological plausibility?)

28. "Therefore, we can understand the mutation rates in the TaNa model as a proxy to varying environmental selection pressure." And p_{mut} is also "the selection pressure introduced by the ecosystem" (lines 210 - 211). Given the lack of clarity regarding what is the environment/ecosystem in the system, I'd hesitate to have either sentence in the text. If the environment includes species other than a focal species, then shouldn't environmental selection pressure already implicitly be included in the model via the interactions between species?

29. I think in order to "sell" the message, the authors need to present examples to help frame the idea. Jumping away from TaNa for a moment, if we had evolution in a classic Lotka-Volterra, I'd hesitate to call the predator and prey a "group of cooperating species acting as an effective unit within an evolutionary process" (to borrow words from line 154). The authors could circumvent that critique with some clarity at the beginning regarding the model and what a group means. It also might help to present (examples of) what groups emerge in the model.

30. Line 218: "Thus, varying the mutation rate provides two interesting transition points..." I don't see the first "interesting transition point". Instead, the first regime seems fairly continuous before not-instantaneously switching to a different regime.

31. Is there a reference justifying the contents of Appendix C? I can't tell if I should be very worried or not at all worried from it. Either way, it needs another go from the authors. I'd especially appreciate a reference regarding the timescales for the sampling if possible, i.e., that the sampling time scale shouldn't be related to the mutation rate. (That'd actually be a nice thing to examine if possible: what happens to the various metrics if you compare before and after some large number of mutations.)

32. I'm not sure why, but a few bits are not line numbered correctly.

33. Hamming distance is named after Richard Hamming.

34. The authors seem to argue that their biological model contains units predicting their future (lines 232 - 235). Could the authors elaborate on how that is the case in the Tangled Nature model?

35. The authors should explain their notation in section 4.2.1.

36. From line 241 to eq 4, could the authors be more specific? How is the random sample taking place? How many different subsets? Which bias is introduced by increasing dimensions?

Version 1:

Reviewer comments:

Reviewer #1

(Remarks to the Author)

Based on the rebuttal letter, the authors have made numerous edits to the text addressing the comments raised, especially in order to improve the accessibility of the paper. I therefore recommend acceptance.

Reviewer #2

(Remarks to the Author)

Thank you to the authors for their work on this revision. I appreciate in particular some of the rearrangement of material the authors have done, which seemed to do a better job of introducing the results, and the text is generally more accessible I believe.

Below, I've tried to arrange my comments to correspond to the ongoing conversations as well as the text order where possible, but I appreciate that I have added additional comments as well.

Overall Concerns:

Unfortunately, I do not feel my queries and criticisms of the analysis, especially regarding Appendix C, have been fully dealt with at this time and I cannot yet recommend publication. I must ask the authors to make another attempt at revising the text. Details are below.

I also think I agree with R1 that the understanding of "mutation" and "species" in this model might have significant impacts on interpretation. Reproduction (clonally) is clear enough, but if mutation were replaced with speciation or diversification, I can't help but think that the manuscript might receive a different reception. This has repercussions for my understanding of the text throughout I believe and especially near the end of the text, as well as whether R1 and I are understanding the same things.

Additionally, I realised in hindsight that I have not seen the code associated with *the analyses* conducted here, although the authors do point to a Github containing Rust code *for the simulations*. I would like to see that in the next round of revisions, as many of my methodological questions could have been answered from looking there. My apologies if I missed the analyses in the Rust code.

Presentation Concerns:

One emergent and overarching question that I had on this read-through is whether the authors are essentially measuring the probability that subsets of a system will change state between one generation and the next. If so, this might be a more accessible framing.

(RC 4) I note that Grammarly does not appear to have been a panacea for some errors, e.g. Appendix C has "tangled nature" and its second sentence reads as incomplete (it should probably be combined with the first sentence). A similar issue was near immediately apparent when I went back to the main text (lines 11 - 12). I surmise the manuscript would benefit from another careful, manual grammar check as there were plenty of similar errors still.

I also note that, while adding color to the mathematics is an interesting idea and sometimes done in the classroom, I'm not keen on it for a manuscript where it might present an accessibility issue (e.g. if the reader tries to read on an alternative background color). This is a minor comment, but worth keeping in mind.

Comments:

(RC 34) Abstract: "where single species gain higher information-processing abilities". If I've understood correctly, species do not have information processing abilities in this model. Instead, an observer of the model receives more or less information about the species as groups in this model. This is supported by lines 35 - 36 and seems a sensible interpretation of Figure 1 and AR 34. If I am correct, the abstract should be changed.

(RC2 1) Related to this; Lines 45 - 46: While this is definitely true from my memory of the Tangled Nature model, I do wonder if it might be more persuasive if the authors mention beforehand that species are single genotype/phenotype, so any mutation results in a technically different species. This ties in to R1's RC 36. It might also be worth commenting how large a difference "a mutation" makes. As I recall, there is not a sense of a clade in the Tangled Nature model. (Indeed, it might be more appropriate to think less of mutation and more of a random but biased replacement of an individual of a species with a member from the pool of species, as the traits seem to be able to completely vary between species in the authors' version of the model, lines 249 and 240 - 242.)

(RC2 2) And, on the subject of mutations, instead of "stable species", it may be more useful to say "stable populations of species" throughout. The former might be interpreted as being about the evolution of a species, which seems confusing in this context, while the latter seems more clearly about whether a population or set of populations persists.

(RC2 3, but also 29) The description of Figure 1 in the text made me expect a model to be solved (lines 52 - 53). The authors may want to instead say the figure is schematic in nature and be more specific about "these measures" since, e.g., it is not immediately obvious if "individuality" is one "of these measures" in Figure 1.

(RC2 4) Also, middle and bottom panels do not make sense for Figure 1 as in the text. I'm also mildly confused as to why the influence of one species over the other is called "statistical". Predictability would seemingly make more sense, although I suspect this is a disciplinary difference and the authors may need to make decisions regarding the target audience.

(RC2 5) Lines 73 - 75: I wonder if another comparison might be more useful here, as the subset of all feasible combinations is likely to be quite smaller than the space of all combinations. I much prefer thinking of it as a continuous time Markov chain defined over the assembly graph myself, but that might be another discipline too many for the audience. (I suggest this because it is now even less clear to me what a "collective" random walk means in this context!)

(RC 6, 31) Regarding Appendix C, as this is an ecology facing article in a biology journal, it might be wise to expand on the technique beyond that it is standard in neuroscience. The authors might also consider that they've directed readers to a 33 page reference in an additional different discipline, and this reference does not use the same terminology as in the manuscript (ensemble and stable were the obvious words to me to look for in the reference, while copula is not mentioned in the manuscript). Minimally, it would be good to establish where in that reference I or the readers should look.

Given the flexibility in the word "simulation" in the literature, it is not immediately obvious to me that the ensemble of simulations should converge in the limit to a stable (sensu?) and stationary normal distribution, nor is it immediately obvious that we have converged to that limit should it exist.

How do we know that the 1000th generation is appropriately well-mixed? It seems likely from Figure 2 that there may not have been a transition regime for low mutation rate models, in which case the system seems to remember its initial state, and that's neglecting that transitions between stable species populations are likely not at random from all possible stable species populations.

How do we know that the "individual generation" time chosen by the authors is the right generation time (recall: definitions in this manuscript are somewhat murky to begin with), as opposed to measuring via "species generation" time? Comparing with the Markov chain Monte Carlo literature, I'm looking for something analogous to convergence diagnostics.

(RC 10, 19) Where related to RC 6 (above) I am not satisfied, but I understand the authors argument for the latter half of my comment. I suspect that line 81 should probably be changed to better capture their response though. Maybe "which is the limit on the mutation rate beyond which one generation is unable to be used to predict the next". This removes awkwardness around "biological information needed for continual survival". If the authors do not think it would be distracting, they might consider adding their figure into Figure 3.

(RC2 6) Line 94: is "organismal" necessary in describing the individuality scores? I understand from 4.2.1 that it is a specific type of individuality, but the way it is introduced in the text does not make that clear and it seems like "organismal" might cause more confusion than it prevents when viewed in the context of the unclear biological scale. As it is, it is somewhat unclear when using individual, group, and species in section 2.2, especially as individuals in this section refers to (groups of) species rather than agents, which are also in the model. Treating this explicitly here would help (a la line 131). Perhaps "different group sizes of species" -> "groups of different numbers of species" might also help a little.

(RC2 7) Line 97: it would be clearer to use explicit repetition, i.e. "knowledge of the group's past population".

(RC 12) Figure 4: I'd like to see a mention of the number of simulations averaged over in each of the mutation rate lines. Additionally, I'm happy with the blues and orange colors, but I'm having a hard time telling the reds apart, esp. 0.04 and 0.042.

(RC2 8) Line 102: "facilitating" seems an odd word choice. Perhaps "modulating" or "regulating" might be more appropriate.

(RC 13) Line 104: Why do the authors list the range of peaks as between 5 and 9? There don't seem to be obvious peaks beyond 7. Are they referring to additional unplotted data?

On a similar note, lines 133 - 134 and Figure 4 don't seem to be consistent. Ignoring the spurious tick marks on the x axis (4b), it would seem that 5 was just as commonly maximal, and possibly over a much broader range. Further (4a) would suggest that 4 - 7 are near equally important for low mutation rate regime. The authors may want to adjust their explanation on lines 133 - 134, as I'm bothered more by the inconsistency than by the scale of analysis.

(RC 14) Line 105: "higher-order organisation still persists". The authors appear to be interpreting the individuality scores here, but it isn't clear from the main text what a high or low score means except in a vague and relative sense. Indeed, it's not clear in a practical sense what it means to measure "the total information shared" (line 304), even if the authors place it on a relative scale (line 318), although they do not place it on an obvious relative scale (Appendix D and Figure E.4, and responses to RC 6 and 31 above)). Could the authors show a practical example of what the individuality scores mean somewhere? I appreciate that this is likely trivial for the authors, but it is an interdisciplinary communication problem.

(RC2 9) Section 2.3 would benefit by reminding readers of definitions of information transfer (species-cross predictability?) and integration (species-mutual predictability?). Figure 1 did show its value here though!

(RC 29) Upon re-examining Figure 4, I agree with the authors that examining a neutral model is an important baseline to compare with. Unfortunately, I don't think the authors have used it as such (e.g. the markedly different scales and intent of Appendix E).

Appendix E: I was a bit surprised at the lack of detail of the neutral model. What is the reasoning for fixed positive only interactions among species? Mutualism is traditionally not well regarded for ecosystem stability arguments (e.g. the introduction of "The stability of mutualism" by Lewi Stone (2020) or the "orgy of mutual benefaction" as oft quoted of Robert May, but I believe there have been many recent works which prevent the instability while incorporating mutualism such as "[B]alance of interaction types..." by Qian and Akçay (2020)), so it doesn't seem surprising that there'd be a lack of persistence, which is what the authors seem to be measuring information about. In this case, it seems less a neutral model and more the not-quite most obvious, negative extreme. (Competition would be a more obvious negative extreme.) Even then, it looks like some of the scales might be higher than that encountered in Figure 4a, although it is hard to tell even side-by-side.

Figure E.3: I'm a bit perplexed. My understanding of the model is that the interaction matrix is initially fixed, and individuals effectively choose a species identity when born, biased by their parents. If the same six species (or are the species numbers spurious?) and same interaction matrix are used every time, how would their network structure change? Similarly, line 200, is it the *observed* interactions become more mutualistic over time, or is the interaction matrix as a whole changing?

(RC2 9) Line 160, Φ^R should likely have a citation at this juncture.

(RC 7) Line 168, I'm not sure how much emphasis the authors want to place on the phase transition. I imagine from their response that it is already known in the literature and so it is fine to acknowledge and remark on, but I don't think for the current audience that it is a particularly biologically interesting conclusion. Indeed, the transition region itself might be of interest, as is the stable region, but the unstable region of breakdown I'd be surprised to hear of a biological analogue for.

(RC 21, 20 below) I think the primary problem in Figure 6 now is I have no idea if these effect sizes are of importance (theoretical or practical), and the paper hasn't given me the tools to understand it. E.g., are the individuality scores (range of 0 to 2?) measured on the same scale as the entropies (range 0 - 0.08), as suggested by lines 156 - 157 and AR 21. In which case Figure 6 is saying about 10% of the "predictability" of the system is from the environment pre-transition but jumps to ~25% for scale 6 during the transition (but, given we're dealing with information, it isn't immediately obvious how this maps to something like a time series forecast(pseudo- R^2)).

(RC 20, 22) It appears that the authors may have forgotten to remove this panel? If not, I'm not sure if the third panel with the new name is necessary if it is (mathematically) equivalent a subset of the above two panels. That's introducing a new term in an already jargon heavy paper for this audience for seemingly no pay-off. If so, if continuity was the goal, perhaps the authors might consider combining Figures 5 and 6.

(RC 24) I am familiar with LaTeX and how it can be customised to place figures in the text. In particular, it does not need to guarantee 1 image per page if the authors do not want it to, and it can make the job of reviewing easier if the figures are located around where they are mentioned in text. Blaming LaTeX is in some sense abrogating responsibility, although I can understand not wanting to overly fiddle when it won't be the final format.

(RC 25) I'm a bit confused by the state of the paragraph lines 169 - 183, especially now that I think I have a better grasp on the manuscript due to improvements elsewhere. It seems to pretty commonly be at odds with itself, but this might be imprecision in phrasing. The authors appear to be saying that

- 1) whether it is easier to predict for a group of species or a single species changes depending on the mutation rate in non-linear ways. (Sure.)
- 2) it is easier ("enhanced information processing [by the observer]") to predict in the transition region (but this seems to conflict with Figures 5 and 6, suggesting that the authors mean only for individual species).
- 3) that the system itself responds to information ("access[es] many different modes of information processing", but the system does not have agents that process information, so this appears to be a misleading phrasing).
- 4) the system changes more often in the transition region ("increases entropy" and thus decreases predictability, but this seems to be in conflict with 2) above).

5) "the system [finds more] stable group-level interactions that *adapt* [at lower mutation rates]" (but if the group is adapting, the system wouldn't be judged to be in a stable community, and it is unclear if systems are "more stable" when found in 0.001 then in 0.04 mutation rates. One would presumably need to look at the probability of exiting the population configuration as a function of the mutation rate and show that population configurations found at low mutation rates have curves everywhere lower than those found at 0.04, or something similar, especially as the mutation rate obviously affects some notions of stability. The comment becomes sort of trivial (of course systems at higher mutation rates are less stable) or completely non-obvious (systems found at higher mutation rates but not found in lower mutation rates may be less stable or may just be harder to observe in general)).

6) "[t]he system then leverages the enhanced variability... of single species to evolve towards metastable states" (but the system doesn't arrive at stable communities when the mutation rate is too high, nor is it clear how the system "evolves" in the biological sense when it is just alternating through species seemingly at random).

Also, above, I said "changes more often", but the authors might wish to remind/show the audience that (it appears that) there is truly more frequent switching between stable community configurations as well as switching taking longer, judging from Figure 2.

(RC 28) Lines 206 - 208: I'm not sure the authors want to connect too closely to harsher environmental conditions, since that would make mutation rates a species level response to what I would expect to be a rise in p_{kill} , which is not at all what the model does.

(RC2 10) Lines 210 - 212: Again, if I've understood the model correctly in that species are not changing their traits through time, I'm not sure that "individual [species] adaptability" makes sense as a conclusion. It seems more likely that if the environment is noise that your current population size becomes more important and predictive than the population sizes of anything else in your (biotic) environment, including your competitors, predators, prey, and mutualists. As such, I don't think the authors have demonstrated this trade-off.

(RC2 11, RC 36) Line 214: "readily applied to real data", but don't the authors need a huge number of simulations of a huge biological length from which they then sample a huge number of subsets of species, judging from what we know from Appendix C and the methods (now lines 283 - 285)? This doesn't seem readily applied, especially as the authors have not dealt with species with varying (individual) generation times.

(RC 27) Line 267: It would be nice to have a small addition saying why these parameter ranges are standard (biological? mathematical? stable? dynamic?). The authors appear to have mentioned Di Collobiano et. al. (2003) twice in their response in this context, but this reference does not appear in the main text. Given it came up twice in their response, I was a bit surprised not to find it and its relevance mentioned in the manuscript.

(RC 35) Line 303 - 304: given the increasingly interdisciplinary nature of the paper (see change to Appendix C), the authors should also define the I operator explicitly.

(RC 36) Line 312: how many simulations is the ensemble?

Line 313: when sampling different subsets, do the authors sample from the simulations at random times and retrieve that subset of species, or do the authors mean the more literal "choose a random simulation, choose a random subset of species from all subsets of species possible in that simulation's parameters, then look for a time in the simulation where that subset is observed"?

Line 313 - 314: Why should we consider a rank-ordered list rather than a simple random sample? That would seem to bias the results towards integration to me.

Version 2:

Reviewer comments:

Reviewer #2

(Remarks to the Author)

Thank you to the authors. I read the manuscript with a fair amount of positivity. I have a few comments that the authors may wish to consider, some responses to their comments, and some corrections that were not caught in the previous revisions. Overall, I think I can recommend that the manuscript needs only what I think of as minor revisions, as the following should be minor textual corrections.

Comments:

1) I am not sure why, but not all of the changes were marked in blue (minimally in the abstract, change about information processing).

2) Apologies for just noticing this, but the authors have not actually defined which "diversity" they are referring to. I think I implicitly assumed it was species richness, but this should be explicitly stated in the manuscript (line 82?).

3) The caption of Figure 1 potentially deserves another pass. All of the content is there, but it could be refined slightly. Lines

2 - 4 of Figure 1 maybe should be broken up differently (I would make the colon a full stop and the full stop before implying a comma). The panel labels are not in the panels themselves. The description of Panel A does not mention information storage directly even if the implication is obvious. The authors say "is measured by information transfer", but this seems analogous to saying the distance from the top of my head to my feet is measured by my height; I think most people would just say that that *is* my height.

4) I wanted to pull this out of RC13, but the authors I think use stable to mean a few related things in the manuscript. Each is used consistently and correctly in its context I believe, but this might be a point to make clear at each first usage. Minimally, I would expect line 303 to mention which version of stable the authors mean, since both timesteps and distributions are mentioned in that same sentence.

5) I also realised as I was inspecting Figure 4 that the authors have not listed exactly what they are plotting throughout in each caption. This should hopefully be a quick fix. (E.g., Figure 4, mean and percentiles? Figure 6, mean and standard deviations?)

6) Looking at the analysis code provided and at Figure A.1, the authors should probably define what they mean by core in Figure A.1. It might also benefit biologically minded readers to remind the reader that reproduction rate in this figure is after taking account of species interactions, etc.

Responses:

RC3, 12: Thank you to the authors for making their analysis code available. The authors should cite JIDT (Lizier's 2014 publication, linked on the JIDT Github page) and mention it and their analysis Github in the manuscript. Aside from that, their code makes it clear that they have assumed Gaussian copulae for their data. Could the authors plot or test their data to show that this assumption is valid and double check (but do not need to show) that there is no further temporal structure?

The authors show in RC 13 that at various time points the simulations are close in distribution at the population level and state that they are happy with the temporal dynamics in RC 14. (I would not be happy in their shoes and would only begin to feel comfortable around 40,000 if I were using their plots, especially as scale 6 still appears to be drifting to me at that time and the authors are using up to scale 15.) I think the information from these plots, combined with mentioning the approximately Gaussian structure address my original concerns about appendix C and should be included therein.

RC15: I do understand that this is the standard way that Tangled Nature models are analysed, but I do disagree with some points in their response that are worth discussing, but probably outside the realm of the manuscript at this point.

$T_{\{gen\}} = N(t)/p_{\{kill\}}$ is the expected number of timesteps that would result in the minimum sufficient number of kill events to remove the existing population assuming that there is no replacement or change in population size. This is slightly different from the average number of timesteps to randomly kill off the existing population, in the sense that an individual through good luck could persist pseudo-indefinitely and be a part of every generation. That is, the average number of timesteps used to randomly kill off a population should be higher than the minimum sufficient number. The latter would require labelling each individual in the current generation and counting how long it takes to kill every labelled individual (after which the cycle would repeat). This describes something more in line with a coupon-collecting problem. The former is in line with a simple sum of geometric distributions (over the number of trials, rather than failures). (Using the word "needed" is ambiguous in this context. I only need in theory 100 eggs to put one in each of 100 baskets, but in practice I need many more if there is error.)

This relates to the asynchronous updating comment because the system is not guaranteed to be entirely updated, although I am happy to admit that Metropolis-Hastings asynchronous updating is something I have not studied in detail.

There are a few different timescales in the Tangled Nature simulations depending on perspective. The authors have chosen a meso-timescale in which they try to aggregate over individuals. In contrast, if one was interested in the behaviour of the individuals themselves (which would be silly here!) one could try to look at singular events over an individual's life span. If one is instead interested in the state of the system as a whole, one could look at the timescale of a species, (if we treat an entire species as a unit of interest, how long does a species last for, e.g., on average) which would be the length of the horizontal lines in Figure 2 of the manuscript. One step broader, one could look at the timescale of the ecosystem, which would be the length of the blocks in Figure 2. I tend to be more interested in the latter two timescales in my work. Given the blocky nature of Figure 2, I surmise that these time scales are not necessarily all strictly linearly related.

That the authors choose a timescale that does not really seem to be what they are interested in is, to my mind, somewhat surprising, but the results from RC13 and 14 and (I assume) the good fit from the Gaussian copulae means it probably is not worth worrying about.

RC21: Taking on board the authors' points, I do wonder what scales they should be calling out in the text. The image (Figure 4) and their explanation seem to indicate a range of possible ranges, with the largest as 3 - ~13 (Figure 4 0.005), and the smallest listed as 5-8 (lines 201 and 206), but I think I also saw 5-9 in the caption and 4-7 in AR21. Are these different ranges intentional? Additionally, I was wondering if there is a statistical test for this, although I understand if there is not.

RC23: I can indeed appreciate the nuances that the authors are indicating here, but these details should be moved into appendix E!

RC27: I do suspect the authors are a good deal more familiar with the concept of entropy than the audience that I think (ecologists and biological modellers reading Communications Biology) would be engaging with this manuscript! While I appreciate the details presented in their response (and perhaps those details should feature in the introduction), they don't actually answer the question I was intending to pose in RC27.

I had intended to ask how, if I were to try to put this into practice for a different system, would I interpret the results quantitatively? Are these values all strictly to be interpreted as relative and not to be compared to a different model or system (or even between figures)? If I look at Figure 6, Species -> Environment, Scale 6, rates 0.01 and 0.02 and surmise that I have a significant difference here, is the difference of ~ 0.03 a big enough value to be excited by, or should I expect other entropies to be so much larger that the difference is effectively meaningless in practice? This is asking a question similar to what is a "big" effect size elsewhere (e.g. discussion of Cohen's d and the like).

RC37: I don't necessarily find AR37 reassuring, but I can understand what the authors are attempting (in the sense that they think there is something in their data and are trying to bring it out). I'd like some of this brought out in the manuscript so that readers can decide for themselves to be bothered or not. I'd recommend refining the new lines 334 - 335 sentence to something like "The rank ordering of subsets of species makes it more likely to sample subsets with similar population distributions and interaction structures, improving comparison across different subset sizes." This makes the authors' justification clearer in the paper itself. Personally, I believe some of my colleagues would be more interested in the unranked samples or in comparing ranked and unranked samples.

Minor Corrections:

- 1) The authors say in a few places, usually in figures, "the paper", but I thought it was in some places ambiguous as to whether it was referring to a citation or the main text. These places would benefit with a link to a section (or citation, as necessary).
- 2) Lines 77 - 78 are also a bit tricky to read. Maybe change "Therefore, triggering mutations among existing species..." to "The population of species experience mutations in response..."
- 3) Line 142: there is a missing parenthesis.
- 4) Figure 4 is missing a space at the beginning of its caption.
- 5) Figure 5 has "Scal1" and inconsistent capitalisation compared to the rest of the manuscript.
- 6) Line 181 seems to have a full stop where I would expect a comma.
- 7) Line 192: should "as mutations increase" be "as mutation rates increase"?
- 8) Line 193 seems to have an odd phrasing of cause and effect. I would have thought the shorter metastable states observed in the transition region reduces predictability at higher organisational levels and enhances predictability at the single species level instead. Instead, the authors have predictability (something I'd expect to be an observation about a system) driving the system.
- 9) Line 210 is missing a space (",since").
- 10) Line 222 "a stable population" or "stable populations"?
- 11) Line 267 seems to have a full stop where I would expect a comma.
- 12) Line 301, Tangled Nature was left lowercase.
- 13) The authors will want to be careful regarding equation 1 in the final form as it showed up differently between the version with tracked changes and the one without.
- 14) Line 326, should it be "known about" or "known from" rather than "known by"?
- 15) Appendix A: Hamming should be capitalized.
- 16) Appendix B: the authors should probably match the colour scheme to Figure 4.
- 17) Appendix C: the first line should have Tangled Nature capitalized and the full stop replaced by a comma.
- 18) Appendix E: 2nd Paragraph, 1st Sentence: this probably should read: "We consistently find a very peculiar interaction structure when looking in systems with mutation rates for which higher-order organization is present."
- 19) Appendix F: a space is missing in "equation6".

Responses to comments from reviewers of paper

Manuscript ID *COMMSBIO-24-3608-T*

“Information dynamics and the emergence of high-order individuality in ecosystems”

RC 1 *Reviewer’s Comment*

AR 1 Authors’ Response

We would like to thank the editor and the reviewers for their thoughtful work in reviewing our paper. The provided comments have helped us substantially improve the revised manuscript. In order to aid the review process, the modified text has been highlighted using colour blue, in the tracked changes version of the manuscript. In this response we refer to these changes using line numbers (in short L#), on the left margin for the revised manuscript.

On behalf of all co-authors, many thanks for your attention.

Sincerely,
Hardik Rajpal

Responses to comments from Reviewer 1

RC 1 *This is an interesting theoretical modelling paper. The paper uses a computational re-implementation of a model of interacting genomes called the ‘tangled nature model’ (Christensen et al, 2002). The authors explain in the methods that the tangled nature model includes a fitness function in which the chance of reproduction by genomes is affected by the interactions between the genomes that coexist at a given time. The ‘genomes’ are strings of zeros or ones of a set length. The ecological interactions are randomly allocated to distinct genomes and can be positive or negative. The study then calculates from the simulations under this ‘tangled nature’ model a range of measures from information theory that aim to quantify ‘interdependencies between time series of species populations obtained from the simulations’.*

Throughout the manuscript the term species appears to refer to a distinct type of genome, therefore I have suggested that the use of species be replaced by another term to avoid confusion with the standard biological species definition and to add clarity with regard to level of selection debates.

AR 1 We agree with the reviewer that there is potential for confusion with these issues. We note that the use of the binary vector in the manuscript corresponds to an abstract representation of a particular species. Members of the species can reproduce, and the resulting mutations to the binary vector can lead to the introduction of new species into the ecosystem. Correspondingly, the

binary vector should be understood as a representation of the *pangenome* of the species, i.e. the part of the genome that is similar across the members of the species. We have clarified these issues in the revised version of the methods. We hope this change addresses the reviewer’s concerns, while clarifying the level at which our analyses take place.

RC 2 *One of the key informatic measures used by the study is an implementation of a measure proposed as an ‘information-theoretic solution to identifying the boundary between an individual and its environment’ by Krakauer et al 2020. These measures could do with more explanation within the current manuscript e.g. definition of the notation used below equation 3.*

AR 2 We appreciate the suggestion. We have expanded the explanation with respect to the applicability of the information measures in an ecological setting in the introduction (L# 30 - 39). In line with the comment made by both reviewers, we have explained the notations used to describe the information measures in the methods (L# 304 - 308).

RC 3 *The paper also considers a measure of transfer entropy, which would also benefit from further plain language explanation.*

AR 3 We have added explanations of the metrics in the introduction (L# 30 - 39). In addition to that, we have also added a figure that visualizes the effect each of the information measures aims to capture (Figure 1).

RC 4 *The authors state ‘In essence, these 249 measures quantify the extent to which the interactions between parts of a system drive the joint temporal evolution of the system 250 as a whole—a system has high integrated information if its dynamics strongly depend on the interactions between its parts.’*

I found it hard to get an overview of what the study is measuring, but I think it may be something like as follows. The study models the reproduction and mutation of model genomes (which are strings of zeros and ones). The genomes have randomly set, pairwise interactions which can be positive or negative if they exist. There is an overall population size and some resource limitation term to the fitness function for each genome type. One can then consider overall population fitness based on the numbers of positive versus negative (or non-existent) interactions, or the fitness of subsets of the overall population of genomes, over time. I think the ‘environment’ may designate the rest of the total population other than a considered subset, though this was not clear to me (e.g. as this is not what the use of the term environment would have initially led me to expect: I would have expected environment to refer to physical environment). I think the transfer entropy measure may measure something like the extent to which current population sizes (which result from the effect of the fitness function on genome reproduction), for a subset of genome types, predict future population sizes for the rest of the population other than the subset (with this remainder constituting the “environment”).

The results state ‘Special attention is paid to the dynamics observed in the vicinity of the ‘error threshold,’ which is the 2/16 limit on the mutation 65 rate for species beyond which any biological information needed for continual survival is destroyed in 66 subsequent generation31, 32.’

So, I interpret that the study is looking at the information on future population sizes inherent in current population sizes (resulting from genome fitnesses based on their interactions) when the genome mutation rate varies (possibly rates?). And, especially near the point where the mutation rate gets so high that, I guess, future genomes get decoupled from genome fitness (e.g. that of the parental genomes), as there is so much mutation that genomes are effectively randomly generated at each time step.

My biggest question about the results and conclusions is to what extent the variation in information measures used is a necessary consequence of the persistence of groups of interacting ‘species’ (which I believe is a previously known result as it has citations 28 and 30 in the text in the first section of results). What alternatives are mathematically possible? If you have fixed versus variable total population size, is it possible to have groups of genomes which are set to have interactions and persist for some period of time, that do not behave e.g. as ‘individuals’ using the information measures shown? At the moment, I don’t clearly see a distinction made between null and alternative hypotheses for the information measures applied, and I think this should be added, particularly as I am wondering whether information individuals are the null (or perhaps only possible?) hypothesis given the known properties of the tangled nature model.

AR 4 We thank the reviewer for this thoughtful interpretation of our work, and for raising this helpful question that let us to explore the properties of the TaNa model that enable higher-order organization. To address reviewer’s concerns, we analyzed the interaction structure of the subset of species at scale 6, which possesses the top 1 percentile of the individuality scores, for different mutation rates. We find that at low mutation rates, where higher-order organization exists, the interactions are asymmetric (implying a hierarchy). Especially a strong predator-prey interaction is dominant. For mutation rates in the transition region, the asymmetric interactions and the predator-prey interaction fade away. These findings highlight the role of the interaction structure in supporting higher-order organization. To test this finding, we simulated a neutral TaNa model where every species interacted with each other with a constant positive weight (+1). Replicating the organismal individuality scores on the simulations generated by the neutral model. We find that no higher-order interactions exist beyond scale 2 for any mutation rate. This analysis is presented in Appendix E and is referred to in the main text (L# 119 - 125).

RC 5 *Because the paper builds on previous theoretical work, I found it quite hard to tell in the abstract and results summary which specific parts were new to the present study. This should be more clearly stated. I believe, from reading the paper, that the novel results are those that relate to the effects of mutation rate specifically and the application of information measures to the studied ecological model. However, mutation rates don’t appear to be clearly highlighted in the abstract, where mutation rates are not explicitly mentioned. I suggest more clearly demarking in the abstract and conclusion the novel results.*

AR 5 We thank the reviewer for raising this very useful suggestion. We have now added the novel results related to mutation rates and scales of organization, and the fact that emergence happens naturally because of mutation and reproduction clearly in the abstract, results summary (L# 54-82) and the discussion section of the paper (L# 208 - 212).

RC 6 *The authors state in the main text ‘We study the 210 changes observed in the dynamics of the model when a key parameter p_{mut} is varied. This parameter represents the selection 211 pressure introduced by the ecosystem. Since changing environmental conditions lead to more mutations, p_{mut} can be considered 212 as a proxy for controlling environmental selection pressures⁴⁴.’*

p_{mut} is the genome mutation rate. This could perhaps do with more introduction and discussion e.g. with regard to the general biological plausibility of variation in mutation rate under changes in environmental conditions. E.g. It would be helpful to briefly review the literature on when mutation rate may vary, or conversely, be invariant, over specific timescales, in different parts of real genomes and biological groups. For example, below is a link to an article with similar suggestions in humans which was presented as controversial. The recent paper below also presents related suggestions as counter to standard views.

I wonder how easy it is to distinguish between variation in mutation rate and variation in survival/fixation rate in the real world for mutations with fitness effects.

AR 6 We have added a brief description on the impact of abiotic environmental factors on the mutation rate in the introduction (L# 60-62) and the methods section (L# 268 - 270). Reviewer makes an interesting point about differentiating the variation in the genome caused by environmental versus fitness effects. A data-driven study of the environmental, genetic and population level features may be able to pick this up. However, it needs future work to identify the key features in each of these categories to pick up the relevant features.

RC 7 *This is a paper which has been submitted to a general biology journal but is likely to be quite difficult to understand for many biologists, as we should not assume the general biological reader will have any familiarity with information theory etc. I therefore suggest a number of improvements to the accessibility of the paper. I have suggested a number of specific points below where more explanation of the concepts used should be given in the abstract and introduction.*

I think all the technical terms in the paper from “information theory” down should be briefly explained at, or close to, first use.

AR 7 We understand the need to make the manuscript more accessible and have incorporated most of the suggestions presented by the reviewer. Especially to introduce measures from information theory, the revised introduction and Figure 1, ensure that a description is provided early in the revised manuscript.

RC 8 *I also believe that terms like ‘species’, ‘environment’ and ‘mass extinction’ are used in ways that differ from standard biological uses of these words, and I suggest changing the terminology to make this less confusing where possible (see details above and below), and clarifying with an explicit definition anywhere else.*

AR 8 We have considered both reviewers’ concerns in addressing the use of these terms and clarifying how they relate to the biological use of the terms. Hopefully, these changes throughout the revised manuscript will make it more accessible to readers.

RC 9 *I also suggest working to make the key mathematical modelling concepts easier to follow. I would suggest listing the notation after each equation, as at the moment one has to pick back through the preceding text to work out what the variables are. I also suggest some presentation methods could be used such as colourised maths, in which conceptual parts of the key equations are coloured with a key to plain English explanations of what they mean. I would also suggest adding simple worked examples to the methods. For example, some toy example genomes could be fed through the equations and the output shown e.g. an example of the calculation of fitness with equation 1 showing explicit examples of the types of inputs and variable values used.*

This (worked examples) would also be extremely helpful for the information measures. I also suggest that plots of the methods would be helpful e.g. it is stated ‘Note that the probability of reproduction is non-linearly related to the fitness function.’ At equation 2. This would be easier to follow if method plots were provided showing examples of inputs and outputs to equation 2. The figure captions would also all benefit from more information, including stating the meaning of all variables and replacing initialisms with full text.

AR 9 Considering the reviewer’s suggestions, we have added colourized maths to the fitness equation and described the notations used. We have also added a figure showing the example of fitness and reproduction probability calculations (Figure 8) in the methods. Figure 1, added in the introduction, introduces the information measures more clearly.

RC 10 *Abstract ‘transition region close to the error-threshold’ Please clarify at this point the usage of transition e.g. state from what to what. Indeed this whole sentence could do with much more explanation. There is a good, brief explanation of the error threshold in the results, but this should be explained earlier.*

AR 10 In the revised abstract, we have added a line briefly explaining the error threshold and improved the structure of the sentence.

RC 11 *‘Our results show that evolutionary dynamics can lead to clusters of species that act as a self-perpetuating group, that acquire information-theoretic agency. However, this higher order organization breaks down during the transition region close to the error-threshold, where single species gain higher information-processing abilities.’*

Please can you clarify whether the first result mentioned above is novel to this paper i.e. not in earlier ‘tangled nature’ analysis. I.e. it would be helpful here to more clearly demark all of the results which are novel to this study specifically.

AR 11 The emergence of information-individuality in groups of species is a novel result of the paper and has not been a focus of previous tangled nature papers. In the revised abstract, we have further highlighted the role of mutation rates in this sentence.

RC 12 *‘Overall, our findings provide quantitative evidence supporting the relevance of high-order structures in evolutionary ecology, which can emerge even from relatively simple processes of adaptation and selection.’*

Relevance to what?

AR 12 We thank the reviewer for pointing out this ambiguity in the sentence. We have rephrased the sentence to clarify that the results highlight the emergence of higher-order interactions from evolutionary dynamics without externally forcing it.

RC 13 *Line 1: grammar: dynamics are <https://english.stackexchange.com/questions/488664/the-dynamics-is-or-the-dynamics-are> See also line 193 etc.*

AR 13 We have corrected this grammatical error throughout the manuscript now.

RC 14 *Species produce offspring or something rather than ‘species reproduces offsprings’.*

There are quite a few apparent plural/singular mismatches throughout, perhaps Grammarly might pick these up if you run the text through.

AR 14 Thanks to Grammarly, we have corrected these errors as suggested by the reviewer.

RC 15 *Line 4: Please complete this sentence so it is comprehensible without looking at the references e.g. such as in experimental evolution.*

AR 15 We have now corrected the sentence structure.

RC 16 *Second paragraph. I would suggest that the standard current emphasis is still on the individual as the fundamental level of selection. E.g. Even under a ‘selfish gene’ view, genes still have to act, to some extent cooperatively, within their vehicle, the individual (e.g. as discussed in Dawkin’s ‘the selfish gene’).*

AR 16 We thank the reviewer for clarifying the issue. We have added a sentence (L# 7 - 9) to clarify the relationship between the genes and the individual under the gene eye view of evolution.

RC 17 *Line 11. Indeed, though these debates don’t just cover above-individual levels. I would suggest debates extend across all possible levels of selection including genes, cells etc. within individuals. E.g. See work of Okasha (already cited) on evolution of cancer etc. So, for accuracy in representing the literature I suggest slightly modifying this sentence to note this.*

AR 17 We appreciate the reviewer’s comment on considering sub-individual scales of organization. We had initially overlooked this, but have now acknowledged the organization at the levels genes, cells etc. in various parts of the introduction. (See L# 12 - 14)

RC 18 *Line 12. Please briefly introduce/further explain persistence theory.*

AR 18 We have added a brief description of the persistence theory in the introduction here (L# 12-14) as well as later where we relate the persistence to information (L# 34-39) individuality.

RC 19 *Line 15. Emphasis from singer to song. It is very surprising to me that you cite the*

species as the singer here when the standard view would be that selection acts on the individual and the species is a higher order group. What is the justification for this?

AR 19 We thank the reviewer for highlighting this difference in understanding. In our reading of the ITSNTS theory of persistence selection. The interaction structure and processes implemented by "entities" are selected for persistence. Doolittle and Inkpen explicitly refer to these "entities" as singers in their 2018 PNAS paper. They also highlight the example where multiple species collaborate to perpetuate the nitrogen cycle. In this example, species are clearly the singers and the nitrogen cycle is the song that arises out of the interaction structure among the species. To further clarify that the persistence selection can target interactions between species and individual organisms we have mentioned them explicitly in L# 17

RC 20 *Line 26. Citation or justification needed.*

AR 20 We have added a justification for the comment (L# 27 - 29). Basically highlighting the fact that novel metrics that can quantify higher-order interaction directly from population timeseries data are needed. As model parameters like the interaction structure are usually not fully known in datasets from field experiments. However, we couldn't find an exact reference. We will welcome any further suggestions from the reviewer on this matter.

RC 21 *Line 27. Do you mean combining or using, if combining, what with what?*

AR 21 We have replaced the word with "using" (L# 30) as it is more appropriate to the use of information theory on the model.

RC 22 *Line 33. The 'tangled nature' model is stated to be well-studied. I had, personally, not heard of this so I looked at the cited paper. This has 9 citations so is not likely to be generally known (I suggest removing 'well-known' from the paper). Please therefore introduce what this model is in the introduction and explain why it is a suitable study model for the research question.*

AR 22 We cited a recent recent revision of the original Tangled Nature Model to direct the readers to the updated interpretations. Although the original TaNa paper (Christensen et. al. 2002) has over 150+ citations, we agree that many readers might not be aware of the model. Therefore, on reviewer's suggestion we have moved the description of the model to the introduction.

RC 23 *The first paragraph of the methods, at least (and possibly the 2nd and 3rd), appears to me better placed in the introduction as it is an introduction of previous work rather than a description of methods new to the current paper. Putting this sort of explanation in the introduction is likely to make the paper more understandable if read from start to finish.*

AR 23 We have taken the reviewer's advice and moved the description of the model to the introduction (see L# 40 - 51)

RC 24 *Given the comms bio format, with methods at the end it might also be valuable to place a clearer explanation of the key information measures in the introduction e.g. in a box.*

AR 24 In the expanded introduction, we have added a brief description of the information measures along with Figure 1, which describes the measures of information storage, transfer and integration.

RC 25 *Please explain in the introduction what you mean by selection effectively acting. Why effectively not just acting?*

AR 25 We have revised this line and provide clarification on what we mean as the action of natural selection, in the context of interaction structure among the species. (L# 14-17)

RC 26 *Line 30 ‘The basic hypothesis behind these approaches is that if a group 30 of individuals can enhance the prediction of their joint future, they can better adapt and thus survive.’. Is this your hypothesis or that of Krakauer et al. 2020, cited in the previous sentence? I read that paper some time ago and what I got from it was that it was about how information theory can be used to identify the level, or levels, of individuality. It would be helpful if you could be more explicit about how far that paper went towards this hypothesis e.g. by quoting what you think is the most relevant text.*

AR 26 To establish the novelty of the findings presented and the paper and to the extent it is different from Krakauer et. al., we have added quotes from the original paper. We have also added a few lines describing how we address the limitations discussed by Krakauer et. al. (L# 38-39 and Footnote 1)

RC 27 *Line 35. Every endeavor should be made to make this paper accessible to a biological reader. Therefore, I suggest here briefly introducing in plain language what all the terms/measures mentioned are under an assumption that many potential readers will not be familiar with information theory, integrated information, information dynamics, the measure of individuality mentioned, error in the informatic sense etc.*

AR 27 These terms are briefly described in the text (L# 31-33) and later explained in Figure 1, in the revised manuscript.

RC 28 *Line 37/ ‘Overall, our results provide quantitative evidence 37 suggesting that groups of species exhibit higher-order individuality for biologically plausible mutation rates.’*

Assuming the tangled nature model (which hasn’t at this point been described) is a theoretical model, this should be rephrased to make clear that what is provided is a theoretical analysis (i.e. modelling on a model) in support of the conclusions, not evidence in the sense of experimental or observational biological data. Theory is fine, but it’s not very clear to me reading this whether data has featured or not in the study, so I suggest clarifying.

AR 28 We have completely revised the results summary of the introduction and added the clarification regarding the fact that the results are theoretical, derived from a modelling study. (L# 54-62)

RC 29 *Line 38 plural individuals*

AR 29 As the sentence refers to a single species, we have used the singular "individual" to refer to it.

RC 30 *Line 37-43. There is a results summary here which looks like it would be better as a conclusion. Here, it's not very easy to follow. Especially, if you want to leave a results summary at the beginning, I suggest first expanding the introduction so the reader has more preparatory information at this point.*

AR 30 Taking the reviewer's insightful comment, we have now significantly expanded the introduction and modified the results summary to reflect the role of mutation rates, error threshold and emergence of higher-order individuality.

RC 31 *The captions for the figures particularly, 4 onwards are too brief. Imagine a general biological reader reading in order of abstract then figures before the main text. I think the captions are incomprehensible if used to initially survey the paper and, ideally, they should be understandable without referring to the text. E.g. Fig 4. 'Mean AIS for the two scales of organization – Individual (Scale 1) and Higher Order (Scale 6) – varying with mutation rates.' Please briefly explain what all these things are without abbreviations and set them in some context. Please look at all the figure captions in this regard. I suggest starting with a title in plain English that gives an overview of the point of the figure, then explanations of all the technical terms and information on where the data come from for each figure e.g. including info such as "theoretical model of x based on y with parameters z showing a" etc.*

AR 31 We have taken the reviewer's suggestion seriously and updated the captions of all the figures presented in the paper.

RC 32 *Typo in caption of Fig. 6 species.*

AR 32 We have now corrected the error in the revised manuscript.

RC 33 *Line 188 Please explain what a binary genome is.*

AR 33 We have added a sentence explaining what does the binary vector, which is an abstract representation of the pangenome of the species. (L# 225-228)

RC 34 *Line 189 Please give more detail on the level/unit of individuality here. Species are described as the units but replicating genomes are also mentioned: are the species not higher-level groups of genomes i.e. individuals? It seems to me that the genome i.e. individual is the unit. If not, why not?*

AR 34 We thank the reviewer for giving the opportunity to clarify this crucial point. Here, the species are represented by a binary vector which is an abstraction of the pangenome of this species. Such a modelling choice is common in many mathematical models of co-evolution where the binary vector identifies a species. When a member of a species reproduces and undergoes mutation a new

member of a different species is created. We have clarified that the binary vector is a representation of the pangenome of the species rather than a genome (which can ofcourse vary within a species). These changes are explicitly mention I L# 225-228 and then used throughout.

RC 35 *‘No groups or hierarchies 190 are defined a priori.’ Then how are genomes matched to species?*

AR 35 We clarify that no ”structured” interactions are imposed as the interactions are sampled uniformly at random.

RC 36 *I am wondering here if the terms species is being used for what is more like a biological individual or perhaps a clone: a copy of a genome. Species may perhaps be used more in the chemical sense of an identical molecule than in the standard biological sense. ‘Each species is defined using a unique binary genome of length L , comprising an ecosystem of a total of $M = 2^L$ possible species.’ This suggests that each ‘species’ used in the current manuscript is one of the combinatorially possible, distinct genomes, where a genome is a string of zeros and/or ones of a set length L . If this is the case, I suggest changing the term species to individual, or some other more suitable term, or simply using something like “distinct genome type” (without adding some extra name) throughout the manuscript. In any case, please clarify the definition of each name used and explain how this relates to standard definitions e.g. of biological species, individual etc., particularly considering asexual/sexual reproduction (as asexual reproduction is modelled in the paper). First, this is currently confusing since a biological species, in standard use, is a group of sexually reproductively connected individuals, which are not all expected to have identical genomes. Second, the paper relates to the unit of selection and if the ‘species’ in the paper are potentially most like biological clones i.e. copies of the same individual, everything stated as at the level of species in the paper would, I would have thought, be potentially better biologically interpreted as at the level of the individual.*

AR 36 We agree with the reviewer’s assessment that species as represented in the model using a single binary vector is different from the biological sense. In the revised manuscript we refer to this binary vector as an abstract representation of the pangenome of the species rather than a genome to avoid the confusion related to the within-species variability of the genome. We thank the reviewer again for pointing out this oversight.

RC 37 *‘populations of existent species’ This is confusing because it would be more common to have biological populations within species. If you are referring to groups of species please use a more appropriate terms e.g. clade if they are monophyletic groups of species, or a more general term such as set or group, in which case please explain how membership is defined.*

AR 37 We thank the reviewer for bringing this terminology to our attention. We have rephrased the sentence and used ”subset” to refer to the initial configuration of the system (L# 246).

RC 38 *Line 194. Singular strategy?*

Line 196 what does q -ESS mean? I would suggest avoiding abbreviations as much as possible,

throughout, as the paper is likely to be very hard for an average biological reader to understand even without them.

AR 38 In the revised manuscript, we have introduced the term quasi evolutionary stable strategies (qESS) in the introduction and the methods. However, incorporating reviewer’s suggestion we have removed the abbreviation q-ESS and replaced it with metastable state referring to the duration when stable species exist in the Tangled Nature Model.

RC 39 *‘Starting from a random set of populations of existent species, each timestep starts with an annihilation step where a member of a species, selected uniformly at random, is killed with a probability p_{kill} . This is followed by asexual reproduction, where a member of a species, selected uniformly at random, creates an offspring with probability p_{off} . Each element of the genome of the offspring undergoes a mutation with a probability p_{mut} . These mutations introduce new species in the ecosystem which then compete with the existing species. The state of the system is recorded after each generation, $N(t)/p_{kill}$ timesteps, which is the average number of timesteps required to kill all currently existing species. The reproduction probability p_{off} depends upon the fitness of the species at the given timestep.’*

‘a member of a species, selected uniformly at random, is killed with a probability p_{kill} ’ suggests that survival is random. While ‘The reproduction probability p_{off} depends upon the fitness of the species at the given timestep’. If I follow this, it suggests a model of evolution in which reproduction is subject to a fitness function but survival is not. This strikes me as biologically surprising. One could perhaps collapse survival and reproduction into reproduction probability, but in that case why have p_{kill} – it seems to me this may introduce a genetic drift component, which is the population is small may have a large effect. What is all of this supposed to be modelling, how and why?

AR 39 The reviewer is right to point out that the overall survival depends upon both reproduction and annihilation of the member of the species. Clearly longevity varies significantly amongst different types of organisms, for example contrast annual plants to redwoods. Nevertheless, the death of an individual is often accidental and independent of the average lifespan of a given type. E.g. a forest fire may eliminate redwood trees together with the annual flowers. This is in contrast to reproduction, which is more directly related to the capability of a given type to replicate and thrive in a given biotic and abiotic environment. This is the reason why we for simplicity have chosen the offspring probability to reflect the type of the organism while we let annihilation act as an indiscriminatory Poisson process.

RC 40 *Line 203 ‘However, of all possible interactions, the 203 number of permitted interactions is controlled by coupling probability θ .’ Please explain this e.g. how are permitted/disallowed interactions set.*

AR 40 Interaction between any two species is allowed with the coupling probability Θ . Thus this parameter controls the overall connectivity or sparsity of the interaction matrix. We have added this description to the text (L# 242-244).

RC 41 *Line 225. Please explain what mutual information is in this context. Please also explain*

stationary and non-stationary evolution in your context.

AR 41 In the revised manuscript we have provided a description of the relevance of information measures in the introduction. We have added a few sentences explaining non-stationary evolution in this section. Including, how we implement MI measures here over an ensemble of simulations as requested by Reviewer 2. Hopefully, these additions to the paper will explain the meaning and methodology more clearly (see L# 283-287).

RC 42 *Lines 231-235 ‘In recent work, Krakauer and others¹⁸ put forward an information-theoretic solution to identifying the boundary between an individual and its environment. This definition is based in principles of optimal self-prediction — i.e. if a subsystem can predict its future better than any of its parts, and any addition to the subsystem hinders its predictability, then that subsystem is deemed an information individual. This optimal self-predictability enables biological entities to control and navigate their environment⁴⁸, implying that selection for persistence^{10, 11} could be a putative explanation for their emergence.’*

I think this would also be better placed in the introduction to increase the understandability of the text and as it seems more appropriately placed there. It would be helpful to expand also on the explanation of predicting the future and the justification for the definition of corresponding individuality e.g. is this theoretical or actual, where is this information potentially or actually held and what does this mean if only theoretical. For example, most individual people might think of themselves as paradigmatically an “individual” but much physiological, nervous and cognitive function is apparently subconscious and/or distributed. Therefore, it doesn’t seem to me that the human brain holds an integrated model of itself or the wider body anywhere, nor is it able to predict its future states in a full sense. Would this mean that under the definition above, people are not individuals because they cannot predict the future states of their brain, body or their sub-components? Can you give a/some examples of how this prediction you hypothesise might actually occur in biological systems. A justification of these underpinnings in the introduction, if possible, might considerably strengthen the paper as well as making it more accessible.

AR 42 We have moved the section suggested by the reviewer to the introduction, describing the thesis of information individuality. Especially, defining what predictability means in terms of persistence and adaptability instead of cognitive predictions (see L# 34-39). Hopefully the added description provides clarity to the meaning of information individuality and enhances readability.

RC 43 *‘Carrier capacity’ Do you mean “carrying capacity”? Please also give a bit more detail on this variable μ to make clear what the last term in equation 1 is like. E.g. If this was carrying capacity (i.e. maximum population size in standard use) and carrying capacity was higher this would then be multiplied with population size and subtracted to make fitness lower, which is counterintuitive as, I would have thought, overall population fitness should be higher if resource constraints are reduced i.e. carrying capacity is higher. So, more detail would be helpful here to help the reader figure out how the model does work.*

RC 43 *‘Thus, the fitness of a given species depends not only on how it interacts with other neighbouring species but with the rest of the environment as well.’ Please explain thus i.e. what specifically does this follow from?*

Please consider/explain the use of the term ‘environment’ for equation 3. This looks like it is a population size. How does this have anything to do with “environment” in the usual biological sense?

AR 43 In response to this comment raised by both the reviewers, we clarify that the individual-environment distinction here is guided by the partition identified by maximizing individuality. By putting a boundary around a subsystem, we separate the system into the individual and its environment. We clarify that this is an informational definition which does not include abiotic factors which are not modelled in this paper (see L# 295-298).

RC 44 *Please define the notation for the individuality measures at their use immediately below equation 3.*

AR 44 We have added a brief description of the individuality measures along with the definition of the notions (L# 304-308).

RC 45 *Line 47 distributions plural*

AR 45 We have fixed this and other grammatical errors in the revised manuscript.

RC 46 *Line 67 expected why? Because it is your hypothesis or because previous work has shown this result?*

AR 46 We thank the reviewer for pointing out our ignorance. It is expected in the context of the Tangled Nature model, as we have been working with the model for a long time. We have removed the term *as expected* and point the readers directly to the results.

RC 47 *Line 70 ‘quasi stable q-ESS and the transition regime’ need to be explained.*

AR 47 We thank the reviewer for pointing out that these terms were not introduced. The introduction of the model has been moved to the introduction now where we introduce q-ESS (L# 47). In this section of the results we have renamed the transition regime to *reconfiguration* regime to avoid the confusion with the transition region near the error threshold discussed later in the paper.

RC 48 *Figure 3. Please change the colour scheme so the line colours transition in a perceptible spectrum through the key values.*

AR 48 We have updated the colourmap of the plot indicating a transition from low mutation rate values to the error transition region. We also added a plot indicating the optimal scale according to individuality score for each mutation rate (see Figure 4).

RC 49 *There are a few grammar errors in the first paragraph of the results.* **AR 49** We have now used Grammarly to address the grammatical errors in the paper.

RC 50 *Line 160 in the discussion. This seems speculative. Can you investigate this more directly, or if not rephrase as a hypothesis for future work.*

AR 50 We have added a sentence that these interpretations drawn out of the paper need to be tested in more biologically focused settings.

RC 51 *I suggest removing the word ‘important’ from the conclusion (and elsewhere if used): this should be for the reader to decide, in context, from a clear statement of the aspects of novelty of the analysis and results.*

AR 51 We agree with the spirit of the reviewer’s comments and have removed the word *important* from the last paragraph of the discussion section.

Responses to comments from Reviewer 2

Summary: The authors use an existing explicit mechanistic model that simulates evolutionary systems which the authors believe has emergent “higher-order selection”. Their central claim is that they can retrieve the amount of selection induced by the parameters chosen at various units of organization using information theoretic tools. While it is not clear to me if solving the debate the authors are aiming at will be revolutionary for the field in some way, the topic and paper are definitely of at least theoretical interest, and the paper is interesting to read.

Overall: Generally, the paper is good, but the narrative around the paper is repeatedly unclear due to inconsistent or poorly explained language. Given the interdisciplinary nature of the paper, the definitions should be tight and the language clear and consistent without hoping the audience will be up-to-date on all of the papers cited. Almost all of my critiques are about the use of language, rather than any clearly biologically or mathematically flawed arguments. With that having been said, I cannot comment on the appropriateness and validity of any statistical analyses until Appendix C is clear and explicit, as all analyses in the paper hinge on it (see lines 226 - 227). Due to its influential nature, I am unsure why it is left as an appendix. As to whether I am convinced, I am not, but I think that may be due near exclusively to the miscommunications I’ve noted below.

RC 1 *I always find the linguistic usage of processes, like evolution or selection, and actions to be controversial with quite a few of my biologically minded colleagues. I don’t mind, e.g., the first sentence of the abstract, but I think some might be annoyed at the idea that selective pressure “acts”. Maybe “At what level (or perhaps scale?) does selective pressure apply?” The rest of the abstract agrees with my understanding of my colleagues’ usage, but some of the text wobbles back over. (E.g., “selective unit” is fine, but I don’t think they’d be pleased by saying individuals or genes are the “drivers of evolutionary change”.)*

AR 1 We appreciate these thoughtful suggestions. We have updated these terms throughout the paper considering the suggestions from both reviewers.

RC 2 *Lines 24 - 25 can be misread as indicating the authors are using a model that explicitly*

simulates higher-order selection, rather than that these properties are emergent.

AR 2 Many thanks for noticing this. We have revised the text, adding now the term *emergent higher-order selection* to clarify this misinterpretation.

RC 3 *Line 38: it isn't quite clear from this sentence what it means for "single species [to] act as information individual". More generally, I find this paragraph a bit hard to follow. I'm not sure if the same concepts are being referred to or if they are new each time because the language is not quite consistent. E.g., is "this higher-order phenomenon" "this cooperation", the "information individual", the "higher-order individuality", or the "error-transition threshold"? And in line 40, does "or the model" mean the authors conflate the analysis with the model, or are they not sure which does not assume a preferred level of selection, or do both not assume a preferred level of selection?*

AR 3 We thank the reviewer for raising this issue. To clarify the presentation, in our revised manuscript we have stuck to the notion of information individuality, while removing the reference to the model in relationship with the preferred level of selection to avoid potential confusion. Although, the model as defined does not impose any structural level of organization, we have highlighted this point later in the methods section (L# 63-67 and 312-315).

RC 4 *There are occasional distracting grammar configurations and mistakes, e.g., line 50: "Based on this fitness, species reproduces offsprings..." A general readthrough by the authors would probably pick up most of these.*

AR 4 We have taken the input from both reviewers and proofread the manuscript both manually and using Grammarly to correct the grammatical errors in the manuscript.

RC 5 *Is it intended that line 56's "generation 1" links to Figure 1?*

AR 5 Yes it is. That said, We have clarified this by explicitly mentioning the reference to Figure 1.

RC 6 *It would be useful to note whether there is some relationship to a temporal scale compared to the generational scale, as well as whether there is time resolution of the dynamics within each generation (by which I mean, are the populations dynamically adjusting through time or is the next steady-state calculated and the system assumed to reach it immediately). This is in the methods, but is not quite as explicit there as I would expect. It's easy to miss that the 10^5 generations are not 10^5 timesteps, for instance. It's also not immediately clear that "generations" is the right timescale, since it invokes the size of the population, which is affected by the mutation rate (Figure 2).*

AR 6 The reviewer is right to point out that since the system follows an asynchronous update rule, where an agent is selected uniformly at random at each timestep. We need to allow for sufficient updates to ensure every existing agent is updated at least once. This condition ensures that we compare populations at a steady state across mutation rates. This condition is widely used in monte-carlo simulations with asynchronous updates to generate stable configurations for different

system sizes. We have updated our description in the methods to explain why generational timestep used here is an appropriate timescale for the analysis (see L# 250 - 252).

RC 7 *Do the dynamics described in lines 56 - 61 constitute a phase transition?*

AR 7 Yes indeed, the error threshold discussed in the following section focuses on the transition where stable species exist across generational timesteps to a state where no stable species are no longer formed.

RC 8 *The legend in Figure 2c could be improved, as there are 3 vertical lines that could be interpreted at a glance as $p_{mut} = 0.042$.*

AR 8 We have replaced the boundaries of the error transition region with the shaded area in order to avoid confusion.

RC 9 *While visually the transition regions are obvious, how do the authors define it precisely? How long/stable do the states need to be to be q-ESS?*

AR 9 A stable state/q-ESS ends when there is a sudden change in the population configuration of the species. We use the measure of cross entropy to capture the similarity of population distributions between subsequent generations. We have added a section in the appendix to further explain this calculation, with links to the appendix in the main text.

RC 10 *It isn't obvious to me why the error threshold as defined should be interpreted as if-and-only-if the results shown in Figure 2. How do I know, for instance, that this isn't an artefact of the resolution or timescale the data is displayed at? The error threshold implies something akin to white noise, but it isn't clear to me if that is the sort of noise I am seeing at mutation rate 0.05, or if there might be some memory in the form of coloured noise. Could the authors explain?*

AR 10 Since the proportion of generations spent in the reorganization period becomes close to 1, beyond the error threshold. This implies that the population configuration changes significantly at every generation for mutation rates higher than the error threshold. Furthermore, looking at the entropy of population distribution reaches close to the maximum entropy at mutation rate of 0.05 (see figure below 1). These two data points point to the fact that the species between time points are uncorrelated (due to significant changes in population distribution) and randomly distributed (as population is equally distributed among existing species). Both these points indicate to the dynamics beyond error threshold. While exact value of error threshold depends upon the length of the genome, probability of killing and weight distribution (Di Collobiano et. al. 2003). The parameter sweep of populations provides a computationally approximate way of identifying the error threshold.

RC 11 *Maybe I have misread, but it appears that the authors refer to both $0.04 \leq p_{mut} \leq 0.05$ and the actual dynamics of the populations both as the transition regime, see caption of Figure 1 versus lines 72 - 73 or Figure 4.*

Figure 1: Average Entropy of population distributions in quasi stable states for different mutation rates in the Tangled Nature Model

AR 11 We thank the reviewer for identifying this error. We have now used separate terms for the two cases. The range of mutation between 0.04 and 0.05 is referred to as transition region and the generations between two stable q-ESS is called the renormalization regime, to avoid the confusion.

RC 12 *Figure 3: I think the main message is the mutation rates and scales, rather than the individuality scores. The colour scale doesn't help with this (e.g., two greens are visually adjacent but are 0.001 and 0.03). Maybe plot the scale at which the maximum is achieved against the mutation rate and move this to a supplement? The accompanying text is hard to follow as a result.*

AR 12 We have updated the Figure 4 with a colormap that transitions between low and high mutation rates. We have left out very low mutation rate $p_{mut} = 0.001$ to make this plot more visually accessible. We have added a plot with optimal scales as requested by the reviewer.

RC 13 *When and how is scale 6 identified as the optimal scale of organismal individuality?*

AR 13 We have rephrased this sentence. Scale 6 is optimal for some values of mutation rates but not all. We wish to represent a scale that exhibits higher individuality than scale 1 before the transition region. We have clarified the reason for this selection in the main text now. (L# 133-134)

RC 14 *Is normalisation based on the size of the group mathematically obvious? See 36 below for more queries about this at the end of the paper.*

AR 14 The mutual information between the current and future state of species/group of species is bounded by their joint entropy. Since we are calculating mutual information over values across an ensemble of simulations, these are normally distributed random variables. For Gaussian variables,

Figure 2: Variation of different modes of information processing at Scale 1 between the species and the environment. It can be seen that a valley is clearly visible in the transition region for the measures in the first row. However, it is not the case for the rest of the measures.

the Entropy scales with the number of dimensions of the system. Therefore, we use the size of the group as the normalizer. We have added details in supplementary material to explain the normalization and a reference to the appendix in the main text.

RC 15 *Are the heights of the individual curves in Figure 3 of any interest?*

AR 15 Yes, the heights average level of persistence of the groups at every scale . We have added this description in the presentation of the results (L# 112-114).

RC 16 *Is the valley in the transition region of Figure 4, scale 1 interesting?*

AR 16 We appreciate the reviewer's keen observation. We found the valley interesting as well, especially in other information measures we explored as part of this project. In some measures the peak was observed only on one of sides of the valley. In others, twin peaks are observed, thus prompting further exploration. We attach an image of some of these measures below 2 The presence of the valley is an interesting feature although requires further investigation. However, it is beyond the scope of the current paper which focuses on the emergence of higher-order interactions.

RC 17 *I'm not sure that "environment" or "ecosystem" are clearly defined in the text in reference to the model. I get the sense that the environment of a focal species is everything not of that species in this paper, but some ecologists exclude everything biotic from the environment. While I don't*

go quite that far, I do find the description uncomfortable here. Is ecosystem then the environment along with the focal species? This is defined in the methods, but without reference here.

AR 17 We agree with the reviewer that clarifying the usage of these terms would be helpful. As mentioned in our response to Reviewer 1. Under the information individuality framework, we are trying to identify an informational boundary within which a group of species persist cohesively. Once such a boundary is identified, the rest of the species act as the environment that affects the population of the individual. This definition of individual-environment partition is central to the information individuality framework (Krakauer et. al. 2020). Therefore, we have explained the notion of environment used in the paper in results (L# 130-132) and the methods sections (L# 295-298).

RC 18 *Line 114 "the information that a source variable about the future state of a target variable"? Information provided by? And presumably this means that previously the authors meant information at time $t-1$ predicting time $t+1$, whereas here they are interested in time t predicting time $t+1$? Additionally, which timescale is this t acting at?*

AR 18 We thank the reviewer for pointing out this error. We have now corrected the sentence to match with the mathematical definitions.

RC 19 *When the authors discuss information transfer, I find myself wondering what is "information" here. The language used suggests that it is something exchanged between species and environment, but the description suggests instead that it would be something an observer would make use of. Returning to earlier parts of the paper didn't make the subject clearer. This is critical for getting the right message across to the audience.*

AR 19 We have added a sentence explaining what TE is trying to measure in the case of co-evolving species and environment. In the granger causal sense, we can understand TE as a measure of statistical influence of one variable on the future of another. We have added this sentence to the results (L# 143 - 144). We have also enhanced the description of information measures and their applicability to ecological settings in the introduction (L# 30-33 and Figure 1).

RC 20 *Lines 124 - 125: does equivalent mean "mathematically equivalent", or "essentially the same as"? If the former, why bother with two names for the same thing for an audience who is likely to be unfamiliar with the subject?*

AR 20 Transfer entropy is a directed measure of statistical influence from one variable to another. Therefore, the measure enables us to quantify the influence of species on the environment and vice-versa. However, only the TE from environment to species, is mathematically equivalent to environment determined individuality. It gives us the opportunity to discuss other individuality measures discussed in Krakauer et. al. 2020, to provide a broader context to the higher order organization results presented in this paper.

RC 21 *Lines 124 - 129 provokes many questions. How much environment versus group? If the environment is predicting things about the group, should the group subsume more of the environ-*

ment, since these are mathematically, if not user, determined rather than biologically determined? Is scale 6 the correct scale to evaluate this at? How do the scales change with the environmental information provided? Does it mean anything that "higher-order organisation still possesses environment determined individuality in the error-transition region", since the authors have effectively argued that higher-order organisation is non-existent? Is there a necessity to control for noise here?

AR 21 Reviewer's comments gave us the opportunity to clarify certain aspects of the environment determined individuality. First, its contribution is significantly smaller as compared to organismal individuality scores. Therefore, the groups that show higher order organization are mostly driven by within group interactions. Environment provides a small (1/10th of the size of organismal individuality), but significant effect. Since these values are calculated over an ensemble of 10,000 simulations, the effects of noise are averaged out. Overall, the results show that the groups at scale 6 lose organismal individuality, yet have some marginal influence from the environment.

RC 22 *Figure 5: is the last panel necessary? I thought the text said that the curves in the last panel are equivalent to two of the curves above.*

AR 22 The last panel is presented to compare the values at two scales in region with low mutation rates and in the error transition region. This panel provides some continuity with figure 4 for organismal individuality scores. However, having explained these differences in the text we have removed the last panel from the figures.

RC 23 *Line 130 appears to lack all useful definitions. Lines 130 - 134 strike me as unfinished and unexplained for the reader.*

AR 23 We agree with the reviewer, more context was needed for the measure of integrated information. We have now added this context and rephrased the description of the results (L# 160-164).

RC 24 *The authors may want to reconsider the placement of the figures, as Figure 6 is mentioned on page 5 but is placed at the bottom of page 7.*

AR 24 We understand the reviewer's concerns regarding formatting of the images. Given the sizes of the figures, L^AT_EX allocates enough space between subsequent large images. This results in 1 image per page resulting in disconnect between text and images. However, this is not the final formatting and this issue will be addressed during the publication process.

RC 25 *The last paragraph before Section 3 is not clear to me and again provokes many questions. Where does, for instance, resource sharing occur in the model? What does robustness mean in this context? Are there mutations that happen at a level other than that of individual species? What do the authors mean by criticality? What is processing information in this context? What do adaptability and dynamic range mean in this context? What does "in order to maintain persistence" mean? Is the system actively resisting in some way the transition regime?*

AR 25 In line with the other changes made in this section, we have updated the last paragraph of

the results section to address the questions raised by the reviewer. We discuss the probable causes of increased information in different modes at the level of single species close to the error-transition. The relationship to critical point in phase transitions to enhanced information processing is also expanded upon. Finally we discuss the persistence of group level interactions for low mutation rate but as fluctuations increase, the variability at the single species level is leveraged for adaptation. We hope these changes will make the revised results section more accessible.

RC 26 *As written, this paper seems to really struggle with having the methods section last. Some reorganization of material might help here, but it might be satisfactory for the authors to better direct readers to the methods at opportune moments. As it is, the methods is referenced once, as if the reader should read the entire methods at that point. If that's the case, why put them last? Possibly similarly for the supplement, as I didn't see it mentioned in the text.*

AR 26 We have taken into consideration the reviewer's comment and highlighted the references to the methods section and the appendix in the results whenever new measures are introduced.

RC 27 *It might be beneficial for the authors to discuss, if not test the robustness of, other parameters. Is the transition governed solely by mutation rate, or is it modulated by the number of species or any of the other standard parameters? (And why are these parameters standard? Are they selected for biological plausibility?)*

AR 27 In previous work, Di Collobiano et. al. (2003), it was shown that the error-threshold depends upon the length of the genome L and the probability of killing a member of the random species p_{kill} . Varying these parameters can move the error-threshold and thus modify the transition region. The length of the genome L , might also effect the scale at which maximum individuality emerges, we plan on exploring this effect in future work. The standard parameters have been used to generate species-abundance curve comparable to the observed functional forms (Hall et. al. 2002).

RC 28 *"Therefore, we can understand the mutation rates in the TaNa model as a proxy to varying environmental selection pressure." And p_{mut} is also "the selection pressure introduced by the ecosystem" (lines 210 - 211). Given the lack of clarity regarding what is the environment/ecosystem in the system, I'd hesitate to have either sentence in the text. If the environment includes species other than a focal species, then shouldn't environmental selection pressure already implicitly be included in the model via the interactions between species?*

AR 28 We thank the reviewer for noticing this disparity. Having defined the environment in the species-environment interaction, it is imperative to clarify the usage in this sentence. We have added some text to clearly describe the mutation rates to be related to the abiotic factors like temperature and weather changes. Rather than the interactions with the biotic environment which is modelled explicitly in the TaNa model.

RC 29 *I think in order to "sell" the message, the authors need to present examples to help frame the idea. Jumping away from TaNa for a moment, if we had evolution in a classic Lotka-Volterra, I'd hesitate to call the predator and prey a "group of cooperating species acting as an effective unit*

within an evolutionary process” (to borrow words from line 154). The authors could circumvent that critique with some clarity at the beginning regarding the model and what a group means. It also might help to present (examples of) what groups emerge in the model.

AR 29 Inspired by the comments of both the reviewers, we have done an interaction structure analysis and simulated a neutral model where all interactions among the species are constant and equal. We have added Appendix E and refer to the findings in the main text (L# 119-125). We find that a hierarchical, asymmetric interaction structure is observed in low mutation rates at scale 6, which is absent in the transition region. Finally, when individuality scores are recalculated for the neutral model, no higher-order organization is observed above scale 2. These findings highlight the role of sparse asymmetric interaction structure in supporting higher-order organization.

RC 30 *Line 218: “Thus, varying the mutation rate provides two interesting transition points...” I don’t see the first “interesting transition point”. Instead, the first regime seems fairly continuous before not-instantaneously switching to a different regime.*

AR 30 We agree with the reviewers assessment and have modified the text to focus on the error transition as the main phase transition in the system.

RC 31 *Is there a reference justifying the contents of Appendix C? I can’t tell if I should be very worried or not at all worried from it. Either way, it needs another go from the authors. I’d especially appreciate a reference regarding the timescales for the sampling if possible, i.e., that the sampling time scale shouldn’t be related to the mutation rate. (That’d actually be a nice thing to examine if possible: what happens to the various metrics if you compare before and after some large number of mutations.)*

AR 31 As mentioned before, we have added further explanation for using the ensemble of simulations to estimate the information measures at each time point instead. This framework allows for tracking information measures across time and multiple simulations in the ensemble provide stable probability distributions to estimate information measures. We have provided a reference to the neuroimaging studies where this is used (L# 283-289).

We have also provided an explanation that the generational timestep is the average time taken to update every agent of the ecosystem at least once. This sampling time is commonly used in monte-carlo simulations to ensure the system has equilibrated before recording its state at a given time (L# 250-252).

RC 32 *I’m not sure why, but a few bits are not line numbered correctly.*

AR 32 We have recompiled the manuscript and visually scanned the revised manuscript to check for errors in the line-numbers.

RC 33 *Hamming distance is named after Richard Hamming.*

AR 33 We thank the reviewer for bringing this to our attention. We have corrected the usage of the measure as *Hamming distance*, with first letter capitalization for proper nouns.

RC 34 *The authors seem to argue that their biological model contains units predicting their future (lines 232 - 235). Could the authors elaborate on how that is the case in the Tangled Nature model?*

AR 34 We have clarified this point in our previous comments describing the notion of individuality as persistence. Prediction here means, the ability to predict future population from current state. The enhanced explanation of information measures and especially the individuality scores will address the concerns raised by both the reviewers on this matter.

RC 35 *The authors should explain their notation in section 4.2.1.*

AR 35 We have added the explanations for the notations of individuality measures, as pointed out by both the reviewers (L# 304-308).

RC 36 *From line 241 to eq 4, could the authors be more specific? How is the random sample taking place? How many different subsets? Which bias is introduced by increasing dimensions?*

AR 36 We have added more detail on how we sample the subsets in order to ensure that species which are most dominant in the ecosystem are included first in the 10^5 subsets sampled (L# 312-314). We have added more details about the normalization in Appendix D, regarding the bias introduced by increased entropy at higher dimensions.

Responses to comments from reviewers of paper

Manuscript ID *COMMSBIO-24-3608-T*

“Information dynamics and the emergence of high-order individuality in ecosystems”

RC 1 *Reviewer’s Comment*

AR 1 Authors’ Response

We thank the editor and the reviewers for their thoughtful work in reviewing our paper. The comments provided have helped us to substantially improve the revised manuscript. To aid the review process, the modified text has been highlighted using colour blue, in the tracked changes version of the manuscript. In this response, we refer to these changes using line numbers (in short L#), on the left margin of the revised manuscript.

On behalf of all co-authors, I thank you for your attention.

Sincerely,
Hardik Rajpal

Responses to comments from Reviewer 2

RC 1 *Thank you to the authors for their work on this revision. I appreciate in particular some of the rearrangement of material the authors have done, which seemed to do a better job of introducing the results, and the text is generally more accessible I believe.*

Below, I’ve tried to arrange my comments to correspond to the ongoing conversations as well as the text order where possible, but I appreciate that I have added additional comments as well.

Overall Concerns: Unfortunately, I do not feel my queries and criticisms of the analysis, especially regarding Appendix C, have been fully dealt with at this time and I cannot yet recommend publication. I must ask the authors to make another attempt at revising the text. Details are below.

AR 1 We thank the reviewer for diligently assessing our revisions. Their comments have been beneficial in the first revision. We understand that the reviewer still has concerns about the analysis and has raised additional comments. In this revision, we have added more clarity to the analysis conducted, including the link to the code used for the analysis. We have attempted to address the additional comments in the timeframe provided for the revision. Where we could not implement the specific analysis requested by the reviewer, we have provided alternate arguments and changes.

RC 2 *I also think I agree with R1 that the understanding of "mutation" and "species" in this model might have significant impacts on interpretation. Reproduction (clonally) is clear enough, but if mutation were replaced with speciation or diversification, I can't help but think that the manuscript might receive a different reception. This has repercussions for my understanding of the text throughout I believe and especially near the end of the text, as well as whether R1 and I are understanding the same things.*

AR 2 We have taken the inputs from both reviewers seriously regarding this matter, and we believe that the previous revision clarified the reviewer's comment in the definition of the paper. In the definition, we refer to the binary vector as the pangenome of the species; any reproduction event that leads to a change in the pangenome can certainly be understood as a speciation event (L# 47-49). However, since the present study focuses on interdependencies between the species, only clonal reproduction is considered. Although minimal, we believe that the discussed version of the model can simulate the higher-order interactions among the species, which is the focus of the current study. We hope that this clarifies the concerns of the reviewer, and we hope that both reviewers understand the assumptions behind the model.

RC 3 *Additionally, I realised in hindsight that I have not seen the code associated with *the analyses* conducted here, although the authors do point to a Github containing Rust code *for the simulations*. I would like to see that in the next round of revisions, as many of my methodological questions could have been answered from looking there. My apologies if I missed the analyses in the Rust code.*

AR 3 As we use the JIDT open-source library for information-theoretic calculations, we have not shared a particular implementation of the measures on the simulation data. However, at the reviewer's request, we have put together a GitHub repository, which shows our implementation of the JIDT toolbox, along with a subset of the simulation data to test the scripts on. Hopefully, this adds to the clarity of the implementation.

RC 4 *Presentation Concerns: One emergent and overarching question that I had on this read-through is whether the authors are essentially measuring the probability that subsets of a system will change state between one generation and the next. If so, this might be a more accessible framing.*

AR 4 Yes, the reviewer is right to understand mutual information as a property of the joint probability distribution of the variables. In our implementation, since the variables are the population of subsets of species in consecutive timesteps, the individuality scores basically track how the joint probability distribution of a subset of species evolves over time. Following the reviewer's suggestion, we have added this interpretation to the methods section to aid interpretability (L# 322-323).

RC 5 *(RC 4) I note that Grammarly does not appear to have been a panacea for some errors, e.g. Appendix C has "tangled nature" and its second sentence reads as incomplete (it should probably be combined with the first sentence). A similar issue was near immediately apparent when I went back to the main text (lines 11 - 12). I surmise the manuscript would benefit from another careful, manual grammar check as there were plenty of similar errors still.*

AR 5 We thank the reviewer for bringing this to our attention. We had an issue with the Grammarly plugin and noticed more grammatical errors than were detected during our previous revision. We thank the reviewer for their patience while reviewing the manuscript. We reviewed the manuscript again with Grammarly and manually proofread the text (highlighted in blue). Hopefully, all grammatical errors are now corrected.

RC 6 *I also note that, while adding color to the mathematics is an interesting idea and sometimes done in the classroom, I'm not keen on it for a manuscript where it might present an accessibility issue (e.g. if the reader tries to read on an alternative background color). This is a minor comment, but worth keeping in mind.*

AR 6 We have partially addressed the reviewer's concern regarding accessibility by adding coloured descriptor text below an underbrace in the different parts of the equation, while leaving the equation in black. We have also minimized the coloured text in the description below to refer to the key parameters as suggested by Reviewer 1 (L# 272 - 274).

RC 7 *Comments: (RC 34) Abstract: "where single species gain higher information-processing abilities". If I've understood correctly, species do not have information processing abilities in this model. Instead, an observer of the model receives more or less information about the species as groups in this model. This is supported by lines 35 - 36 and seems a sensible interpretation of Figure 1 and AR 34. If I am correct, the abstract should be changed.*

AR 7 We thank the reviewer for pointing out this crucial linguistic difference. We have modified the relevant lines of the abstract to highlight that we mean the information processing observed in the dynamics rather than the information processed by species. We hope that the rephrased abstract moves away from the confusion.

RC 8 *(RC2 1) Related to this; Lines 45 - 46: While this is definitely true from my memory of the Tangled Nature model, I do wonder if it might be more persuasive if the authors mention beforehand that species are single genotype/phenotype, so any mutation results in a technically different species. This ties in to R1's RC 36. It might also be worth commenting how large a difference "a mutation" makes. As I recall, there is not a sense of a clade in the Tangled Nature model. (Indeed, it might be more appropriate to think less of mutation and more of a random but biased replacement of an individual of a species with a member from the pool of species, as the traits seem to be able to completely vary between species in the authors' version of the model, lines 249 and 240 - 242.)*

AR 8 Reviewer has identified an important place in the introduction where the pangenome/ mutation/ speciation dynamics of the model can be introduced to the reader early on. We have added an explanation as suggested by the reviewer (see L# 47-48). This added pointer in the introduction, along with the previous revisions in the methods, will clarify the peculiar modelling choices of the Tangled Nature Model as compared to other models of evolution.

RC 9 *(RC2 2) And, on the subject of mutations, instead of "stable species", it may be more useful to say "stable populations of species" throughout. The former might be interpreted as being about the evolution of a species, which seems confusing in this context, while the latter seems more*

clearly about whether a population or a set of populations persists.

AR 9 We agree with the reviewer that "stable population of species" is a better phrasing in the context of the study. Therefore, we have replaced this in the revised manuscript.

RC 10 *(RC2 3, but also 29) The description of Figure 1 in the text made me expect a model to be solved (lines 52 - 53). The authors may want to instead say the figure is schematic in nature and be more specific about "these measures" since, e.g., it is not immediately obvious if "individuality" is one "of these measures" in Figure 1.*

(RC2 4) Also, middle and bottom panels do not make sense for Figure 1 as in the text. I'm also mildly confused as to why the influence of one species over the other is called "statistical". Predictability would seemingly make more sense, although I suspect this is a disciplinary difference and the authors may need to make decisions regarding the target audience.

AR 10 We thank the reviewer for pointing this out. We understand how the current framing of the sentence can be misleading. Therefore, we have revised the sentence referring to Figure 1 (L# 55-56). We have corrected the labels of the panels in the caption. We have also rephrased the description of information transfer to clarify it to a broad interdisciplinary readership.

RC 11 *(RC2 5) Lines 73 - 75: I wonder if another comparison might be more useful here, as the subset of all feasible combinations is likely to be quite smaller than the space of all combinations. I much prefer thinking of it as a continuous time Markov chain defined over the assembly graph myself, but that might be another discipline too many for the audience. (I suggest this because it is now even less clear to me what a "collective" random walk means in this context!)*

AR 11 We have now rephrased the description of the reorganization region as suggested by the reviewer. We now focus on how the reorganization is triggered and a new subset of species is explored. We also provide a reference to the figure in Becker and Sibani 2014 that clearly explains this dynamic (L# 76-78).

RC 12 *(RC 6, 31) Regarding Appendix C, as this is an ecology facing article in a biology journal, it might be wise to expand on the technique beyond that it is standard in neuroscience. The authors might also consider that they've directed readers to a 33 page reference in an additional different discipline, and this reference does not use the same terminology as in the manuscript (ensemble and stable were the obvious words to me to look for in the reference, while copula is not mentioned in the manuscript). Minimally, it would be good to establish where in that reference I or the readers should look.*

AR 12 We take the reviewer's concern about the novelty of the method to biology readers seriously. We have updated Appendix C and added a figure that illustrates the Ensemble MI estimation to clearly explain how information measures can be estimated at each timestep across an ensemble of simulations. Upon the reviewer's suggestion, we also looked for a paper by Gomez et al. that discusses the ensemble approach in detail. We have added the reference to the page and section in Ince et al. for readers interested in this approach.

Figure 1: KL divergence of the total population distribution across the ensemble after three different generational timesteps (1000, 50000 and 100000 generations). It can be seen that even for a small number of simulations sampled from the simulated 10,000 simulations, low values of KL divergence are observed, which converge to zero as the number of simulations is increased.

RC 13 *Given the flexibility in the word "simulation" in the literature, it is not immediately obvious to me that the ensemble of simulations should converge in the limit to a stable (sensu?) and stationary normal distribution, nor is it immediately obvious that we have converged to that limit should it exist.*

AR 13 In the context of an ensemble of simulations for a Markov process, with different initial conditions but implementing the same dynamics, similarities are expected in rank-matched species across simulations. However, given the interdependent non-stationary nature of the dynamics, we understand that these factors can affect convergence. Therefore, we show the convergence for varying numbers of simulations for two mutation rates. We compare the Kullback-Leibler divergence between the distribution of the total population of the system for a given number of simulations in an ensemble k with the population size distribution of the full ensemble of 10,000 simulations (see Figure 1) It can be seen that the K-L divergence, which estimates the distance between the two distributions, converges. Note that KL divergence values are very small to begin with, for a number of simulations, ≈ 1000 , the divergence drops below 0.01, indicating almost identical probability distributions. Furthermore, convergence is observed in all generational time steps: 1000, 50000, and 100000.

RC 14 *How do we know that the 1000th generation is appropriately well-mixed? It seems likely from Figure 2 that there may not have been a transition regime for low mutation rate models, in which case the system seems to remember its initial state, and that's neglecting that transitions between stable species populations are likely not at random from all possible stable species populations.*

AR 14 As we have explained the MI estimation procedure in detail across the ensemble of simulations, we look at the change in probability distribution of species for the consecutive generations. This MI can be estimated at each timestep, including at the 1000th generation, where we have shown that the probability distributions across 10,000 simulations converge to a stable distribution (see Figure 1). Therefore, the ensemble method can estimate MI values after every 1000th generation. We present the MI values for the two distinct mutation rates, 0.01 and 0.045, which are estimated at every 1000th generation in time from 1000 - 100,000 generations in Figure 2 below.

Figure 2: Stable values of Individuality scores is observed at both scale 1 and 6 across time using the ensemble method.

It can be seen that these values are quite stable over time. When presenting the averages in the individuality plot, we disregard the first few MI values to eliminate the effect of any transience. We hope that this answers the concerns of the reviewer regarding estimating MI values and their stability over time.

RC 15 *How do we know that the "individual generation" time chosen by the authors is the right generation time (recall: definitions in this manuscript are somewhat murky to begin with), as opposed to measuring via "species generation" time? Comparing with the Markov chain Monte Carlo literature, I'm looking for something analogous to convergence diagnostics.*

AR 15 We would like to clarify to the reviewer that we have not made an ad-hoc choice regarding the generation time in this study. The definition of the generation time used here $T_{gen} = N(t)/p_{kill}$ was defined in the original 2002 paper of the Tangled Nature model (Christensen et. al. 2002) and every study of the Tangled Nature Model has used the same generation time, as it is the average number of timesteps needed to randomly kill off the existing population. When asynchronously updating parts of the system using a Metropolis-Hastings algorithm, the time to update each part of the system is usually taken as the timestep at which the state of the system is recorded.

Since the dynamics of systems with lower mutation rates are much slower, the state of the system should be recorded after many time steps. This is reflected in a longer generation time as $N(t)$ is much higher for lower mutation rates (see Figure 3). On the other hand, for high mutation rates, the dynamics is much faster and the state of the system is recorded at much fewer timesteps. This definition of generation times has been used to explore time-dependent extinction rates, species abundance, waiting times between extinction events, and other ecological temporal properties. Therefore, we believe the generation time defined in the original 2002 paper is a reasonable choice for our analysis.

Could the reviewer explain what they mean by species generation times? The generation time here is the time after which the state of the system is recorded. Is the reviewer suggesting that we record the state of different species after different numbers of updates? We do not quite see how that would be meaningful, and it would be impossible to do any temporal analysis that considers interactions among the species. Since the Tangled Nature Model is an asynchronously updating stochastic

model, only one existing agent is updated at random at each timestep. Therefore, a sufficient number of updates need to happen before the state of the system is recorded. For conducting any convergence diagnostics, it can be done after a sufficient number of updates have been made. Our previous analysis of KL divergence shows that for the chosen generation timesteps, the population distribution across independent simulations converges (Figure 1), showing that it is a meaningful timescale to observe the system.

Figure 3: Average population vs mutation rate

RC 16 (RC 10, 19) *Where related to RC 6 (above) I am not satisfied, but I understand the authors argument for the latter half of my comment. I suspect that line 81 should probably be changed to better capture their response though. Maybe "which is the limit on the mutation rate beyond which one generation is unable to be used to predict the next". This removes awkwardness around "biological information needed for continual survival". If the authors do not think it would be distracting, they might consider adding their figure into Figure 3.*

AR 16 We have taken into account the reviewer’s suggestion and changed the description of the error threshold in the manuscript (L#85-85). Now we focus only on the predictability of the future population of species. This description is in line with the theme of the paper, and bringing in the biological relevance does not enhance the interpretability of the paper. Therefore, we take the reviewer’s suggestion and stick to the scope of the error threshold relevant to the paper. We also added the entropy results to Figure 3 to provide the complete picture of the species distributions close to the error threshold.

RC 17 (RC2 6) *Line 94: is "organismal" necessary in describing the individuality scores? I understand from 4.2.1 that it is a specific type of individuality, but the way it is introduced in the text does not make that clear and it seems like "organismal" might cause more confusion than it prevents when viewed in the context of the unclear biological scale. As it is, it is somewhat unclear when using individual, group, and species in section 2.2, especially as individuals in this section refers to (groups of) species rather than agents, which are also in the model. Treating this explicitly here would help (a la line 131). Perhaps "different group sizes of species" -i "groups of different*

numbers of species” might also help a little.

AR 17 We agree that introducing the type of individuality score doesn’t help the reader here. As detailed definitions are provided later in the methods, we have removed the word ”organismal” from our initial presentation of the results. Furthermore, we have also added the clarification that groups refer to groups of different species.

RC 18 (RC2 7) *Line 97: it would be clearer to use explicit repetition, i.e. ”knowledge of the group’s past population”.*

AR 18 We have now added explicitly that the predictability refers to both the past and the future populations of the groups of species (L#97 - 100).

RC 19 (RC 12) *Figure 4: I’d like to see a mention of the number of simulations averaged over in each of the mutation rate lines. Additionally, I’m happy with the blues and orange colors, but I’m having a hard time telling the reds apart, esp. 0.04 and 0.042.*

AR 19 As discussed in the response to previous comment # 14, all the individuality scores are estimated over 10,000 simulations for each timestep and the averages reported are the temporal averages over generations. Individuality scores from the first few generations are discarded to eliminate any transient effects (see Figure 2). We realise the language in the caption was unclear about this, so we have now rephrased it. We have also fixed the colourscale to differentiate 0.04, 0.042 and 0.045 mutation rates.

RC 20 (RC2 8) *Line 102: ”facilitating” seems an odd word choice. Perhaps ”modulating” or ”regulating” might be more appropriate.*

AR 20 We agree with the reviewer and have now replaced ”facilitating” with ”modulating” (L#106).

RC 21 (RC 13) *Line 104: Why do the authors list the range of peaks as between 5 and 9? There don’t seem to be obvious peaks beyond 7. Are they referring to additional unplotsed data?*

On a similar note, lines 133 - 134 and Figure 4 don’t seem to be consistent. Ignoring the spurious tick marks on the x axis (4b), it would seem that 5 was just as commonly maximal, and possibly over a much broader range. Further (4a) would suggest that 4 - 7 are near equally important for low mutation rate regime. The authors may want to adjust their explanation on lines 133 - 134, as I’m bothered more by the inconsistency than by the scale of analysis.

(RC 14) *Line 105: ”higher-order organisation still persists”. The authors appear to be interpreting the individuality scores here, but it isn’t clear from the main text what a high or low score means except in a vague and relative sense. Indeed, it’s not clear in a practical sense what it means to measure ”the total information shared” (line 304), even if the authors place it on a relative scale (line 318), although they do not place it on an obvious relative scale (Appendix D and Figure E.4, and responses to RC 6 and 31 above)). Could the authors show a practical example of what the individuality scores mean somewhere? I appreciate that this is likely trivial for the authors, but it*

Figure 4: Comparison of Individuality Scores at Scale 1 with scales 4-7.

is an interdisciplinary communication problem.

AR 21 We apologize for the unclear explanation of the results. We have now rewritten the second paragraph of the results section 2.2. In the revised paragraph, we clearly state that, in Figure 4, the individuality scores are to be interpreted relatively across the scales for a given mutation rate (L# 106-109). Increased individuality score at higher scales as compared to scale 1, is interpreted as the group of species interacting together to enhance their persistence. The ability of the groups of species to leverage the networked interaction structure is what we call the higher-order organisation. In this sense, for comparison with scale 1, scales 4-7 are equally as important. It can be seen in Figure 4 that although there are slight differences in absolute values of individuality scores of scales 4-7, they all follow the same declining trend as the mutation rates increase. They all cross over with Scale 1 at the beginning of the transition region at $p_{mut} = 0.04$. While Scale 1 shows a distinct behaviour to all higher scales and peaks in the transition region. Figure 5 in the manuscript focuses on highlighting this diminishing of higher-order organization in the transition region. Therefore, scale 6 was selected for the simplicity of comparison. We hope that this clarifies the confusion regarding how to interpret these results.

RC 22 (RC2 9) Section 2.3 would benefit by reminding readers of definitions of information transfer (species-cross predictability?) and integration (species-mutual predictability?). Figure 1 did show its value here though!

AR 22 We have now added a brief description of the measures (L# 138-140) and have pointed the readers back to Figure 1 in the beginning of section 2.3 to enhance readability. We thank the reviewer for identifying the gap, as these measures were introduced during the introduction but are applied only at the end of the results section.

RC 23 (RC 29) Upon re-examining Figure 4, I agree with the authors that examining a neutral model is an important baseline to compare with. Unfortunately, I don't think the authors have used it as such (e.g. the markedly different scales and intent of Appendix E).

Appendix E: I was a bit surprised at the lack of detail of the neutral model. What is the reasoning

for fixed positive only interactions among species? Mutualism is traditionally not well regarded for ecosystem stability arguments (e.g. the introduction of "The stability of mutualism" by Lewi Stone (2020) or the "orgy of mutual benefaction" as oft quoted of Robert May, but I believe there have been many recent works which prevent the instability while incorporating mutualism such as "[B]alance of interaction types..." by Qian and Akçay (2020)), so it doesn't seem surprising that there'd be a lack of persistence, which is what the authors seem to be measuring information about. In this case, it seems less a neutral model and more the not-quite most obvious, negative extreme. (Competition would be a more obvious negative extreme.) Even then, it looks like some of the scales might be higher than that encountered in Figure 4a, although it is hard to tell even side-by-side.

AR 23 We would like to clarify the choice of the neutral model in order to answer the reviewer's concerns. In Figure E.4 it can be seen that from a sparse network of positive and negative interactions, the system seems to be selecting for a particular balance between the two. To test whether this specific interaction structure was the reason behind the emergence of higher-order organisation that exhibits higher information individuality scores as compared to the scale of single species, we wanted to simulate the dynamics on a model where no such structure exists. By fixing the interaction network as a fully connected network with equal weight, we ensure that the system always ends up in a trivial interaction structure. Appendix E shows that in such a system where balanced positive and negative interactions do not exist, organisation at higher scales is not observed. We hope that the reviewer appreciates why we chose a fully connected network with weight +1 as our neutral model. The scale of the values of individuality scores depends on the degree of correlation among species. When we introduce a fully connected network we increase the degree of correlations, therefore, the scale of the values in the neutral model is higher. Therefore, we only use the neutral model to see if a higher-order organisation exists, i.e., do higher scales have a relatively higher individuality score than scale 1? We find that scales higher than two do not show increased individuality scores as compared to the original TaNa model with sparse positive and negative interactions.

Now we agree that having a network with only negative interactions is also a possible neutral model. However, we wanted to highlight the role of balanced sub-structures, therefore we used the above mentioned neutral model. In the case of negative-only interactions, the TaNa model has two attractor states. In a dense interaction network with only negative weights, all existing species will die out because inter-species interactions will reduce the fitness of all species. However, in a sparse interaction network, the species can exist in different parts of the interaction structure, uncorrelated with each other. In this uncorrelated case, the normalized individuality score will remain constant with increasing scales.

RC 24 *Figure E.3: I'm a bit perplexed. My understanding of the model is that the interaction matrix is initially fixed, and individuals effectively choose a species identity when born, biased by their parents. If the same six species (or are the species numbers spurious?) and same interaction matrix are used every time, how would their network structure change? Similarly, line 200, is it the *observed* interactions become more mutualistic over time, or is the interaction matrix as a whole changing?*

AR 24 We thank the reviewer for bringing up this misunderstanding. Perhaps, we didn't clarify what we meant by the observed interaction matrix. The reviewer is correct in observing that the interactions between all possible species are fixed in the beginning. However, in any given

simulation, only a subset of all possible species exists (see Figure 3). Therefore, we discuss the interaction structure of the existing species in Appendix E. We had stated (previously in line 200) that *interactions within the existing species become more mutualistic over time*. We have now explicitly added this description of the interaction matrix and its dynamics to Appendix E to address this concern.

RC 25 (RC2 9) Line 160, Φ_R should likely have a citation at this juncture.

AR 25 We thank the reviewer for pointing out this oversight. We have now added the relevant references (L#171) and have pointed the readers to the methods section, where the measure is described in detail (L# 175).

RC 26 (RC 7) Line 168, *I'm not sure how much emphasis the authors want to place on the phase transition. I imagine from their response that it is already known in the literature and so it is fine to acknowledge and remark on, but I don't think for the current audience that it is a particularly biologically interesting conclusion. Indeed, the transition region itself might be of interest, as is the stable region, but the unstable region of breakdown I'd be surprised to hear of a biological analogue for.*

AR 26 The relationship between information processing and phase transition is that the literature mostly focuses on physical systems such as the Ising Model, Random Boolean Networks, etc. We find it interesting that it extends to the co-evolutionary dynamics presented here. In the biological context, the increased information-processing at the level of the species can therefore serve as an indicator of the system moving towards the error threshold. However, we see that this inference from the model is a hypothesis whose generality needs to be tested. Therefore, we have now limited the discussion of phase transitions to a remark in the revised summary of the results. (see L# 184-185)

RC 27 (RC 21, 20 below) *I think the primary problem in Figure 6 now is I have no idea if these effect sizes are of importance (theoretical or practical), and the paper hasn't given me the tools to understand it. E.g., are the individuality scores (range of 0 to 2?) measured on the same scale as the entropies (range 0 - 0.08), as suggested by lines 156 - 157 and AR 21. In which case Figure 6 is saying about 10% of the "predictability" of the system is from the environment pre-transition but jumps to 25% for scale 6 during the transition (but, given we're dealing with information, it isn't immediately obvious how this maps to something like a time series forecast(pseudo-)R²).*

AR 27 We regret that even after describing the measures in detail and providing the mathematical description of the measures, the interpretability of the measures is unclear. We want to take this opportunity to clarify the measures once again to the reviewer. The entropy of a random variable X , can be understood as the measure of total variability or total information provided by the variable. All or part of this variability can be explained by the knowledge of other variables, this amount of information is quantified using mutual information between the variables. In the context of time-series data, the other variable can be the past of X itself or any other variable Y . Now, the excess information provided by the past of Y over what is already known by knowing the past of X , is what we define as transfer entropy (Methods 4.2.2). This is the extra predictability or

amount of information provided by one variable (species or environment in our case), about the other. Figure 6 presents this measure of the statistical influence of one variable over the other for different values of mutation rates. Yes, the figure does highlight that the information provided by the species/environment about the other increases during the transition region, which we have discussed in detail in the results and the discussion.

Now, information theory provides the tools to quantify the information shared between the two variables; it is a model-free approach that provides a statistical measure of information based on the joint probability distributions of the two variables. Now, in order to exploit this information, one can train a regression model to predict one variable from the other and even be used for feature selection (see Frenay et al. 2013 for a detailed discussion). However, here we discuss information measures as a statistical quantification of information one variable has about the other. We hope that this description clarifies the concerns of the reviewer.

RC 28 *(RC 20, 22) It appears that the authors may have forgotten to remove this panel? If not, I'm not sure if the third panel with the new name is necessary if it is (mathematically) equivalent a subset of the above two panels. That's introducing a new term in an already jargon heavy paper for this audience for seemingly no pay-off. If so, if continuity was the goal, perhaps the authors might consider combining Figures 5 and 6.*

AR 28 On reviewer's suggestion, we have now removed the third panel. We agree with the reviewer that this improves accessibility and the text in the manuscript is sufficient to explain the parallel between Transfer Entropy and Environmental Determined Individuality.

RC 29 *(RC 24) I am familiar with LaTeX and how it can be customised to place figures in the text. In particular, it does not need to guarantee 1 image per page if the authors do not want it to, and it can make the job of reviewing easier if the figures are located around where they are mentioned in text. Blaming LaTeX is in some sense abrogating responsibility, although I can understand not wanting to overly fiddle when it won't be the final format.*

AR 29 We apologise to the reviewer for not being able to accommodate the formatting suggestion. We agree that ideally, all figures should be placed right next to the text where they are referred to. However, we wanted to balance the location of the figures without changing the submission format too much. It was our decision, and we take responsibility for presenting the manuscript. We sincerely apologise again for the inconvenience.

RC 30 *(RC 25) I'm a bit confused by the state of the paragraph lines 169 - 183, especially now that I think I have a better grasp on the manuscript due to improvements elsewhere. It seems to pretty commonly be at odds with itself, but this might be imprecision in phrasing. The authors appear to be saying that*

1) whether it is easier to predict for a group of species or a single species changes depending on the mutation rate in non-linear ways. (Sure.)

2) it is easier ("enhanced information processing [by the observer]") to predict in the transition region (but this seems to conflict with Figures 5 and 6, suggesting that the authors mean only for

individual species).

3) that the system itself responds to information ("access[es] many different modes of information processing", but the system does not have agents that process information, so this appears to be a misleading phrasing).

4) the system changes more often in the transition region ("increases entropy" and thus decreases predictability, but this seems to be in conflict with 2) above).

*5) "the system [finds more] stable group-level interactions that *adapt* [at lower mutation rates]" (but if the group is adapting, the system wouldn't be judged to be in a stable community, and it is unclear if systems are "more stable" when found in 0.001 than in 0.04 mutation rates. One would presumably need to look at the probability of exiting the population configuration as a function of the mutation rate and show that population configurations found at low mutation rates have curves everywhere lower than those found at 0.04, or something similar, especially as the mutation rate obviously affects some notions of stability. The comment becomes sort of trivial (of course systems at higher mutation rates are less stable) or completely non-obvious (systems found at higher mutation rates but not found in lower mutation rates may be less stable or may just be harder to observe in general)).*

6) "[t]he system then leverages the enhanced variability... of single species to evolve towards metastable states" (but the system doesn't arrive at stable communities when the mutation rate is too high, nor is it clear how the system "evolves" in the biological sense when it is just alternating through species seemingly at random).

Also, above, I said "changes more often", but the authors might wish to remind/show the audience that (it appears that) there is truly more frequent switching between stable community configurations as well as switching taking longer, judging from Figure 2.

AR 30 We thank the reviewer for collating the issues related to the results summary, which were unclear. In the revised version of the manuscript, we have tried to clarify the results presented and address the concerns raised by the reviewer. Here, we have provided point-specific responses below,

1 and 2) Yes, the individuality scores presented in the paper quantify how much information a single species or a group of species provide about their future. Therefore, this measure is an important metric for assessing group/single-species level stability at any given mutation rate of the system. We specify the species' scale in the revised summary when discussing different information measures. The reviewer is right to note that, in the transition region, information measures are enhanced at the level of a single species, and we have now clarified that (L# 188-194).

3) We have taken the reviewer's comments seriously when discussing information shared/processed among the parts of the system and moved away from talking about the "system" to be processing information. We hope that the revised summary clarifies this language.

4 and 5) We understand the conflict the reviewer is referring to comes from the imprecise description of the scale at which the dynamics of the system are talked about. As mutation rates increase, the global coordinated dynamics of the system break down due to increased noise (as identified by the reviewer). The decrease in individuality score among groups of species is indicative of this trend.

However, at the level of a single species, the increased mutation rate (in the transition region) leads to more predictability. Had the dynamics of the system been completely trivial, increasing the mutation rate would have led to a decrease in predictability at all scales due to increased noise. However, we do not observe this monotonic decrease at the level of a single species. We hope this clarifies the conflict that the reviewer is referring to.

6) Mutations in the system are triggered only when a given species reproduces. The probability of reproduction depends on the species' fitness (its interactions with other species). In a state with high mutation rates, yes, the system takes longer to find groups of species that can coevolve together and stay in these configurations for a shorter duration of time because of the increased rate of mutations. So, the reviewer is right in noting the decreased stability of the joint configurations. Until the error threshold is reached, the mutations and reproduction of a given species are linked to its interactions and do not happen completely at random. Beyond the error threshold, there is no role of interactions left in the dynamics, and the switching is entirely random. As suggested by the reviewer, we have pointed the readers to Figure 2 to highlight the shorter metastable states during the transition region.

RC 31 (RC 28) Lines 206 - 208: *I'm not sure the authors want to connect too closely to harsher environmental conditions, since that would make mutation rates a species level response to what I would expect to be a rise in p_{kill} , which is not at all what the model does.*

AR 31 Reviewer raises an excellent point. In this study, we have not analyzed the effect of p_{kill} on the co-evolution of species. We have mentioned it as a limitation in the discussion section, and that future work needs to be done to study the impact of adversarial environment conditions more directly (L# 219-221).

RC 32 (RC2 10) Lines 210 - 212: *Again, if I've understood the model correctly in that species are not changing their traits through time, I'm not sure that "individual [species] adaptability" makes sense as a conclusion. It seems more likely that if the environment is noise that your current population size becomes more important and predictive than the population sizes of anything else in your (biotic) environment, including your competitors, predators, prey, and mutualists. As such, I don't think the authors have demonstrated this trade-off.*

AR 32 We hope that the revised results summary and our response to RC 30 clarify the trade-off discussed here. As mentioned in our previous response, in a trivial system, increased mutation rates should lead to a decrease in future predictability at all scales (including the scale of a single species). However, as shown in Figure 5, as mutation rates increase into the transition region, predictability increases at the level of a single species. It is not just the case that the current population is more important than anything else in the biotic environment. The amount of information about the future increases even though the mutation rates increase the overall noise in the system. The noise affects the system's higher-order predictability (see Figure 5 Scale 6), as expected. We hope the reviewer can now appreciate this trade-off in light of the response. However, we agree that given the limitations of the model of not capturing trait changes, we have replaced "individual adaptability" with "individual persistence", as that's what is measured in the study.

RC 33 (RC2 11, RC 36) Line 214: "readily applied to real data", but don't the authors need a huge number of simulations of a huge biological length from which they then sample a huge number of subsets of species, judging from what we know from Appendix C and the methods (now lines 283 - 285)? This doesn't seem readily applied, especially as the authors have not dealt with species with varying (individual) generation times.

AR 33 The reviewer is right to point this out, and perhaps "readily applied" is not the best characterization of the methodology, and more work is needed to translate these information-theoretic methods to real-world datasets (L# 227-230). We now acknowledge the need for future work to translate the framework to real-world datasets. We also point to recent advancements in Bayesian methods in ecology as a possible bridge in the efficient estimation of probability distributions needed to estimate information-theoretic measures (see Dormann et. al. 2018).

RC 34 (RC 27) Line 267: It would be nice to have a small addition saying why these parameter ranges are standard (biological? mathematical? stable? dynamic?). The authors appear to have mentioned Di Collobiano et. al. (2003) twice in their response in this context, but this reference does not appear in the main text. Given it came up twice in their response, I was a bit surprised not to find it and its relevance mentioned in the manuscript.

AR 34 As suggested by the reviewer, we have now corrected this oversight by explaining that the parameters chosen here from previous studies produce the intermittent co-evolutionary dynamics observed in fossil records. Second, we now refer to Di Collobinao et al. (2003) in describing the parameters and in the methods where we define mutation rates to point the reader to the relevant literature (L# 283-284).

RC 35 (RC 35) Line 303 - 304: given the increasingly interdisciplinary nature of the paper (see change to Appendix C), the authors should also define the I operator explicitly.

AR 35 We have taken into account the reviewer's suggestion and explicitly defined the mutual information operator near the definitions of the individuality scores (L# 321). In the revision, we have also updated Appendix C to address the concerns raised by the reviewer in previous comments.

RC 36 (RC 36) Line 312: how many simulations is the ensemble?

AR 36 We simulated 10,000 independent simulations with different initial conditions for each mutation rate discussed in the paper. We have now clarified this in the methods (L# 331).

RC 37 Line 313: when sampling different subsets, do the authors sample from the simulations at random times and retrieve that subset of species, or do the authors mean the more literal "choose a random simulation, choose a random subset of species from all subsets of species possible in that simulation's parameters, then look for a time in the simulation where that subset is observed"?

Line 313 - 314: Why should we consider a rank-ordered list rather than a simple random sample? That would seem to bias the results towards integration to me.

AR 37 We address these comments together as they are related. As explained in our previous responses, even with the same interaction matrix to begin with, the system only explores a very small subspace of all possible species in any given simulation, depending upon the initial conditions. However, we also know from previous studies and Appendix E presented in this paper that a similarity exists in the population distribution and the interaction structure among the subset of species the system relaxes to. Therefore, to ensure a like-to-like comparison of subsets across the ensemble of simulations, we rank the species in each simulation.

The top 10^K subsets are selected in decreasing order of population to ensure that the most populous species (which drive the dynamics in steady state) are always picked. Therefore, rank ordering ensures coherence across simulations in an ensemble and ensures that the most important species of the ecosystem are part of the subsets along with other less populous species. We have now added a sentence explaining this rationale in the manuscript (L# 334-335).

Responses to comments from reviewers of paper

Manuscript ID *COMMSBIO-24-3608B*

“Information dynamics and the emergence of high-order individuality in ecosystems”

RC 1 *Reviewer’s Comment*

AR 1 Authors’ Response

We thank the editor and the reviewers for their thoughtful work in reviewing our paper. The comments provided have helped us to substantially improve the revised manuscript. To aid the review process, the modified text has been highlighted using colour blue, in the tracked changes version of the manuscript. In this response, we refer to these changes using line numbers (in short L#), on the left margin of the revised manuscript.

On behalf of all co-authors, I thank you for your attention.

Sincerely,
Hardik Rajpal

Responses to comments from Reviewer 2

RC 1 *I am not sure why, but not all of the changes were marked in blue (minimally in the abstract, change about information processing).*

AR 1 We apologize if we missed to highlight any changes mentioned in the review. We will cross-verify with latexdiff and manually mark the changes made in this revision to ensure that all new changes are highlighted.

RC 2 *Apologies for just noticing this, but the authors have not actually defined which “diversity” they are referring to. I think I implicitly assumed it was species richness, but this should be explicitly stated in the manuscript (line 82?).*

AR 2 We thank the reviewer for noticing this, we have now added a line explaining that we use the exponential of the shannon index, also known as the Hill’s Diversity qD for $q = 1$

RC 3 *The caption of Figure 1 potentially deserves another pass. All of the content is there, but it could be refined slightly. Lines 2 - 4 of Figure 1 maybe should be broken up differently (I would make the colon a full stop and the full stop before implying a comma). The panel labels are not in the panels themselves. The description of Panel A does not mention information storage directly*

*even if the implication is obvious. The authors say "is measured by information transfer", but this seems analogous to saying the distance from the top of my head to my feet is measured by my height; I think most people would just say that that *is* my height.*

AR 3 We understand the concerns raised by the reviewer regarding the framing of the caption of Figure 1. We have now elaborated on the measures used as well as included the panel names in the figure. We thank the reviewer for highlighting this missing information in the figure.

RC 4 *I wanted to pull this out of RC13, but the authors I think use stable to mean a few related things in the manuscript. Each is used consistently and correctly in its context I believe, but this might be a point to make clear at each first usage. Minimally, I would expect line 303 to mention which version of stable the authors mean, since both timesteps and distributions are mentioned in that same sentence.*

AR 4 We understand that the reviewer's concern and have added a clarification around the statistical stability of the distribution in L# 304. Additionally, to clarify the use of word stable in the context of population of species, we have added the clarification at the point of first use as referring to temporal persistence (L# 61).

RC 5 *I also realised as I was inspecting Figure 4 that the authors have not listed exactly what they are plotting throughout in each caption. This should hopefully be a quick fix. (E.g., Figure 4, mean and percentiles? Figure 6, mean and stand deviations?)*

AR 5 We have now updated the figure captions of Figures 3-7 adding in the details that the lines represent the ensemble averaged values and the errorband corresponds to the standard deviation.

RC 6 *Looking at the analysis code provided and at Figure A.1, the authors should probably define what they mean by core in Figure A.1. It might also benefit biologically minded readers to remind the reader that reproduction rate in this figure is after taking account of species interactions, etc.*

AR 6 In order to avoid further linguistic confusion regarding the core-species, we have replaced it with "highly populated existing species" in the caption. Upon reviewer's suggestion we have also added the definition of the reproduction rate plotted in the figure.

RC 7 *Thank you to the authors for making their analysis code available. The authors should cite JIDT (Lizier's 2014 publication, linked on the JIDT Github page) and mention it and their analysis Github in the manuscript. Aside from that, their code makes it clear that they have assumed Gaussian copulae for their data. Could the authors plot or test their data to show that this assumption is valid and double check (but do not need to show) that there is no further temporal structure?*

AR 7 We apologize for not adding the analysis repository in the main manuscript in the previous round of revisions. We have now added the reference to the JIDT estimators used along with the paper and the GitHub repository of the analysis. The Gaussian Copula transformation for estimation of MI is agnostic of the marginal distributions of the variables, i.e. the MI of the

variables does not depend upon the marginal distribution and only on the copula. Therefore, the marginal distributions do not need to be gaussian for estimating MI. This is clearly demonstrated on page 13-14 and Figure 8 of Ince et. al.

In short the joint entropy of two variables X and Y can be written in terms of their joint copula as,

$$H(X, Y) = H(X) + H(Y) + H(c)$$

Where $H(c)$ represents the entropy of the copula. Finally substituting this definition of the joint entropy in the equation for Mutual Information, $I(X; Y) = H(X) + H(Y) - H(X, Y) = -H(c)$. Since the Gaussian Copula transformation preserves the copula of the original data, the marginal distributions of the original data can be non-Gaussian for this implementation.

However, the copula transformation assumes a Gaussian joint distribution in the transformed data. It can be seen that in Figure 1, the correlation structure across the diagonals is preserved in the transformed data. In order to further validate the applicability of Gaussian copula transformation, we compared the scale 1 mutual information values in time obtained from the Gaussian estimator to the non-parameteric Kraskov-Stogbauer-Grassberger (KSG) estimator. Although computationally expensive the KSG estimator does not assume an underlying probability distribution of the data. When looking at the Pearson’s correlation coefficient across generations (see Figure 2). We find a high and stable correlation between the two estimators across generations. Since the KSG estimator doesn’t scale data-efficiently for increasing scales, we believe the Gaussian estimators deployed on the copula transformed data is a reasonable choice for this dataset.

We hope that these tests address the reviewer’s concerns regarding the validity of Gaussian copula transformation and the stability of the estimator over time.

RC 8 *The authors show in RC 13 that at various time points the simulations are close in distribution at the population level and state that they are happy with the temporal dynamics in RC 14. (I would not be happy in their shoes and would only begin to feel comfortable around 40,000 if I were using their plots, especially as scale 6 still appears to be drifting to me at that time and the authors are using up to scale 15.) I think the information from these plots, combined with mentioning the approximately Gaussian structure address my original concerns about appendix C and should be included therein.*

AR 8 We understand the reviewer’s concerns regarding the slow-nonstationarity of the Tangled Nature Model. The dynamics evolves into stable configuration over generations and the system relaxes very slowly into a steady state. To avoid the effect of this non-stationarity in mutual information estimation, we deployed the ensemble method discussed in Appendix C. As reviewer rightly points out the drift in the MI values settles down around 40,000 generations 3a However, given that we are comparing values across scales and mutation rates, it can be seen that the the magnitude of the drift is insignificant as compared to the effect of the scales and the mutation rates. The deviation introduced by the drift (along with simulation noise) is highlighted as errorbands in the individuality scores for all scales 3b. It can be seen that there is no significant scale dependence of the errorbands. As suggested by the reviewer, we have added the temporally estimated individuality scores to the Appendix C.

Figure 1: Top populations across an ensemble of 2,000 simulations at $T_{gen} = 50,000$. Panels (a) and (b) show the marginal and the joint distribution of the raw data. Panels (c) and (d) show the Gaussian Copula-Transformed dataset. It can be seen that the correlation structure along the diagonal is preserved.

RC 9 *I do understand that this is the standard way that Tangled Nature models are analysed, but I do disagree with some points in their response that are worth discussing, but probably outside the realm of the manuscript at this point.*

$T_{gen} = N(t)/p_{kill}$ is the expected number of timesteps that would result in the minimum sufficient number of kill events to remove the existing population assuming that there is no replacement or change in population size. This is slightly different from the average number of timesteps to randomly kill off the existing population, in the sense that an individual through good luck could persist pseudo-indefinitely and be a part of every generation. That is, the average number of timesteps used to randomly kill off a population should be higher than the minimum sufficient number. The latter

Figure 2: Pearson’s correlation coefficient estimated between the mutual information values estimated by the Gaussian estimators on copula transformed data and the non-parametric KSG estimator.

would require labelling each individual in the current generation and counting how long it takes to kill every labelled individual (after which the cycle would repeat). This describes something more in line with a coupon-collecting problem. The former is in line with a simple sum of geometric distributions (over the number of trials, rather than failures). (Using the word "needed" is ambiguous in this context. I only need in theory 100 eggs to put one in each of 100 baskets, but in practice I need many more if there is error.)

This relates to the asynchronous updating comment because the system is not guaranteed to be entirely updated, although I am happy to admit that Metropolis-Hastings asynchronous updating is something I have not studied in detail.

There are a few different timescales in the Tangled Nature simulations depending on perspective. The authors have chosen a meso-timescale in which they try to aggregate over individuals. In contrast, if one was interested in the behaviour of the individuals themselves (which would be silly here!) one could try to look at singular events over an individual’s life span. If one is instead interested in the state of the system as a whole, one could look at the timescale of a species, (if we treat an entire species as a unit of interest, how long does a species last for, e.g., on average) which would be the length of the horizontal lines in Figure 2 of the manuscript. One step broader, one could look at the timescale of the ecosystem, which would be the length of the blocks in Figure 2. I tend to be more interested in the latter two timescales in my work. Given the blocky nature of Figure 2, I surmise that these time scales are not necessarily all strictly linearly related.

That the authors choose a timescale that does not really seem to be what they are interested in is, to my mind, somewhat surprising, but the results from RC13 and 14 and (I assume) the good fit from

(a) Individuality scores for Scale 1 and 6 estimated using the Ensemble method at every 1000th generation.

(b) Average individuality scores at every scale. The errorbands represent the standard deviation due to simulation noise and generational drift.

the Gaussian copulae means it probably is not worth worrying about.

AR 9 We now appreciate the concern of the reviewer regarding timescale. Since, the updates are implemented in the Tangled Nature model at the level of individual members of the species, we had to choose an appropriate timescale in which each existing member of the total population is updated conditioned on the p_{kill} . Thus, the generational time scale presented here is the primary timescale over which the dynamics act and other timescales (i.e. the meso-level stable states) emerge out of this primary dynamics.

As the reviewer suggests that at the meso-scale we could observe the block of co-existing species as a single unit and analyse the dynamics at the scale of metastable states. However, this analysis would require a different level of coarse-graining and mathematical tools to study the collective evolution of species. It would be interesting to study either the Tangled Nature Model or other models of evolution that primarily simulate the dynamics directly at the level of groups of species and ecosystems instead of individual members to observe the differences in information propagation at different timescales.

We would try to explore this interesting line of inquiry in a future study with an appropriate model. We have added a line in the discussion highlighting the need of such an analyses (L # 235-237).

RC 10 *Taking on board the authors' points, I do wonder what scales they should be calling out in the text. The image (Figure 4) and their explanation seem to indicate a range of possible ranges, with the largest as 3 - 13 (Figure 4 0.005), and the smallest listed as 5-8 (lines 201 and 206), but I think I also saw 5-9 in the caption and 4-7 in AR21. Are these different ranges intentional? Additionally, I was wondering if there is a statistical test for this, although I understand if there is not.*

AR 10 We thank the reviewer for noticing this in the consistency in the ranges mentioned across the manuscript. These are not intentional and were introduced locally in the context of the discussion. We agree that we should be focusing the reader on a consistent range throughout the manuscript. Therefore, we have replaced the range of 3-9 species as increased individuality is seen in this range across most mutation rates before the error-transition region.

RC 11 *RC23: I can indeed appreciate the nuances that the authors are indicating here, but these details should be moved into appendix E!*

AR 11 We have now updated Appendix E to expand on the reason for using positive weights and added an explanation for the case of negative-only interactions among the species.

RC 12 *I do suspect the authors are a good deal more familiar with the concept of entropy than the audience that I think (ecologists and biological modellers reading Communications Biology) would be engaging with this manuscript! While I appreciate the details presented in their response (and perhaps those details should feature in the introduction), they don't actually answer the question I was intending to pose in RC27.*

I had intended to ask how, if I were to try to put this into practice for a different system, would I interpret the results quantitatively? Are these values all strictly to be interpreted as relative and not to be compared to a different model or system (or even between figures?)? If I look at Figure 6, Species \rightarrow Environment, Scale 6, rates 0.01 and 0.02 and surmise that I have a significant difference here, is the difference of ~ 0.03 a big enough value to be excited by, or should I expect other entropies to be so much larger that the difference is effectively meaningless in practice? This is asking a question similar to what is a "big" effect size elsewhere (e.g. discussion of Cohen's d and the like).

AR 12 We apologise for misunderstanding the reviewer on this point. We now understand that the comment pertains to comparing the absolute values across different simulations and other datasets, to which the framework may be applied. This is an important point, in this study, where the rest of the parameters of the dynamics are fixed, the values can be directly compared across different mutation rates (after normalising for different scales). We present the raw values with errorbars and the distribution of values, say for two different mutation rates can be used to estimate effect sizes using Cohen's d or other statistical tests. However, if we change the interaction structure, like in the case of the neutral model, the dynamics becomes significantly more correlated as compared

to the original model and the effect sizes is no longer comparable.

To address this gap, there has been some work on developing appropriate normalisation to compare information measures across datasets (e.g. Liardi et al. 2024). Such a framework will need to be adapted to the individuality measures, where all values are compared to an uncorrelated case with similar joint entropy. However, such an analysis is beyond the scope of this manuscript. We have mentioned in the discussion the need to develop such normalisation methodologies (L# 234).

RC 13 *I don't necessarily find AR37 reassuring, but I can understand what the authors are attempting (in the sense that they think there is something in their data and are trying to bring it out). I'd like some of this brought out in the manuscript so that readers can decide for themselves to be bothered or not. I'd recommend refining the new lines 334 - 335 sentence to something like "The rank ordering of subsets of species makes it more likely to sample subsets with similar population distributions and interaction structures, improving comparison across different subset sizes." This makes the authors' justification clearer in the paper itself. Personally, I believe some of my colleagues would be more interested in the unranked samples or in comparing ranked and unranked samples.*

AR 13 We have now rephrased the lines in the methods (L# 344-345) to add clarity

RC 14 *The authors say in a few places, usually in figures, "the paper", but I thought it was in some places ambiguous as to whether it was referring to a citation or the main text. These places would benefit with a link to a section (or citation, as necessary).*

AR 14 Considering the reviewer's suggestion, we have replaced the paper with the section's name whenever pointing to the current manuscript (especially in the figure captions). We have also made sure that relevant citations are present whenever other papers are referred to in the text.

RC 15 *Lines 77 - 78 are also a bit tricky to read. Maybe change "Therefore, triggering mutations among existing species..." to "The population of species experience mutations in response..."*

AR 15 We agree with the author's phrasing and the enhanced readability it enables. We have replaced the phrase in L# 78 accordingly.

RC 16 *Line 142: there is a missing parenthesis.*

AR 16 We have now added the missing parenthesis in L# 145

RC 17 *Figure 4 is missing a space at the beginning of its caption.*

AR 17 The missing space is now added in the caption.

RC 18 *Figure 5 has "Scal1" and inconsistent capitalisation compared to the rest of the manuscript.*

AR 18 We thank the reviewer for noticing the error and inconsistent capitalization. We have now fixed the spelling and used lowercase "scale" throughout the manuscript.

RC 19 *Line 181 seems to have a full stop where I would expect a comma.*

AR 19 The reviewer is right to point out that our phrasing was a bit unfinished. We have now modified the sentence for clarity (L#183-184).

RC 20 *Line 192: should "as mutations increase" be "as mutation rates increase"?*

AR 20 We have now corrected this error in the revised manuscript.

RC 21 *Line 193 seems to have an odd phrasing of cause and effect. I would have thought the shorter metastable states observed in the transition region reduces predictability at higher organisational levels and enhances predictability at the single species level instead. Instead, the authors have predictability (something I'd expect to be an observation about a system) driving the system.*

AR 21 The reviewer is right to point out that the previous framing implied opposite causality than what is observed. We have now corrected this framing in the revised manuscript (L# 195-197).

RC 22 *Line 210 is missing a space ("since").*

AR 22 Missing space is now added.

RC 23 *Line 222 "a stable population" or "stable populations"?*

AR 23 The reviewer is correct. We have now updated the text to "stable populations".

RC 24 *Line 267 seems to have a full stop where I would expect a comma.*

AR 24 We have now rephrased the sentences to enhance readability (L# 268 - 269).

RC 25 *Line 301, Tangled Nature was left lowercase.*

AR 25 We have now corrected this error in the revised manuscript L# 304

RC 26 *The authors will want to be careful regarding equation 1 in the final form, as it showed up differently between the version with tracked changes and the one without.*

AR 26 We would like to confirm that the equation in the non-tracked changes version of the manuscript is correct. The tracked changes version seems to have misplaced the labels corresponding to the parts of the equations.

RC 27 *Line 326, should it be "known about" or "known from" rather than "known by"?*

AR 27 We have changed the phrasing to "known from", which, as the reviewer points out, is a better fit.

RC 28 *Appendix A: Hamming should be capitalized.*

AR 28 We thank the reviewer for pointing this out, we have now corrected this error.

RC 29 *Appendix B: the authors should probably match the colour scheme to Figure 4.*

AR 29 We have now updated the colour scheme of the Colonial Individuality and the Environment determined individuality plots to that of Figure 4.

RC 30 *Appendix C: the first line should have Tangled Nature capitalized and the full stop replaced by a comma.*

AR 30 We have now fixed the capitalization and phrasing at the beginning of Appendix C.

RC 31 *Appendix E: 2nd Paragraph, 1st Sentence: this probably should read: "We consistently find a very peculiar interaction structure when looking in systems with mutation rates for which higher-order organization is present."*

AR 31 We have now rephrased the sentence as suggested by the reviewer.

RC 32 *Appendix F: a space is missing in "equation6".*

AR 32 The missing space is now added for the reference.